# Synthesizing theories of human language with Bayesian program induction

Kevin Ellis [1] ✉, Adam Albright [2], Armando Solar-Lezama[3], Joshua B. Tenenbaum[4] & Timothy J. O'Donnell[5,6,7]

Automated, data-driven construction and evaluation of scientific models and theories is a long-standing challenge in artificial intelligence. We present a framework for algorithmically synthesizing models of a basic part of human language: morpho-phonology, the system that builds word forms from sounds. We integrate Bayesian inference with program synthesis and representations inspired by linguistic theory and cognitive models of learning and discovery. Across 70 datasets from 58 diverse languages, our system synthesizes human-interpretable models for core aspects of each language's morphophonology, sometimes approaching models posited by human linguists. Joint inference across all 70 data sets automatically synthesizes a meta-model encoding interpretable cross-language typological tendencies. Finally, the same algorithm captures few-shot learning dynamics, acquiring new morphophonological rules from just one or a few examples. These results suggest routes to more powerful machine-enabled discovery of interpretable models in linguistics and other scientific domains.

A key aspect of human intelligence is our ability to build theories about the world. This faculty is most clearly manifested in the historical development of science[1] but also occurs in miniature in everyday cognition[2] and during childhood development[3]. The similarities between the process of developing scientific theories and the way that children construct an understanding of the world around them have led to the child-as-scientist metaphor in developmental psychology, which views conceptual changes during development as a form of scientific theory discovery[4,5]. Thus, a key goal for both artificial intelligence and computational cognitive science is to develop methods to understand—and perhaps even automate—the process of theory discovery[6–13].

In this paper, we study the problem of AI-driven theory discovery, using human language as a testbed. We primarily focus on the linguist's construction of language-specific theories, and the linguist's synthesis of abstract cross-language meta-theories, but we also propose connections to child language acquisition. The cognitive sciences of language have long drawn an explicit analogy between the working scientist constructing grammars of particular languages and the child learning their languages[14,15]. Language-specific grammar must be formulated within a common theoretical framework, sometimes called universal grammar. For the linguist, this is the target of empirical inquiry, for the child, this includes those linguistic resources that they bring to the table for language acquisition.

Natural language is an ideal domain to study theory discovery for several reasons. First, on a practical level, decades of work in linguistics, psycholinguistics, and other cognitive sciences of language provide diverse raw material to develop and test models of automated theory discovery. There exist corpora, data sets, and grammars from a large variety of typologically distinct languages, giving a rich and varied testbed for benchmarking theory induction algorithms. Second, children easily acquire language from quantities of data that are

[1]Department of Computer Science, Cornell University, Ithaca, NY 14850, USA. [2]Department of Linguistics, Massachusetts Institute of Technology, Cambridge, MA 02139, USA. [3]Department of Electrical Engineering and Computer Science, Massachusetts Institute of Technology, Cambridge, MA 02139, USA. [4]Department of Brain and Cognitive Sciences, Massachusetts Institute of Technology, Cambridge, MA 02139, USA. [5]Department of Linguistics, McGill University, Montréal, QC H3A0G4, Canada. [6]Canada CIFAR AI Chair, Montréal, QC H2S3H1, Canada. [7]Quebec Artificial Intelligence Institute (Mila), Montréal, QC H2S3H1, Canada. ✉e-mail: kellis@cornell.edu

modest by the standards of modern artificial intelligence[16–18]. Similarly, working field linguists also develop grammars based on very small amounts of elicited data. These facts suggest that the child-as-linguist analogy is a productive one and that inducing theories of language is tractable from sparse data with the right inductive biases. Third, theories of language representation and learning are formulated in computational terms, exposing a suite of formalisms ready to be deployed by AI researchers. These three features of human language—the availability of a large number of highly diverse empirical targets, the interfaces with cognitive development, and the computational formalisms within linguistics—conspire to single out language as an especially suitable target for research in automated theory induction.

Ultimately, the goal of the language sciences is to understand the general representations, processes, and mechanisms that allow people to learn and use language, not merely to catalog and describe particular languages. To capture this framework-level aspect of the problem of theory induction, we adopt the paradigm of *Bayesian Program Learning* (BPL: see ref. [19]). A BPL model of an inductive inference problem, such as theory and grammar induction, works by inferring a generative procedure represented as a symbolic program. Conditioned on the output of that program, the model uses Bayes' rule to work backward from data (program outputs) to the procedure that generated it (a program). We embed classic linguistic formalisms within a programming language provided to a BPL learner. Only with this inductive bias can a BPL model then learn programs capturing a wide diversity of natural language phenomena. By systematically varying this inductive bias, we can study elements of the induction problem that span multiple languages. By doing hierarchical Bayesian inference on the programming language itself, we can also automatically discover some of these universal trends. But BPL comes at a steep computational cost, and so we develop new BPL algorithms which combine techniques from program synthesis with intuitions drawn from how scientists build theories and how children learn languages.

We focus on theories of natural language *morpho-phonology*—the domain of language governing the interaction of word formation and sound structure. For example, the English plurals for *dogs*, *horses*, and *cats* are pronounced /dagz̲/, /hɔrsə̲z/, and /kæts̲/, respectively (plural suffixes underlined; we follow the convention of writing phoneme sequences between slashes). Making sense of this data involves realizing that the plural suffix is actually /z/ (part of English *morphology*), but this suffix transforms depending on the sounds in the stem (English *phonology*). The suffix becomes /əz/ for *horses* (/hɔrsəz/) and other words ending in stridents such as /s/ or /z/; otherwise, the suffix becomes /s/ for *cats* (/kæts/) and other words ending in unvoiced consonants. Full English morphophonology explains other phenomena such as syllable stress and verb inflections. Figure 1a–c shows similar phenomena in Serbo-Croatian: just as English morphology builds the plural by adding /z/, Serbo-Croatian builds feminine forms by adding /a/. Just as English phonology inserts /ə/ at the end of /hɔrsəz/, Serbo-Croatian modifies a stem such as /yasn/ by inserting /a/ to get /yasan/. Discovering a language's morphophonology means inferring its stems, prefixes, and suffixes (its *morphemes*), and also the phonological rules that predict how concatenations of these morphemes are actually pronounced. Thus acquiring the morphophonology of a language involves solving a basic problem confronting both linguists and children: to build theories of the relationships between form and meaning given a collection of utterances, together with aspects of their meanings.

We evaluate our BPL approach on 70 data sets spanning the morphophonology of 58 languages. These data sets come from phonology textbooks: they have high linguistic diversity, but are much simpler than full language learning, with tens to hundreds of words at most, and typically isolate just a handful of grammatical phenomena. We will then shift our focus from linguists to children, and show that the same approach for finding grammatical structure in natural language also captures classic findings in the infant artificial grammar learning literature. Finally, by performing hierarchical Bayesian inference across these linguistic data sets, we show that the model can distill universal cross-language patterns, and express those patterns in a compact, human understandable form. Collectively, these findings point the way toward more human-like AI systems for learning theories, and for systems that learn to learn those theories more effectively over time by refining their inductive biases.

## Results

One central problem of natural language learning is to acquire a grammar that describes some of the relationships between form (perception, articulation, etc.) and meaning (concepts, intentions, thoughts, etc.; Supplementary Discussion 1). We think of grammars as generating form-meaning pairs, $\langle f, m \rangle$, where each form corresponds to a sequence of phonemes and each *meaning* is a set of meaning features. For example, in English, the word *opened* has the form/meaning $\langle /\text{opɛnd}/, [\textbf{stem} : \text{OPEN}; \textbf{tense} : \text{PAST}] \rangle$, which the grammar builds from the form/meaning for *open*, namely $\langle /\text{opɛn}/, [\textbf{stem} : \text{OPEN}] \rangle$, and the past-tense form/meaning, namely $\langle /\text{d}/, [\textbf{tense} : \text{PAST}] \rangle$. Such form-meaning pairs (stems, prefixes, suffixes) live in a part of the grammar called the *lexicon* (Fig. 1c). Together, morpho-phonology explains how word pronunciation varies systematically across inflections, and allows the speaker of a language to hear just a single example of a new word and immediately generate and comprehend all its inflected forms.

## Model

Our model explains a set **X** of form-meaning pairs $\langle f, m \rangle$ by inferring a theory (grammatical rules) **T** and lexicon **L**. For now, we consider maximum aposteriori (MAP) inference—which estimates a single $\langle \textbf{T}, \textbf{L} \rangle$—but later consider Bayesian uncertainty estimates over $\langle \textbf{T}, \textbf{L} \rangle$, and hierarchical modeling. This MAP inference seeks to maximize $P(\textbf{T}, \textbf{L}|\text{UG})\prod_{\langle f, m \rangle \in \textbf{X}} P(f, m|\textbf{T}, \textbf{L})$, where UG (for universal grammar) encapsulates higher-level abstract knowledge across different

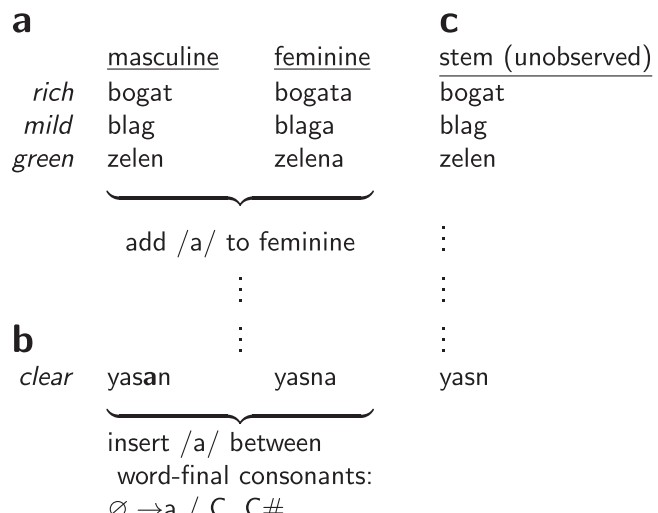

**Fig. 1 | A morpho-phonology problem. a** Serbo-Croatian data (simplified). This language's *morphology* is illustrated for masculine and feminine forms. The data motivate a morphological rule which forms the feminine form by appending /a/. **b** illustrates a counterexample to this analysis: the masculine, feminine forms of *clear* are /yasan/, /yasna/. These pronunciations are explained by Serbo-Croatian *phonology*: the sound /a/ is inserted between pairs of consonants at the end of words, notated ∅ →a / C_C#. This rule requires that the true stem for /yasan/, /yasna/ is /yasn/. **c** shows further stems inferred for this data. These stems are stored in the *lexicon*.

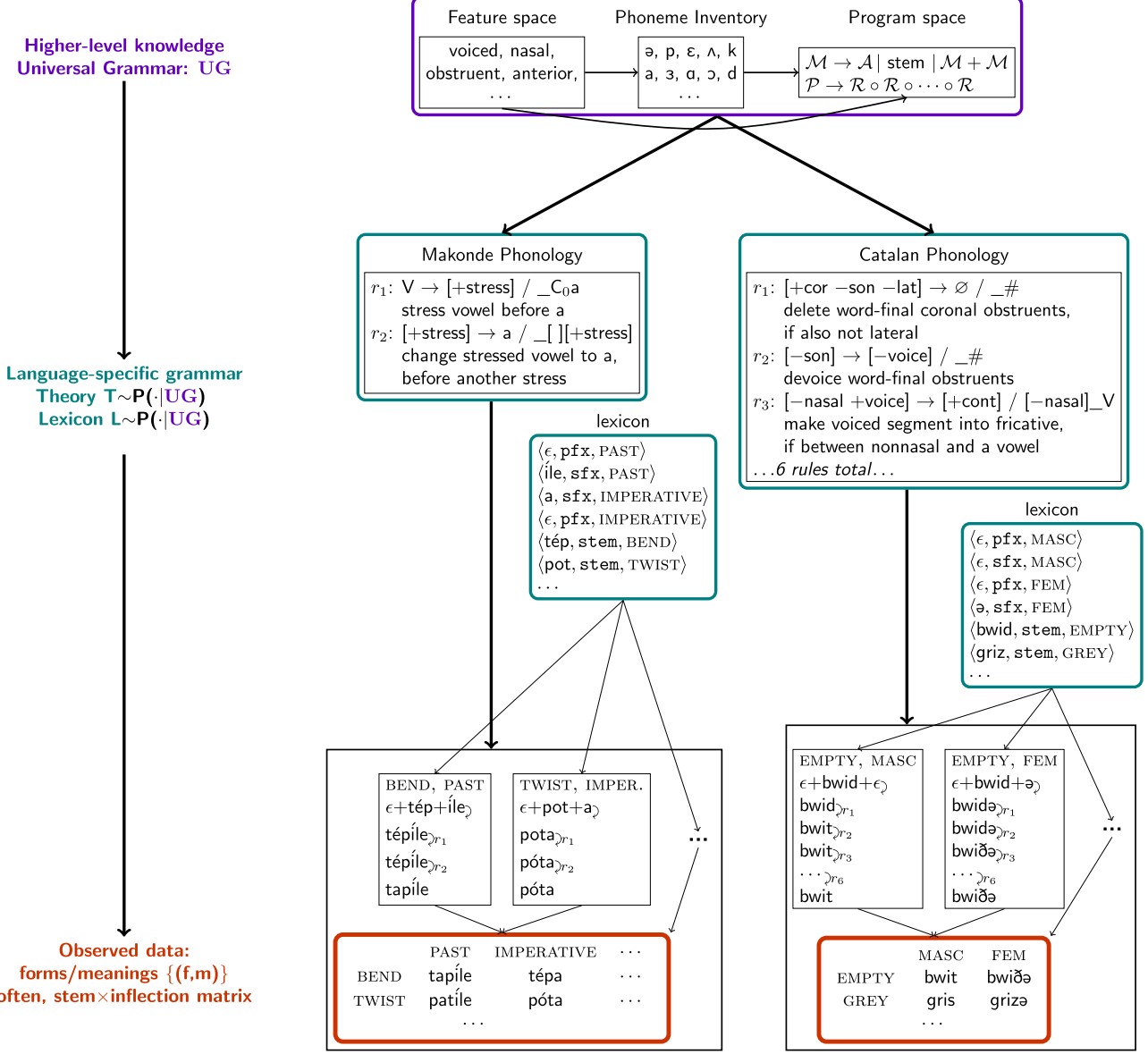

**Fig. 2 | The generative model underlying our approach.** We infer grammars (teal) for a range of languages, given only form/meaning pairs (orange) and a space of programs (purple). Form/meaning pairs are typically arranged in a stem × inflection matrix. For example, the lower right matrix entry for Catalan means we observe the form/meaning pair ⟨/grizə/,[**stem**:GREY; **gender**:FEM]⟩. Grammars include phonology, which transforms concatenations of stems and affixes into the observed surface forms using a sequence of ordered rules, labeled $r_1$, $r_2$, etc. The grammar's lexicon contains stems, prefixes, and suffixes, and morphology concatenates different suffixes/prefixes to each stem for each inflection. $\epsilon$ refers to the empty string. Each rule is written as a context-dependent rewrite, and beneath it, an English description. In the lower black boxes, we show the inferred derivation of the observed data, i.e. the execution trace of the synthesized program. Grammars are expressed as programs drawn from a universal grammar, or space of allowed programs. Makonde and Catalan are illustrated here. Other examples are in Fig. 4 and Supplementary Figs. 1–3.

languages. We decompose each language-specific theory into separate modules for morphology and for phonology (Fig. 2). We handle inflectional classes (e.g. declensions) by exposing this information in the observed meanings, which follows the standard textbook problem structure but simplifies the full problem faced by children learning the language. In principle, our framing could be extended to learn these classes by introducing an extra latent variable for each stem corresponding to its inflectional class. We also restrict ourselves to concatenative morphology, which builds words by concatenating stems, prefixes, and suffixes. Nonconcatenative morphologies[20]—such as Tagalog's reduplication, which copies syllables—are not handled. We assume that each morpheme is paired with a *morphological category*: either a prefix (pfx), suffix (sfx), or stem. We model the lexicon as a function from pairs of meanings and morphological categories to phonological forms. We model phonology as $K$ ordered rules, written $\{r_k\}_{k=1}^K$, each of which is a function mapping sequences of phonemes to sequences of phonemes. Given these definitions, we express the theory-induction objective as:

$$\arg\max_{\mathbf{T},\mathbf{L}} P(\mathbf{T},\mathbf{L}|\mathrm{UG}) \prod_{(f,m)\in\mathbf{X}} \mathbb{1}\left[f = \mathrm{Phonology}(\mathrm{Morphology}(m))\right]$$

where $\mathrm{Morphology}([\mathbf{stem}: \sigma; i]) = \mathbf{L}(i,\mathrm{pfx}) \cdot \mathbf{L}(\sigma,\mathrm{stem}) \cdot \mathbf{L}(i,\mathrm{sfx})$
*concatenate prefix, stem, suffix*
$\mathrm{Phonology}(m) = r_1(r_2(\cdots r_K(m)\cdots))$
*apply ordered rewrite rules*

(1)

where [**stem**: σ; *i*] is a meaning with stem σ, and *i* are the remaining aspects of meaning that exclude the stem (e.g., *i* could be [**tense**:PAST; **gender**:FEMALE]). The expression $\mathbb{1}[\cdot]$ equals 1 if its argument is true and 0 otherwise. In words, Eq. (1) seeks the highest probability theory that exactly reproduces the data, like classic MDL learners[21]. This equation forces the model to explain every word in terms of rules operating over concatenations of morphemes, and does not allow wholesale memorization of words in the lexicon. Eq. (1) assumes fusional morphology: every distinct combination of inflections fuses into a new prefix/suffix. This fusional assumption can emulate arbitrary concatenative morphology: although each inflection seems to have a single prefix/suffix, the lexicon can implicitly cache concatenations of morphemes. For instance, if the morpheme marking tense precedes the morpheme marking gender, then **L**([**tense**:PAST; **gender**:FEMALE], pfx) could equal **L**([**tense**:PAST], pfx) · **L**([**gender**:FEMALE], pfx). We use a description-length prior for $P(\mathbf{T}, \mathbf{L}|\mathrm{UG})$ favoring compact lexica and fewer, less complex rules (Supplementary Methods 3.4).

The data **X** typically come from a *paradigm matrix*, whose columns range over inflections and whose rows range over stems (Supplementary Methods 3.1). In this setting, an equivalent Bayesian framing ("Methods") permits probabilistic scoring of new stems by treating the rules and affixes as a generative model over paradigm rows.

## Representing rules and sounds

Phonemes (atomic sounds) are represented as vectors of binary features. For example, one such feature is *nasal*, for which e.g. /m/, /n/, are +nasal. Phonological rules operate over this feature space. To represent the space of such rules we adopt the classical formulation in terms of context-dependent rewrites[22]. These are sometimes called *SPE-style rules* since they were used extensively in the *Sound Pattern of English*[22]. Rules are written (focus) → (structural change)/(left trigger)_(right trigger), meaning that the *focus* phoneme(s) are transformed according to the *structural change* whenever the left/right triggering environments occur immediately to the left/right of the focus (Supplementary Fig. 5). Triggering environments specify conjunctions of features (characterizing sets of phonemes sometimes called natural classes). For example, in English, phonemes which are [−sonorant] (such as /d/) become [-voice] (e.g., /d/ becomes /t/) at the end of a word (written #) whenever the phoneme to the left is an unvoiced nonsonorant ([− voice − sonorant], such as /k/), written [-sonorant] → [-voice]/[-voice -sonorant]_#. This specific rule transforms the past tense *walked* from /wɔkd/ into its pronounced form /wɔkt/. The subscript $_0$ denotes zero or more repetitions of a feature matrix, called the "Kleene star" operator (i.e., [+ voice]$_0$ means zero or more repetitions of [+ voice] phonemes). When such rules are restricted to not be able to cyclically apply to their own output, the rules and morphology correspond to 2-way rational functions, which in turn correspond to finite-state transducers[23]. It has been argued that the space of finite-state transductions has sufficient representational power to cover known empirical phenomenon in morpho-phonology and represents a limit on the descriptive power actually used by phonological theories, even those that are formally more powerful, including Optimality Theory[24].

To learn such grammars, we adopt the approach of Bayesian Program Learning (BPL). In this setting, we model each **T** as a program in a programming language that captures domain-specific constraints on the problem space. The linguistic architecture common to all languages is often referred to as universal grammar. Our approach can be seen as a modern instantiation of a long-standing approach in linguistics that adopts human-understandable generative representations to formalize universal grammar[22].

## Inference

We have defined the problem a BPL theory inductor needs to solve, but have not given any guidance on how to solve it. In particular, the space

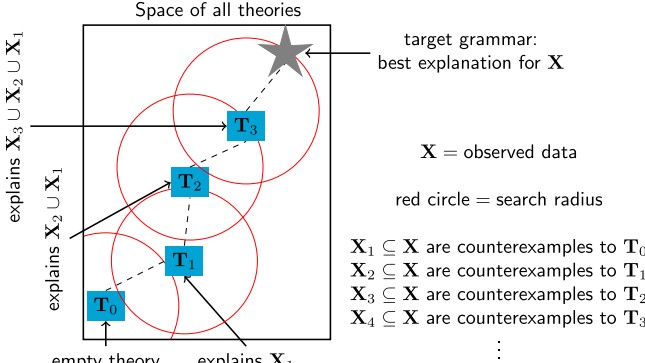

**Fig. 3 | Inference method for Bayesian Program Learning.** To scale to large programs explaining large corpora, we repeatedly search for small modifications to our current theory. Such modifications are driven by counterexamples to the current theory. Blue:grammars. Red: search radius.

of all programs is infinitely large and lacks the local smoothness exploited by local optimization algorithms like gradient descent or Markov Chain Monte Carlo. We adopt a strategy based on constraint-based program synthesis, where the optimization problem is translated into a combinatorial constraint satisfaction problem and solved using a Boolean Satisfiability (SAT) solver[25]. These solvers implement an exhaustive but relatively efficient search and guarantee that, given enough time, an optimal solution will be found. We use the Sketch[26] program synthesizer, which can solve for the smallest grammar consistent with some data, subject to an upper bound on the grammar size (see "Methods").

In practice, the clever exhaustive search techniques employed by SAT solvers fail to scale to the many rules needed to explain large corpora. To scale these solvers to large and complex theories, we take inspiration from a basic feature of how children acquire language and how scientists build theories. Children do not learn a language in one fell swoop, instead progressing through intermediate stages of linguistic development, gradually enriching their mastery of both grammar and lexicon. Similarly, a sophisticated scientific theory might start with a simple conceptual kernel, and then gradually grow to encompass more and more phenomena. Motivated by these observations, we engineered a program synthesis algorithm that starts with a small program, and then repeatedly uses a SAT solver to search for small modifications that allow it to explain more and more data. Concretely, we find a counterexample to our current theory, and then use the solver to exhaustively explore the space of all small modifications to the theory which can accommodate this counterexample. This combines ideas from *counter-example guided inductive synthesis*[26] (which alternates synthesis with a verifier that feeds new counterexamples to the synthesizer) with *test-driven synthesis*[27] (which synthesizes new conditional branches for each such counterexample); it also exposes opportunities for parallelism (Supplementary Methods 3.3). Figure 3 illustrates this incremental, solver-aided synthesis algorithm, while Supplementary Methods 3.3 gives a concrete walk-through of the first few iterations.

This heuristic approach lacks the completeness guarantee of SAT solving: it does not provably find an optimal solution, despite repeatedly invoking a complete, exact SAT solver. However, each such repeated invocation is much more tractable than direct optimization over the entirety of the data. This is because constraining each new theory to be close in theory-space to its preceding theory leads to polynomially smaller constraint satisfaction problems and therefore exponentially faster search times, because SAT solvers scale, in the worst case, exponentially with problem size.

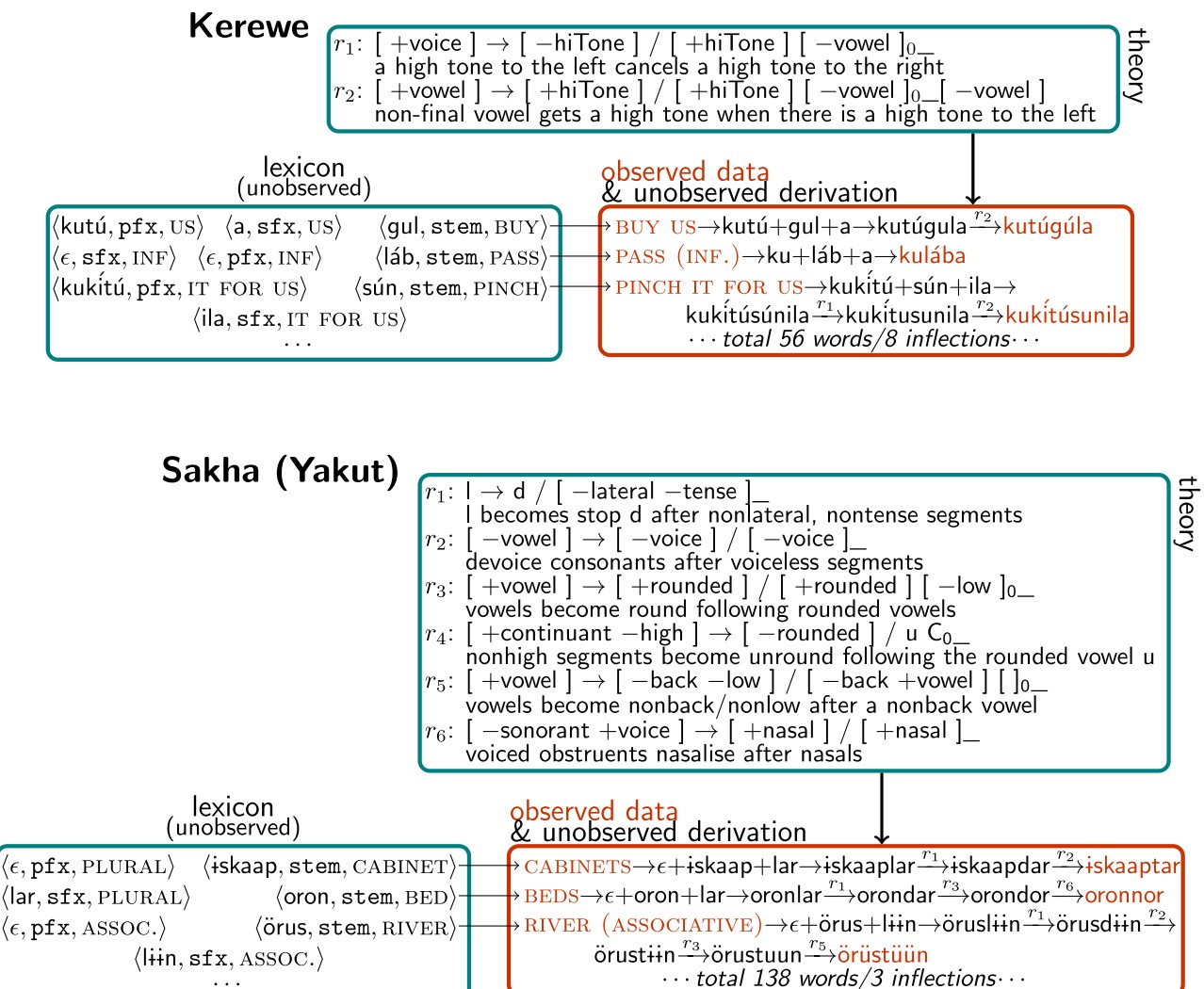

**Fig. 4 | Qualitative results on morpho-phonological grammar discovery illustrated on phonology textbook problems.** The model observes form/meaning pairs (orange) and jointly infers both a language-specific theory (teal; phonological rules labeled $r_1$, $r_2$, ...) and a data set-specific lexicon (teal) containing stems and affixes. Together the theory and lexicon explain the orange data via a derivation where the morphology output (prefix+stem+suffix) is transformed according to the ordered rules. Notice interacting nonlocal rules in Kerewe, a language with tones.

Notice multiple vowel harmony rules in Sakha. Supplementary Figs. 1–3 provide analogous illustrations of grammars with epenthesis (Yowlumne), stress (Serbo-Croatian), vowel harmony (Turkish, Hungarian, Yowlumne), assimilation (Luma-saaba), and representative partial failure cases on Yowlumne and Somali (where it recovers a partly correct rule set that fails to explain 20% of the data, while also illustrating spirantization).

## Quantitative analysis

We apply our model to 70 problems from linguistics textbooks[28–30]. Each textbook problem requires synthesizing a theory of a number of forms drawn from some natural language. These problems span a wide range of difficulties and cover a diverse set of natural language phenomena. This includes tonal languages, for example, in Kerewe, *to count* is /kubala/, but *to count it* is /kukíbála/, where accents mark high tones; languages with vowel harmony, for example Turkish has /el/, /tʃan/ meaning *hand*, *bell*, respectively, and /el-ler/, /tʃan-lar/ for the plurals *hands*, *bells*, respectively (dashes inserted at suffix boundaries for clarity); and many other linguistic phenomena such as assimilation and epenthesis (Fig. 4 and Supplementary Figs. 1–3).

We first measure the model's ability to discover the correct lexicon. Compared to ground-truth lexica, our model finds grammars correctly matching the entirety of the problem's lexicon for 60% of the benchmarks, and correctly explains the majority of the lexicon for 79% of the problems (Fig. 5a). Typically, the correct lexicon for each problem is less ambiguous than the correct rules, and any rules which generate the full data from the correct lexicon must be observationally

equivalent to any ground truth rules we might posit. Thus, agreement with ground-truth lexica should act as a proxy for whether the synthesized rules have the correct behavior on the data, which should correlate with rule quality. To test this hypothesis we randomly sample 15 problems and grade the discovered rules, in consultation with a professional linguist (the second author). We measure both *recall* (the fraction of actual phonological rules correctly recovered) and *precision* (the fraction of recovered rules which actually occur). Rule accuracy, under both precision and recall, positively correlates with lexicon accuracy (Fig. 5c): when the system gets all the lexicon correct, it rarely introduces extraneous rules (high precision), and virtually always gets all the correct rules (high recall).

Prior approaches to morphophonological process learning either abandon theory induction by learning black-box probabilistic models[31], or induce interpretable models but do not scale to a wide range of challenging and realistic data sets. These interpretable alternatives include unsupervised distributional learners, such as the MDL genetic algorithm in Rasin et al.[32], which learns from raw word frequencies. Other interpretable models leverage strong supervision:

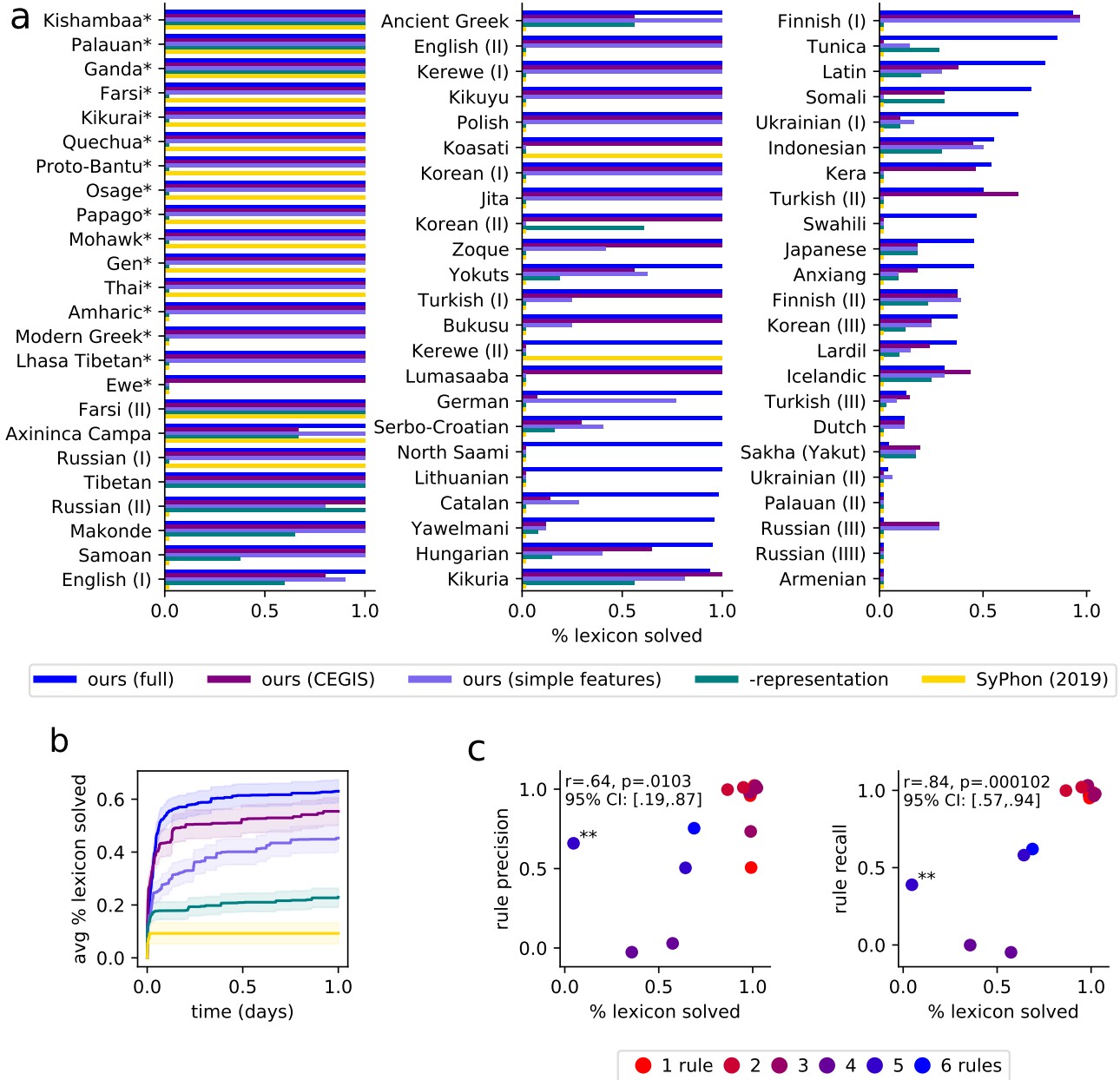

**Fig. 5 | Models applied to data from phonology textbooks. a** Measuring % lexicon solved, which is the percentage of stems that match gold ground-truth annotations. Problems marked with an asterisk are allophony problems and are typically easier. For allophony problems, we count % solved as 0% when no rule explaining an alternation is found and 100% otherwise. For allophony problems, full/CEGIS models are equivalent, because we batch the full problem at once (Supplementary Methods 3). **b** Convergence rate of models evaluated on the 54 non-allophony problems. All models are run with a 24-h timeout on 40 cores. Only our full model can best tap this parallelism (Supplementary Methods 3.3). Our models typically converge within a half-day. SyPhon[36] solves fewer problems but, of those it does

solve, it takes minutes rather than hours. Curves show means over problems. Error bars show the standard error of the mean. **c** Rule accuracy was assessed by manually grading 15 random problems. Both precision and recall correlate with lexicon accuracy, and all three metrics are higher for easier problems requiring fewer phonological rules (red, easier; blue, harder). Requiring an exact match with a ground-truth stem occasionally allows solving some rules despite not matching any stems, as in the outlier problem marked with **. Pearson's *r* confidence intervals (CI) were calculated with two-tailed test. Points were randomly jittered ±0.05 for visibility. Source data are provided as a Source data file.

Albright et al.[33] learns rules from input–outputs, while ref. 34 learns finite state transducers in the same setting. Other works attain strong theoretical learning guarantees by restricting the class of rules: e.g., ref. 35 considers 2-input strictly local functions. These interpretable approaches typically consider 2–3 simple rules at most. In contrast, Goldwater et al.[34] scales to tens of rules on thousands of words by restricting itself to non-interacting local orthographic rules.

Our results hinge on several factors. A key ingredient is a correct set of constraints on the space of hypotheses, i.e. a universal grammar.

We can systematically vary this factor: switching from phonological articulatory features to simpler acoustic features degrades performance (simple features in Fig. 5a, b). Our simpler acoustic features come from the first half of a standard phonology text[28], while the articulatory features come from the latter half, so this comparison loosely models a contrast between novice and expert phonology students (Supplementary Methods 3.5). We can further remove two essential sources of representational power–Kleene star, which allows arbitrarily long-range dependencies, and phonological

**a**

| grammar | example input to learner | inferred grammar | natural language analogues |
|---|---|---|---|
| ABB Marcus et al. 1999. | wofefe lovivi fimumu | $\varnothing \rightarrow \sigma_i \; / \; \sigma_i\_\#$ | reduplication, e.g., Tagalog |
| ABA Marcus et al. 1999. | wofewo lovilo fimufi | $\varnothing \rightarrow \sigma_i \; / \; \sigma_i\sigma\_\#$ | reduplication |
| AAx Gerken 2006. | wowoka loloka fifika | stem+ka $\varnothing \rightarrow \sigma_i \; / \; \#\_\sigma_i$ | reduplication concatenative morphology Mandarin (Fig. 6, lower right) |
| AxA Gerken 2006. | wokawo lokalo fikafi | ka+stem $\varnothing \rightarrow \sigma_i \; / \; \#\_\sigma\sigma_i$ | infixing, e.g. Arabic reduplication |
| Pig Latin | pɪg→ɪgpe latɪn→atɪle æsk→æske | $\varnothing \rightarrow C_i \; / \; \#C_i[\;]_0\_\#$ $\varnothing \rightarrow e \; / \; \_\#$ $C \rightarrow \varnothing \; / \; \#\_$ | child language games |

**b**

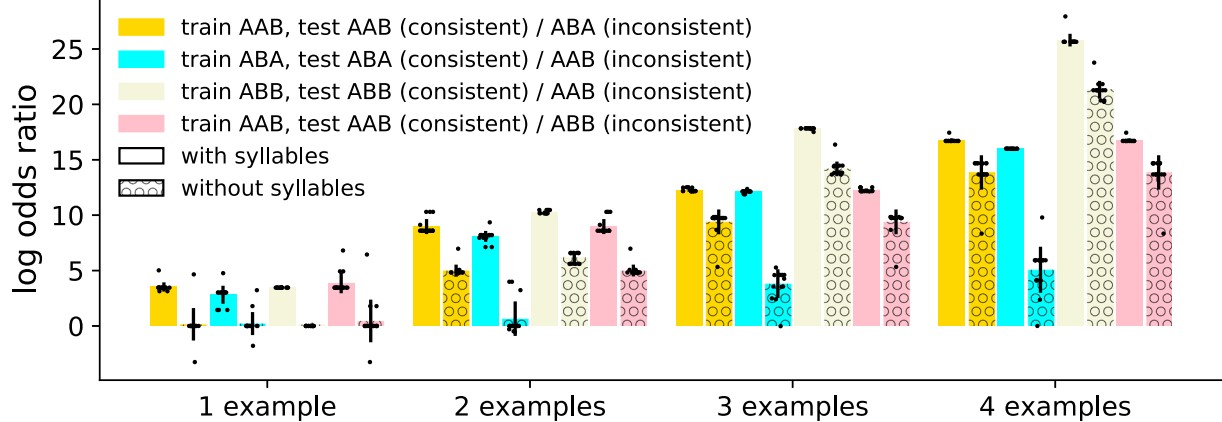

**Fig. 6 | Modeling artificial grammar learning. a** Children can few-shot learn many qualitatively different grammars, as studied in controlled conditions in AGL experiments. Our model learns these as well. Grammar names ABB/ABA/AAx/AxA refer to syllable structure: A/B are variable syllables, and x is a constant syllable. For example, ABB words have three syllables, with the last two syllables being identical. NB: Actual reduplication is subtler than syllable-copying[20]. **b** Model learns to discriminate between different artificial grammars by training on examples of grammar (e.g., AAB) and then testing on either unseen examples of words drawn from the same grammar (consistent condition, e.g., new words following the AAB pattern); or testing on unseen examples of words from a different grammar (inconsistent condition, e.g. new words following the ABA pattern), following the paradigm of ref. 39. We plot log-odds ratios of consistent and inconsistent conditions: log $P$(consistent|train)/$P$(inconsistent|train) ("Methods"), over $n = 15$ random independent (in)consistent word pairs. Bars show mean log odds ratio over these 15 samples, individually shown as black points, with error bars showing stddev. We contrast models using program spaces both with and without syllabic representations, which were not used for textbook problems. Syllabic representation proves important for few-shot learning, but a model without syllables can still discriminate successfully given enough examples by learning rules that copy individual phonemes. See Supplementary Fig. 4 for more examples. Source data are provided as a Source data file.

features, which allow analogizing and generalizing across phonemes. Removing these renders only the simplest problems solvable (·-representation in Fig. 5a, b). Basic algorithmic details also matter. Building a large theory at once is harder for human learners, and also for our model (CEGIS in Fig. 5a, b). The recent SyPhon[36] algorithm strikes a different and important point on the accuracy/coverage tradeoff: it aims to solve problems in seconds or minutes so that linguists can interactively use it. In contrast, our system's average solution time is 3.6 h (Fig. 5b). SyPhon's speed comes from strong independence assumptions between lexica and individual rules, and from disallowing non-local rules. These assumptions degrade coverage: SyPhon fails to solve 76% of our data set. We hope that their work and ours sets the stage for future systems that run interactively while also more fully modeling the richness and diversity of human language.

### Child language generalization

If our model captures aspects of linguistic analysis from naturalistic data, and assuming linguists and children confront similar problems, then our approach should extend to model at least some aspects of the

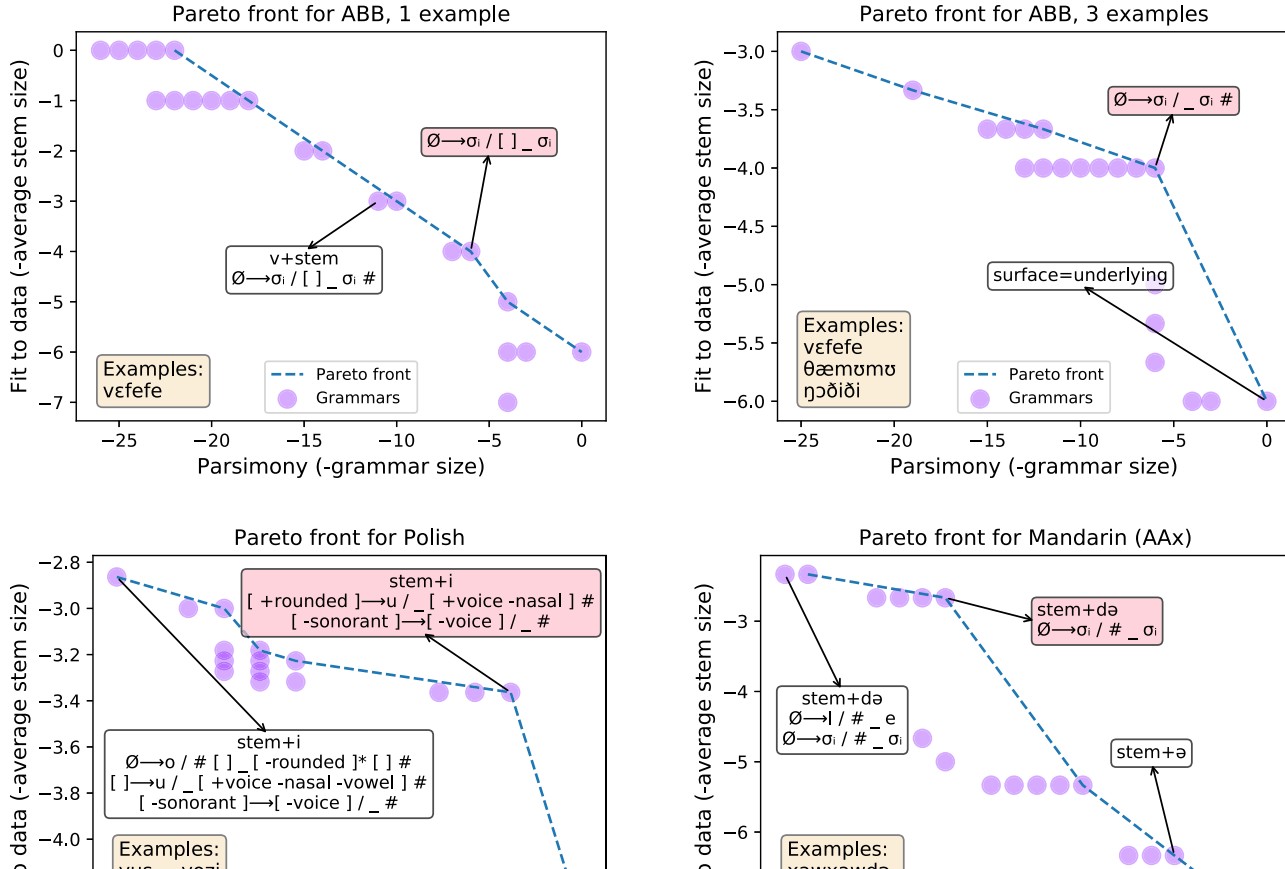

**Fig. 7 | Modeling ambiguity in language learning.** Few-shot learning of language patterns can be highly ambiguous as to the correct grammar. Here we visualize the geometry of generalization for several natural and artificial grammar learning problems. These visualizations are Pareto frontiers: the set of solutions consistent with the data that optimally trade-off between parsimony and fit to data. We show Pareto fronts for ABB (ref. [39]; top two) & AAX (Gerken[53]; bottom right, data drawn from isomorphic phenomena in Mandarin) AGL problems for either one example word (upper left) or three example words (right column). In the bottom left we show the Pareto frontier for a textbook Polish morpho-phonology problem. Rightward on x-axis corresponds to more parsimonious grammars (smaller rule size + affix size) and upward on y-axis corresponds to grammars that best fit the data (smaller stem size), so the best grammars live in the upper right corners of these graphs. N.B.: Because the grammars and lexica vary in size across panels, the x and y axes have different scales in each panel. Pink shade: correct grammar. As the number of examples increases, the Pareto fronts develop a sharp kink around the correct grammar, which indicates a stronger preference for the correct grammar. With one example the kinks can still exist but are less pronounced. The blue lines provably show the exact contour of the Pareto frontier, up to the bound on the number of rules. This precision is owed to our use of exact constraint solvers. We show the Polish problem because the textbook author accidentally chose data with an unintended extra pattern: all stems vowels are /o/ or /u/, which the upper left solution encodes via an insertion rule. Although the Polish MAP solution is correct, the Pareto frontier can reveal other possible analyses such as this one, thereby serving as a kind of linguistic debugging. Source data are provided as a Source data file.

child's linguistic generalization. Studying children (and adult's) learning of carefully constructed artificial grammars has a long tradition in psycholinguistics and language acquisition[37–39], because it permits controlled and careful study of the generalization of language-like patterns. We present our model with the artificial stimuli used in a number of AGL experiments[38–40] (Fig. 6a), systematically varying the quantity of data given to the model (Fig. 6b). The model demonstrates few-shot inference of the same language patterns probed in classic infant studies of AGL.

These AGL stimuli contain very little data, and thus these few-shot learning problems admit a broad range of possible generalizations. Children select from this space of possible generalizations to select the linguistically plausible ones. Thus, rather than producing a single grammar, we use the model to search a massive space of possible grammars and then visualize all those grammars that are Pareto-optimal solutions[41] to the trade-off between parsimony and fit to data.

Here parsimony means size of rules and affixes (the prior in Eq. (10)); fit to data means average stem size (the likelihood in Eq. (10)); and a Pareto-optimal solution is one which is not worse than any other along both these competing axes. Figure 7 visualizes Pareto fronts for two classic artificial grammars while varying the number of example words provided to the learner, illustrating both the set of grammars entertained by the learner and how the learner weighs these grammars against each other. These figures show the exact contours of the Pareto frontier: these problems are small enough that exact SAT solving is tractable over the entire search space, so our heuristic incremental synthesizer is unneeded. With more examples the shape of the Pareto frontier develops a sharp kink around the correct generalization; with fewer examples, the frontier is smoother and more diffuse. By explaining both natural language data and AGL studies, we see our model as delivering on a basic hypothesis underpinning AGL research: that artificial grammar learning must engage some cognitive resource

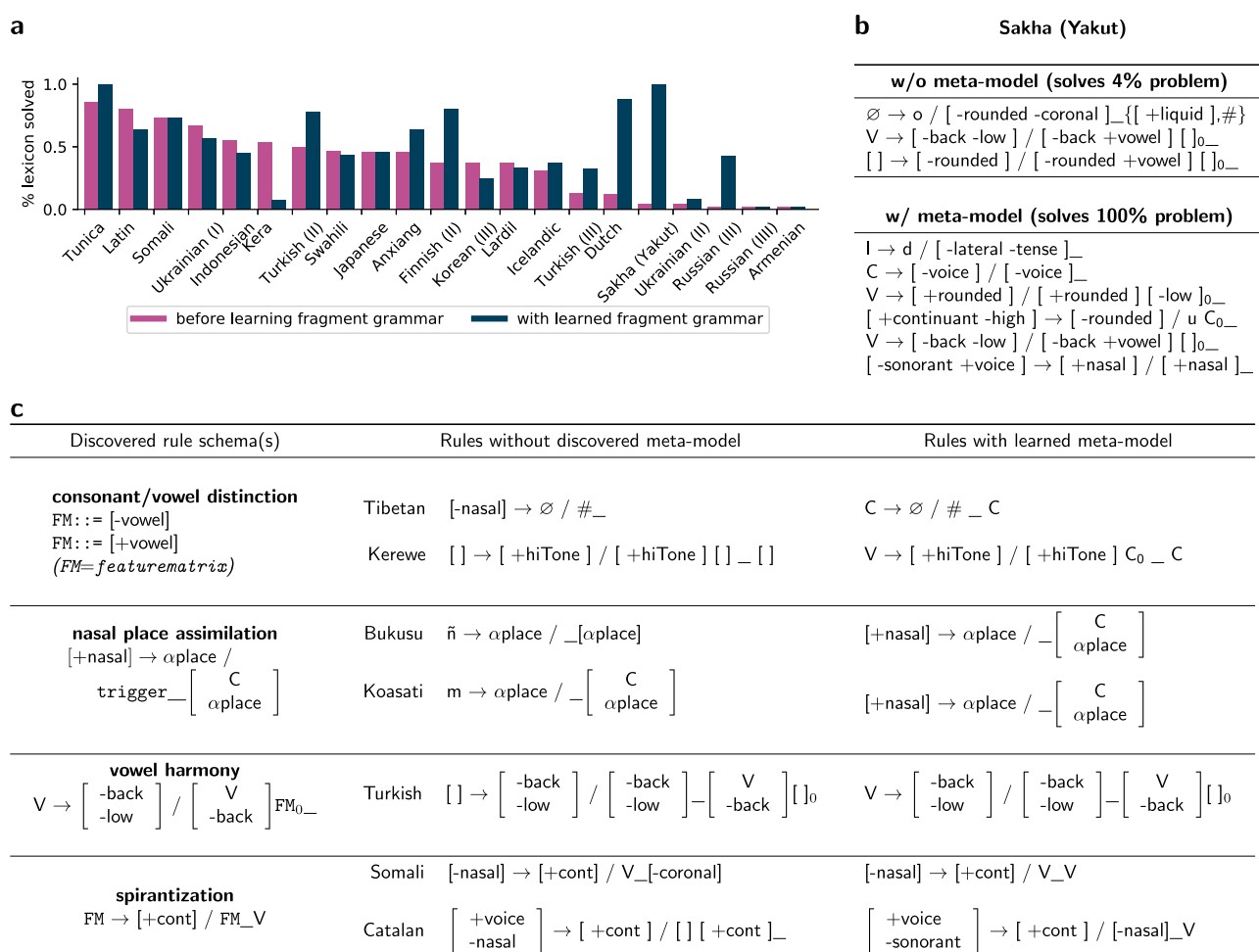

**Fig. 8 | Discovering and using a cross-language metatheory. a** Re-solving the hardest textbook problems using the learned fragment grammar metatheory leads to an average of 31% more of the problem being solved. **b** illustrates a case where these discovered tendencies allow the model to find a set of six interacting rules solving the entirety of an unusually complex problem. **c** The metatheory comprises rule schemas that are human understandable and often correspond to motifs previously identified within linguistics. Left column shows four out of 21 induced rule schemas (Supplementary Fig. 6), which encode cross-language tendencies. These learned schemas include vowel harmony and spirantization (a process where stops become fricatives near vowels). The symbol FM means a slot that can hold any feature matrix, and trigger means a slot that can hold any rule triggering context. Middle column shows model output when solving each language in isolation: these solutions can be overly specific (Koasati, Bukusu), overly general (Kerewe, Turkish), or even essentially unrelated to the correct generalization (Tibetan). Right column shows model output when solving problems jointly with inferring a metatheory. Source data are provided as a Source Data file.

shared with first language acquisition. To the extent that this hypothesis holds, we should expect an overlap between models capable of learning real linguistic phenomena, like ours, and models of AGL phenomena.

## Synthesizing higher-level theoretical knowledge

No theory is built from scratch: Instead, researchers borrow concepts from existing frameworks, make analogies with other successful theories, and adapt general principles to specific cases. Through analysis and modeling of many different languages, phonologists (and linguists more generally) develop overarching meta-models that restrict and bias the space of allowed grammars. They also develop the phonological common sense that allows them to infer grammars from sparse data, knowing which rule systems are plausible based on their prior knowledge of human language, and which systems are implausible or simply unattested. For example, many languages devoice word-final obstruents, but almost no language voices word-final obstruents (cf. Lezgian[42]). This cross-theory common-sense is found in other sciences. For example, physicists know which potential energy functions tend to occur in practice (radially symmetric, pairwise, etc.). Thus a key

objective for our work is the automatic discovery of a cross-language metamodel capable of imparting phonological common sense.

Conceptually, this meta-theorizing corresponds to estimating a prior, **M**, over language-specific theories, and performing hierarchical Bayesian inference across many languages. Concretely, we think of the meta-theory **M** as being a set of schematic, highly reusable phonological-rule templates, encoded as a probabilistic grammar over the structure of phonological rules, and we will estimate both the structure and the parameters of this grammar jointly with the solutions to textbook phonology problems. To formalize a set of meta-theories and define a prior over that set, we use the Fragment Grammars formalism[43], a probabilistic grammar learning setup that caches and reuses fragments of commonly used rule subparts. Assuming we have a collection of $D$ data sets (e.g., from different languages), notated $\{\mathbf{X}^d\}_{d=1}^D$, our model constructs $D$ grammars, $\{\langle \mathbf{T}^d, \mathbf{L}^d \rangle\}_{d=1}^D$, along with a meta-theory **M**, seeking to maximize

$$P(\mathbf{M}) \prod_{d=1}^D P(\mathbf{T}^d, \mathbf{L}^d | \mathbf{M}) P(\mathbf{X}^d | \mathbf{T}^d, \mathbf{L}^d) \qquad (2)$$

where $P(\mathbf{M})$ is a prior on fragment grammars over SPE-style rules. In practice, jointly optimizing over the space of $\mathbf{M}$s and grammars is intractable, and so we instead alternate between finding high-probability grammars under our current $\mathbf{M}$, and then shifting our inductive bias, $\mathbf{M}$, to more closely match the current grammars. We estimate $\mathbf{M}$ by applying this procedure to a training subset comprising 30 problems, chosen to exemplify a range of distinct phenomena, and then applied this $\mathbf{M}$ to all 70 problems. Critically this unsupervised procedure is not given access to any ground-truth solutions to the training subset.

This machine-discovered higher-level knowledge serves two functions. First, it is a form of human understandable knowledge: manually inspecting the contents of the fragment grammar reveals cross-language motifs previously discovered by linguists (Fig. 8c). Second, it can be critical to actually getting these problems correct (Fig. 8a, b and middle column of Fig. 8c). This occurs because a better inductive bias steers the incremental synthesizer toward more promising avenues, which decreases its chances of getting stuck in a neighborhood of the search space where no incremental modification offers improvement.

To be clear, our mechanized meta-theorizing is not an attempt to learn universal grammar (cf. ref. 44). Rather than capture a learning process, our meta-theorizing is analogous to a discovery process that distills knowledge of typological tendencies, thereby aiding future model synthesis. However, we believe that children possess implicit knowledge of these and other tendencies, which contributes to their skills as language learners. Similarly, we believe the linguist's skill in analysis draws on an explicit understanding of these and other cross-linguistic trends.

## Discussion

Our high-level goal was to engineer methods for synthesizing interpretable theories, using morphophonology as a testbed and linguistic analysis as inspiration. Our results give a working demonstration that it is possible to automatically discover human-understandable knowledge about the structure of natural language. Like linguists, optimal inference hinges on higher-level biases and constraints; but the toolkit developed here permits systematic probing of these abstract assumptions and data-driven discovery of cross-language trends. Our work speaks to a long-standing analogy between the problems confronting children and linguists, and computationally cashes out the basic assumptions that underpin infant and child studies of artificial grammar learning.

Within phonology, our work offers a computational tool that can be used to study different grammatical hypotheses: mapping and scoring analyses under different objective functions, and studying the implications of different inductive biases and representations across a suite of languages. This toolkit can spur quantitative studies of classic phonological problems, such as probing extensionally-equivalent analyses (e.g., distinguishing deletion from epenthesis).

More broadly, the tools and approaches developed here suggest routes for machines that learn the causal structure of the world, while representing their knowledge in a format that can be reused and communicated to other agents, both natural and artificial. While this goal remains far off, it is worth taking stock of where this work leaves us on the path toward a theory induction machine: what are the prospects for scaling an approach like ours to other domains of language, or other domains of science more broadly? Scaling to the full linguistic hierarchy—acoustics, phonotactics, syntax, semantics, pragmatics—requires more powerful programming languages for expressing symbolic rules, and more scalable inference procedures, because although the textbook problems we solve are harder than prior work tackles, full morpho-phonology remains larger and more intricate than the problems considered here. More fundamentally, however, we advocate for hybrid neuro-symbolic models[45–47] to capture crisp systematic productivity alongside more graded linguistic generalizations, such as that embodied by distributional models of language structure[48].

Scaling to real scientific discovery demands fundamental innovations, but holds promise. Unlike language acquisition, genuinely new scientific theories are hard-won, developing over timescales that can span a decade or more. They involve the development of new formal substrates and new vocabularies of concepts, such as force in physics and allele in biology. We suggest three lines of attack. Drawing inspiration from conceptual role semantics[49], future automated theory builders could introduce and define new theoretical objects in terms of their interrelations to other elements of the theory's conceptual repertoire, only at the end grounding out in testable predictions. Drawing on the findings of our work here, the most promising domains are those which are solvable, in some version, by both child learners and adult scientists. This means first investigating sciences with counterparts in intuitive theories, such as classical mechanics (and intuitive physics), or cognitive science (and folk psychology). Building on the findings here and in ref. 11, a crucial element of theory induction will be the joint solving of many interrelated model building problems, followed by the synthesis of abstract over-hypotheses that encapsulate the core theoretical principles while simultaneously accelerating future induction through shared statistical strength.

Theory induction is a grand challenge for AI, and our work here captures only small slices of the theory-building process. Like our model, human theorists do craft models by examining experimental data, but they also propose new theories by unifying existing theoretical frameworks, performing thought experiments, and inventing new formalisms. Humans also deploy their theories more richly than our model: proposing new experiments to test theoretical predictions, engineering new tools based on the conclusions of a theory, and distilling higher-level knowledge that goes far beyond what our Fragment-Grammar approximation can represent. Continuing to push theory induction along these many dimensions remains a prime target for future research.

## Methods
### Program synthesis
We use the Sketch[26] program synthesizer. Sketch can solve the following constrained optimization problem, which is equivalent to our goal of maximizing $P(\mathbf{X}|\mathbf{T}, \mathbf{L})P(\mathbf{T}, \mathbf{L}|\mathrm{UG})$:

$$\text{maximize} \quad F(\mathbf{X},\mathbf{T}) = \sum_{k=1}^{K} \log P(r_k|\mathrm{UG}) + \sum_{\langle f,c,m \rangle \in \mathbf{L}} \log P(f|\mathrm{UG})$$

$$\text{subject to} \quad C(\mathbf{X},\mathbf{T}) = \forall \langle f, [\mathbf{stem}:\sigma; i]\rangle \in \mathbf{X}:$$
$$f = r_1(\cdots r_K(\mathbf{L}(\langle i, \mathtt{pfx}\rangle) \cdot \mathbf{L}(\langle \sigma, \mathtt{stem}\rangle) \cdot \mathbf{L}(\langle i, \mathtt{sfx}\rangle))\cdots)$$

$$(3)$$

given observations $\mathbf{X}$ and bound on the number of rules $K$.

Sketch offers an exhaustive search strategy, but we use *incremental* solving in order to scale to large grammars. Mathematically this works as follows: we iteratively construct a sequence of theories $\mathbf{T}_0$, $\mathbf{T}_1$, ... alongside successively larger data sets $\mathbf{X}_0$, $\mathbf{X}_1$, ... converging to the full data set $\mathbf{X}$, such that the $t^{\text{th}}$ theory $\mathbf{T}_t$ explains data set $\mathbf{X}_t$, and successive theories are close to one another as measured by edit distance:

$$\mathbf{X}_{t+1} = \mathbf{X}_t \cup (\text{a set } \mathbf{X}' \subseteq \mathbf{X} \text{ where } \neg C(\mathbf{T}_t, \mathbf{X}')) \qquad (4)$$

$$D_{t+1} = \min_{D} D, \text{ such that}: \exists \mathbf{T} \text{ where } C(\mathbf{T}, \mathbf{X}_{t+1}) \text{ and } d(\mathbf{T}, \mathbf{T}_t) \leq D \qquad (5)$$

$$\mathbf{T}_{t+1} = \arg\max_{\mathbf{T}} F(\mathbf{X}_{t+1}, \mathbf{T}), \text{ such that}: \mathbf{T} \text{ satisfies } C(\mathbf{T}, \mathbf{X}_{t+1}) \text{ and } d(\mathbf{T}, \mathbf{T}_t) \leq D_{t+1}$$

$$(6)$$

where $d(\cdot,\cdot)$ measures edit distance between theories, $D_{t+1}$ is the edit distance between the theory at iteration $t+1$ and $t$, and we use the $t=0$ base cases $\mathbf{X}_0 = \varnothing$ and $\mathbf{T}_0$ is an empty theory containing no rules. We "minibatch" counterexamples to the current theory ($\mathbf{X}'$ in Eq. (4)) grouped by lexeme, and ordered by their occurrence in the data (e.g., if the theory fails to explain *walk/walks/walked*, and this is the next example in the data, then the surface forms of *walk/walks/walked* will be added to $\mathbf{X}_{t+1}$). See Supplementary Methods 3.3.

We implement all models as Python 2.7 scripts that invoke Sketch 1.7.5, and also use Python 2.7 for all data analysis.

## Allophony problems

Allophony problems comprise the observed form-meaning set $\mathbf{X}$, as well as a *substitution*, which is a partial function mapping phonemes to phonemes (see Supplementary Methods 3.1). This mapping operates over phonemes called 'allophones.' The goal of the model is to recover rule(s) which predicts which element of each allophone pair is an underlying form, and which is merely an allophone. The underlying phonemes are allowed in the lexicon, while the other allophones are not allowed in the lexicon and surface only due phonological rules. For example, an allophony substitution could be $\{b\mapsto p, d\mapsto t, g\mapsto k\}$. We extend such substitutions to total functions on phoneme sequences by applying the substitution to phonemes in its domain, and not applying it otherwise. We call this total function $s(\cdot)$. For instance, using the previous example substitution, $s(abkpg) = apkpk$. Solving an allophone problem means finding rules that either map the domain of $s(\cdot)$ to its range ($\mathbf{T}_1$ below), or vice versa ($\mathbf{T}_2$ below):

$$\mathbf{L}_1(m) = s(f) \text{ when } \exists \langle f, m\rangle \in \mathbf{X}$$
$$\mathbf{L}_2(m) = s^{-1}(f) \text{ when } \exists \langle f, m\rangle \in \mathbf{X}$$
$$\text{For } i \in \{1,2\}:$$
$$\mathbf{T}_i = \underset{\mathbf{T}}{\mathrm{argmax}}\, P(\mathbf{T}|\text{UG})P(\mathbf{X}|\mathbf{T},\mathbf{L}_i)\left(= \underset{\mathbf{T}}{\mathrm{argmax}}\, P(\mathbf{X},\mathbf{T},\mathbf{L}_i|\text{UG})\right) \quad (7)$$

## Probabilistic framing

Our few-shot artificial grammar learning simulations require probabilistically scoring held-out unobserved words corresponding to unobserved stems. We now present a refactoring of our Bayesian learning setup that permits these calculations. Given rules $\mathbf{T}$ and lexicon $\mathbf{L}$, we define a likelihood $P^{\text{Lik}}$ over a paradigm matrix $\mathbf{X}$ when the data $\mathbf{X}$ contain stems disjoint from those in $\mathbf{L}$:

$$P^{\text{Lik}}(\mathbf{X}|\mathbf{T},\mathbf{L}) = \sum_{\mathbf{L}'} P(\mathbf{X}|\mathbf{T},\mathbf{L}\cup\mathbf{L}')P(\mathbf{L}'|\text{UG}) \quad (8)$$

where $\mathbf{L}'$ ranges over lexica which assign forms to the stems present in $\mathbf{X}$, i.e. $\mathbf{L}' \ni \langle f', \texttt{stem}, \sigma\rangle$ iff $\mathbf{X} \ni \langle f, [\textbf{stem}: \sigma; i]\rangle$ for some surface form $f$ and some underlying form $f'$. The term $P^{\text{Lik}}$ can be lower bounded by taking the most likely underlying form for each stem:

$$P^{\text{Lik}}(\mathbf{X}|\mathbf{T},\mathbf{L}) \geq \max_{\mathbf{L}'} P(\mathbf{X}|\mathbf{T},\mathbf{L}\cup\mathbf{L}')P(\mathbf{L}'|\text{UG}) \quad (9)$$

This lower bound will be tightest when each paradigm row admits very few possible stems. Typically only one stem per row is consistent with the rules and affixes, which justifies this bound.

The connection between the Bayesian likelihood $P^{\text{Lik}}$ and the MAP objective (Eq. (1)) can be seen by partitioning the lexicon into affixes (in $\mathbf{L}$) and stems (in $\mathbf{L}'$), which also decomposes the objective into a parsimony-favoring prior and a fit-to-data favoring likelihood term:

$$\max_{\mathbf{T},\mathbf{L}} P(\mathbf{T},\mathbf{L}|\text{UG})P^{\text{Lik}}(\mathbf{X}|\mathbf{T},\mathbf{L}) \geq \max_{\mathbf{T},\mathbf{L},\mathbf{L}'} \underbrace{P(\mathbf{T},\mathbf{L}|\text{UG})}_{\text{prior}} \underbrace{P(\mathbf{L}'|\text{UG})P(\mathbf{X}|\mathbf{T},\mathbf{L}\cup\mathbf{L}')}_{\text{likelihood}} \quad (10)$$

$$= \max_{\mathbf{T},\mathbf{L},\mathbf{L}'} \underbrace{P(\mathbf{T},\mathbf{L}\cup\mathbf{L}'|\text{UG})P(\mathbf{X}|\mathbf{T},\mathbf{L}\cup\mathbf{L}')}_{= \text{Eq.1 w/lexicon set to } \mathbf{L}\cup\mathbf{L}'} \quad (11)$$

## Few-shot artificial grammar learning

We present our system with a training set $\mathbf{X}_{\text{train}}$ of words from a target language, such as the ABA language (e.g., /wofewo/, /mikami/, ...). We model this training set as a paradigm matrix with a single column (single inflection), with each word corresponding to a different stem (a different row in the matrix). Then we compute the likelihood assigned to a held-out word $\mathbf{X}_{\text{test}}$ either consistent with the target grammar (e.g., following the ABA pattern) or inconsistent with the target grammar (e.g., following the ABB pattern, such as /wofefe/, /mikaka/, ...). The probability assigned to a held-out test word, conditioned on the training set, is approximated by marginalizing over the Pareto-optimal grammars for the train set, rather than marginalizing over all possible grammars:

$$P(\mathbf{X}_{\text{test}}|\mathbf{X}_{\text{train}}) = \sum_{\mathbf{T},\mathbf{L}} P(\mathbf{T},\mathbf{L}|\mathbf{X}_{\text{train}})P^{\text{Lik}}(\mathbf{X}_{\text{test}}|\mathbf{T},\mathbf{L})$$

$$\approx \sum_{\langle\mathbf{T},\mathbf{L}\rangle \in \text{ParetoFrontier}(\mathbf{X}_{\text{train}})} \frac{P(\mathbf{X}_{\text{train}},\mathbf{T},\mathbf{L})P^{\text{Lik}}(\mathbf{X}_{\text{test}}|\mathbf{T},\mathbf{L})}{\sum_{\langle\mathbf{T}',\mathbf{L}'\rangle \in \text{ParetoFrontier}(\mathbf{X}_{\text{train}})} P(\mathbf{X}_{\text{train}},\mathbf{T}',\mathbf{L}')} \quad (12)$$

which relies on the fact that Sketch has out-of-the-box support for finding Pareto-optimal solutions to multiobjective optimization problems[26]. We approximate the likelihood $P^{\text{Lik}}(\mathbf{X}_{\text{test}}|\mathbf{T},\mathbf{L})$ using the lower bound in Eq. (9), equivalently finding the shortest stem which will generate the test word $\mathbf{X}_{\text{test}}$, given the affixes in $\mathbf{L}$ and the rules in $\mathbf{T}$.

## Synthesizing a metatheory

At a high level, inference of the cross-language fragment grammar works by maximizing a variational-particle[50] lower bound on the joint probability of the metatheory $\mathbf{M}$ and the $D$ data sets, $\{\mathbf{X}^d\}_{d=1}^{D}$:

$$\log P\left(\mathbf{M},\{\mathbf{X}^d\}_{d=1}^{D}\right) \geq \log P(\mathbf{M}) + \sum_{d=1}^{D} \log \sum_{\substack{\langle\mathbf{T}_d,\mathbf{L}_d\rangle \in \\ \text{support}[Q_d(\cdot)]}} P\left(\mathbf{X}^d|\mathbf{T}_d,\mathbf{L}_d\right)P(\mathbf{T}_d,\mathbf{L}_d|\mathbf{M}) \quad (13)$$

where this bound is written in terms of a set of variational approximate posteriors, $\{Q_d\}_{d=1}^{D}$, whose support we constrain to be small, which ensures that the above objective is tractable. We alternate maximization with respect to $\mathbf{M}$ (i.e., inferring a fragment grammar from the theories in the supports of $\{Q_d\}_{d=1}^{D}$), and maximization with respect to $\{Q_d\}_{d=1}^{D}$ (i.e., finding a small set of theories for each data set that are likely under the current $\mathbf{M}$). Our lower bound most increases when the support of each $\{Q_d\}_{d=1}^{D}$ coincides with the top-$k$ most likely theories, so at each round of optimization, we ask the program synthesizer to find the top $k$ theories maximizing $P(\mathbf{X}^d|\mathbf{T}_d,\mathbf{L}_d)P(\mathbf{T}_d,\mathbf{L}_d|\mathbf{M})$. In practice, we find the top $k=100$ theories for each data set.

We represent $\mathbf{M}$ by adapting the Fragment Grammars formalism[43]. Concretely, $\mathbf{M}$ is a probabilistic context free grammar (PCFG) that stochastically generates phonological rules. More precisely, $\mathbf{M}$ generates the syntax tree of a program which implements a phonological rule. In the Fragment Grammars formalism, one first defines a *base grammar*, which is a context-free grammar. Our base grammar is a

context-free grammar over SPE rules (Supplementary Fig. 6). Learning the fragment grammar consists of adding new productions to this base grammar (the "fragments"), while also assigning probabilities to each production rule. Formally, each fragment is a subtree of a derivation of a tree generated from a non-terminal symbol in the base grammar; informally, each fragment is a template for a piece of a tree, and thus acts as a schema for a piece of a phonological rule. Learning a fragment grammar never changes the set of trees (i.e., programs and rules) that can be generated from the grammar. Instead, through a combination of estimating probabilities and defining new productions, it adjusts the probability of different trees. See Supplementary Fig. 6, which shows the symbolic structure of the learned fragment grammar.

This fragment grammar gives us a learned prior over single phonological rules. We define $P(\mathbf{T}, \mathbf{L}|\mathbf{M})$ by assuming that rules are generated independently and that $\mathbf{M}$ does not affect the prior probability of $\mathbf{L}$:

$$P(\mathbf{T}, \mathbf{L}|\mathbf{M}) = P(\mathbf{L}|\text{UG}) \prod_{r \in \mathbf{T}} P(r|\mathbf{M}) \tag{14}$$

Our prior over fragment grammars, $P(\mathbf{M})$, works by following the original work in this space[43] by assuming that fragments are generated sequentially, with new fragments generated from the current fragment grammar by stochastically sampling them from the current fragment grammar. This encourages shorter fragments, as well as reuse across fragments.

We depart from ref. 43 in our inference algorithm: while ref. 43 uses Markov Chain Monte Carlo methods to stochastically sample from the posterior over fragment grammars, we instead perform hill-climbing upon the objective in Eq. (13). Each round of hillclimbing proposes new fragments by antiunifying subtrees of phonological rules in $\{\mathbf{T}_d\}_{d=1}^{D}$, and re-estimates the continuous parameters of the resulting PCFG using the classic Inside–Outside algorithm[51]. When running Inside-Outside we place a symmetric Dirichlet prior over the continuous parameters of the PCFG with pseudocounts equal to 1.

### Reporting summary

Further information on research design is available in the Nature Research Reporting Summary linked to this article.

## Data availability

All data used and generated in this study have been publicly deposited in GitHub[52] at https://github.com/ellisk42/bpl_phonology along with the accompanying source code (DOI 10.5281/zenodo.6578329) under the GPLv3 license. Source data are provided with this paper.

## Code availability

The code used and developed in this study has been deposited in GitHub[52] at https://github.com/ellisk42/bpl_phonology (DOI 10.5281/zenodo.6578329) under the GPLv3 license.

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

## Acknowledgements

We thank Andrea Jang and Francie Freedman for their assistance in transcribing textbook phonology problems. We thank Edward Gibson, Roger Levy, and our anonymous reviewers for comments and suggestions that greatly improved the manuscript. Supported by grants from the Air Force Office of Scientific Research Grant no. FA9550-19-1-0269 (to J.B.T.), the National Science Foundation-funded Center for Brains, Minds, and Machines under Grant No. 1231216 (J.B.T.), the MIT-IBM Watson AI Lab (to J.B.T.), the Natural Sciences and Engineering Research Council of Canada Grant no. RGPIN-2018-06385 (to T.J.O.), the Fonds de Recherche du Québec Société et Culture Grant no. 254753 (to T.J.O.), the Canada CIFAR AI Chairs Program (to T.J.O.), the National Science Foundation under Grant No. 1918839 (to A.S.-L.), and an NSF graduate fellowship (to K.E.).

## Author contributions

K.E. and T.J.O. conceived of the project and model. K.E. implemented the model and ran simulations. K.E. and T.J.O. wrote the paper. T.J.O., A.A., J.B.T., and A.S.-L. advised the project.

## Competing interests

The authors declare no competing interests.
