## [Peer Review File · Nature Communications]

Synthesizing Theories of Human Language with Bayesian Program InductionREVIEWER COMMENTS

Reviewer #1 (Remarks to the Author):

Overview:

This paper proposes a new system which automatically discovers models of morpho-phonology. With this in mind, the paper uses a modified version of Bayesian Program Induction to learn language-specific lexicons, together with morphological and phonological composition rules. Distinguished from much work in morphological inflection, these learned models are fully interpretable, allowing the researcher to get plausible causal hypotheses for those inflection generation procedures.

Comments:

The paper is very interesting and advances the state of knowledge in the area. The idea of using Bayesian Program Induction is interesting and leads to fully interpretable programs, unlike the black box models used in most of the NLP literature. The strategy seems to work well, leading to high compliance in most languages.

A few points weren't clear to me while reading the paper though. If I understood equation 1 correctly, the rules are deterministic (since you use $1\{f=\text{Phonology}(\text{Morphology}(m))\}$ and all the uncertainty in the model comes from the prior generating them $p(T, L | UG)$. Nonetheless, in Figure 6, you compare the log probabilities of either consistent data instances or inconsistent ones under the training regime $P(\text{consistent} | \text{train})$ and $P(\text{inconsistent} | \text{train})$. How to get these probabilities was not clear to me from the paper. Is this system comparing the probabilities of the extra lexicon and rules which must be added to make the model compliant with these new instances? Or is the model capable of directly producing probabilities for new instances (f, m') on the fly?

It was also not clear to me how this model learns a stem's morphological paradigm or if these are simply directly memorized by the model's morphology function. It seems to me that a memorization methodology may be necessary for some languages. For instance, how would you know (without memorizing paradigms) solely from the word form that German word "Hund" gets an "-e" suffix in plural, while "Radio" gets an "-s". On the other hand, memorization may be inefficient for languages such as Spanish, where infinitive verb endings already encode the wanted paradigm ("-ar", "-er", "-ir"). This is also related to my previous question, of if these morphological functions are deterministically memorized or probabilistically applied.

From the paper, it was also not clear what a phonological problem from a linguistics textbook is. Is it a subset of a grammar which represents specific phenomena? Such as a Turkish vowel harmony (sub)grammar? I believe this paper will be of interest to NLP researchers who only have informal linguistics training, so clearing this up might be of interest to reach this broader audience. It would also be interesting to make clearer the complexity of the problems the proposed model can solve. Could I apply these methods to get the "full" morphological inflection grammar of a language using its entire unimorph data (<https://unimorph.github.io/>)? Or should I only use this method (for now at least) to investigate smaller phenomena? To be clear, I don't believe it to be a big problem if the second is true, but I couldn't fully grasp this from the paper in its current form.

If the models are indeed fully deterministic, some explanation of the potential weaknesses (and strengths) of such a choice could be interesting as well.

Can this model produce new inflections on the fly for previously unseen words?

Strengths:

- * Interesting paper, easy to read. Investigates a relevant and well-defined question.
- * Proposes a new fully interpretable model to solve it.
- * Evaluates the proposed model on a large and typologically diverse set of problems.
- * This model produces plausible causal generational hypothesis for the morpho-phonetic data.
- * Unlike many other models used for this task, this model seems *very* data efficient (it can learn from few examples).

Weaknesses:

- * Some experiments of the paper were not fully clear to me.
- ** If model is deterministic, how do authors get $P(\text{consistent} \mid \text{train})$ and $P(\text{inconsistent} \mid \text{train})$ in Figure 6?
- ** Does the morphological function directly memorize all stem paradigms directly?
- ** What exactly is the size/scope of the phonological problems the paper investigates?

Minor comments:

- * An extra plot in Figure 5 with a results summary could help the reader understand it faster. Maybe a boxplot for each model containing the span of results for the multiple analysed problems.
- * I think there may be an issue in Figure 2, on the Tibetan bottom derivations box. The derivation of ten+nine in it, shows gubdu, while the red box says durgu (I think the second should be the correct). Also, the derivation of 50 shows nabdu, but the red box shows gubdu (I believe the first should probably be the correct one here).

Reviewer #2 (Remarks to the Author):

This is just a wonderful paper. It tackles a problem of substantial relevance to multiple disciplines (linguistics, cognitive science, AI/ML), while breaking down the problem in an accessible way, with clearly interpretable results. I appreciate the work that must have gone into creating this manuscript. The interpretation of the findings strikes just the right middle ground between explaining the relevance of the results, without overstating the results (for example, a number of simplifying assumptions made in this study mean that the models learned in this study---while informative about language acquisition---ought not to be interpreted as models of human learning themselves). I also found the division of labor between the main text and SI just about perfect (the latter contains enough information to replicate and build on this work; the data and code are shared as well).

I don't often recommend outright acceptance---this is probably the third time in my career to do so---but I think this paper can be published with only minor changes.

My one request is that it should be stated more clearly (and earlier in the paper) that the problem sets considered here are a much simplified subset of the true problems faced by learners (and linguists). This is first acknowledged on p12, and reading the abstract/introduction, I first expected the paper to deal with much more complex "datasets" that correspond more closely to actual language learning problems. This is not to downplay the insights obtained by this work, but I suspect that other readers will be similarly misled (not intentionally, of course). In particular, the fact that solutions to these problems have been found by linguists (or that they even are *recognized* as problems---i.e., patterns) suggest that the hardest problems that learners and linguists might face are not represented among the 70 problem sets. This should be acknowledged early and transparently.

Minor suggestions:

- L127/Fig 3: The idea to incrementally expand / revise the grammar to accommodate counter-examples is sometimes termed 'active learning', and has been proposed specifically for language learning, too. A reference to would seem appropriate here. Perfors et al 2010 also comes to mind, though that study is more about incremental, rather than active learning.
- P. 6: I would think that it would at least be recommended to provide a transparent sense of the *scope* of these problems in the main text: e.g., how many words are in the actual problem set? (some of this can be gestimated from the figures but a more explicit statement would be preferable) And what's an estimate of the propotion of words in the language that are affected by the patterns present in the problem set?
- Fig 6: the training-test schemes require a bit more explanation: what does aab, etc. refer to (the syllables, I assume) and how does this related to Fig 6A?
- Fig 7: perhaps note that the y-axis range changes across the two top plots (or fix it).
- L217: "Critically this procedure is not given access to any ground-truth solutions to the training subset" Consider stating more explicitly that the learning is unsupervised?
- L227-8: Please explain "channel .. biases"

Typos:

- l112 "theorieseven"
- l120 spell out MCMC on first use?

Signed (as always, regardless of content)
Florian Jaeger

Reviewer #3 (Remarks to the Author):

As someone interested in unsupervised grammar induction, I found this paper interesting and thought-provoking, but also frustrating. As the authors note, their system is able to solve a much wider range of problems than previous work, which makes it worth understanding, yet after several hours (!) going back and forth between the main paper, supplement, and related work, I still feel there are significant gaps in what I could understand both about *how* the method works and *why* it works. Some of the explanations are incomplete, and in places the terminology is inconsistent, which also makes the explanation more difficult to understand. Several specific issues are listed below.

I also found the framing of the paper hard to understand -- it's not totally clear what the authors intend as the main contribution, or who the primary audience is, since there are several different contributions mentioned, making the message rather diffuse. Parts of the introduction (and later sections) make it sound as though the authors are claiming that their framework is a potential model for the linguist's theory induction, or the child's acquisition of language, while other parts imply that the goal is simply to engineer an AI system that can produce interpretable theories (in a way that may be loosely inspired by linguists or children). As the paper stands, the latter argument is far more convincing to me than the former.

However, regardless of the main claim, I'm concerned that some of the framing may be somewhat misleading (or at least unclear). In particular, the authors seem to argue that applying Bayesian Program Learning (and specifically, the Bayesian aspect of it) are key to the success of their system. To me, this implies a probabilistic system, and likely one that integrates over parameters or hypotheses in some way. It also typically implies a search/optimization algorithm that has some kind of guarantees. But in fact the system is doing approximate MAP inference using a heuristic search that (I think) doesn't guarantee anything (though see remaining confusions under my detailed questions

below). And although the system is technically probabilistic, the likelihood is degenerate (0/1) and the prior is an MDL prior, which can have a probabilistic interpretation but is essentially just a scoring function. I'm not saying there's anything wrong with these things in principle, but to a casual reader it would be easy to miss these points. (I suggest emphasizing the 0/1 likelihood by explaining eq 1 in words, not just as an equation. There are several precedents for this type of model with 0/1 likelihood from the word and morpheme segmentation literature, e.g. see work of Brent and/or Goldwater).

Moreover, since an exhaustive search would be left with no counterexamples (ie all problems solved), the search method must not be exhaustive. Therefore solutions are a product of *both* the model and the search heuristics, but this is rather deemphasized in the paper in favor of the "Bayesian" framing. (See more specific questions about this below.) This is particularly strange given that the idea of applying BPL to this problem is not new --- the authors themselves introduced it in a much simpler version earlier, but also citation [32] has a much more extensive treatment. So the main contributions to method here seem to be in the search and the meta-learning.

This review probably sounds rather negative, so I want to emphasize that there were a lot of interesting ideas and strong results in this paper. E.g. it's interesting to me that such a simple MDL prior seems to work so well. I also really liked the meta-learning idea and the learned grammar looks very impressive, although see my notes below regarding lack of explanation of these parts. So I would definitely like to see this work published eventually, but I feel that in the current form it is trying to over-claim to appeal to a broader audience while skimming over important aspects.

Some more specifics about terminology and explanations:

- Inconsistent terminology: on page 3, "theory", "grammar", and \mathbb{T} are all used to refer to the (phonological and morphological) rules only. However, elsewhere (e.g. Figs 2, 3, ...) "grammar" refers to "theory" \mathbb{T} plus lexicon \mathbb{L} . Still elsewhere, (e.g., Figs 4 and similar), the term "lexicon" isn't used at all, instead it's referred to as "stems". Fig 5 refers to a "rule learner", when in fact it learns both lexicon and rules (a particularly odd term considering that the evaluation is actually against lexica, *not* rules). p14 refers to \mathbb{T}_0 as "containing no phonological rules" -- without mentioning morphological rules.

- Due to these inconsistencies and lack of detail in S3, I'm left with considerable uncertainty as to how the search method actually proceeds. A concrete example going through the first one or two iterations of the procedure would help immensely. For example, suppose the first lexeme has two items: <MEET (INF), [mit]>, <MEET (3SG), [mits]>. The "theory" \mathbb{T} starts out empty, so it isn't even obvious to me why these would serve as counterexamples to that \mathbb{T} . Why can't these items simply be added to the lexicon as is? ie memorizing all of the items provides an explanation with no counterexamples.

Perhaps there is an assumption (not stated anywhere, I think) that we know how many morpheme-meanings are in each item, what they are, and that each must have its own form listed in the lexicon (ie we *must* split up the surface forms somehow). Then I suppose we'd need a lexicon with at least 3 meanings (MEET, INF, 3SG), and to get the observed words we would need need at least two rules (?) to combine them. From just these examples, we might decide that /s/ corresponds to (3SG), but then we would eventually see counterexamples such as <NEED (3SG), [needz]>.

- It is mentioned that the CEGIS method is exhaustive, and that the current search method is needed in order to be more efficient. The way it is described, it sounds like it should still find *some* solution, even if not the best one (p8 Supp: "we progressively increment the maximum edit distance until Sketch discovers a satisfying solution".) Yet this is clearly not the case, since if it were, the evaluation would be 100% for all problems attempted. I assume the limiting factor is still time, but how is this

limit operationalized? (For both your system and for CEGIS, as that seems critical to understanding the comparison in the evaluation.)

If the system fails to find a solution, is it possible that a solution could be found if different mini-batches or data ordering were used, or only by increasing the allowed edit distance from the current solution? I gather that different solutions might be found (?) by changing ordering or initialization, otherwise I'm not sure how the multiple different solutions in the artificial grammar section were found.

- The meta-learning model is not fully explained anywhere. In particular, how are $P(M)$ and $P(T_d, L_d | M)$ defined? I think the former comes from previous work on fragment grammars (though it's not safe to assume most readers would know that), and the latter seems to be the critical piece, since it replaces the MDL prior of the UG and therefore seems to be responsible for all of the improvement achieved by using the meta-model. Fragment grammars define a probability distribution over rule fragments --- how do these relate to the purely symbolic set of rules shown in S5? Is that set of rules the ones with non-negligible probability under the meta-grammar? Are all other rules still possible? If so, how does using the meta-grammar lead to more solutions being found? E.g., is it allowing each iteration of the algorithm to run faster (thus, more iterations can run within the time you have available), or allowing better rules to be picked early on so that the search doesn't go off into a bad part of the space where nearby theories can't solve the problem? (My confusion here is related to the previous point, ie. I don't understand what the limits are that cause certain problems to fail.)

- More terminology: on p7 (and Fig 7), you say that the solutions found are a trade-off between "parsimony (size of grammar) and fit to data (size of lexicon)". But your lexicon is not data, it is latent. *Both* of these terms are part of the parsimony prior, and as noted elsewhere, all of your solutions have a perfect fit to the data. So describing the trade-off in this way is misleading.

- When computing the edit distance between two theories, is it assumed that any rule has a distance of 1 from any other rule? I.e., there is no difference between making a small change to a rule, versus introducing a completely different rule?

- You mention some ablated models in Fig 5, but I don't see an explanation anywhere of what the "acoustic features" are as opposed to the phonological articulatory features; or what "ablating phonetic features and Kleene star" would look like. These should be explained in the Supplement.

- The comparison to [32] points out that [32] fails to solve 76% of the problems. But [32] claims that their system is 2 orders of magnitude faster than yours. There is no "correct" position on the speed/accuracy trade-off, but I would like to see a clear statement of how long your experiments take to run, and an acknowledgment that [32] is much faster. You mentioned 40 CPUs, but no length of time.

- Finally, if the goal of the system is to learn interpretable theories, then I don't understand why the quantitative evaluation focuses only on whether the stems match the gold standard. You say that there are "many rules" that could potentially explain the data -- but typically linguists would consider only one or two possibilities to be reasonable. Why don't you evaluate your learned rules against the gold standard rules? (The main examples provided are very useful, but since you've also listed the system output for every problem, why not add the gold standard for each of those, and compute some summary statistics?)

Various other minor issues:

Why is [32] referred to as "PhonoSynth"? Their paper calls the system "SyPhon".

The first sentence of S1 says "brackets" twice when I believe the second one should say "slashes". That said, the remainder of the section seems rather inconsistent about using brackets vs slashes correctly. (I think there are several cases of slashes that should be brackets, or else need the 't' replaced with a 'd'.)

I'm not sure 'SPE' is ever spelled out, or 'SPE-style' rules defined (or SPE cited).

Similarly, I don't know if it is safe to assume all readers will understand what 'Pareto fronts' are (p7).

Fig S6: I infer from the italicised note that the subscript 0 (as in FeatureMatrix_0) is likely your notation for Kleene star, but that could be stated much more explicitly.

Reviewer #4 (Remarks to the Author):

This submission details the use of program synthesis to solve textbook morphophonology problems. The authors report several results: extensionally correct grammars (as far as the problems, which are often quite brief, are concerned) can be learned by this method, a learned meta-theory/cross-linguistic inductive bias improves performance, and this is useful for modeling child performance in artificial grammar learning experiments as well.

My comments are largely structured around what, normatively, I think the paper should tell the reader. Some of these things are purely rhetorical, others are comparisons or observations I would recommend to the authors before acceptance, others are formal issues whose solutions are out of scope of this paper but will determine the overall influence of this submission.

Below, I many times suggest that the authors provide more information. Realize that there may be length limitations for this format, I would suggest the authors omit discussion of the artificial language learning or relegate it to an appendix. This is an interesting result but feels somewhat far afield; only a small percent of the overall readers will be concerned with the validity of such approaches.

Before I begin in more detail, the review form asks me to address a number of specific questions, which I'll address now.

Is the work convincing? I would tentatively say yes, though with some strong caveats. First, though there are not really any meaningful comparisons to other work. Figure 5 includes a partial comparison to something called Phonosynth. This method is not described in any detail, the comparison is partial (only some languages were fed to Phonosynth perhaps the authors have used results from that paper rather than replicated the results themselves?), Phonosynth matches or outperforms (depending on condition and language) the proposed methods, and finally no overall accounting is given for the comparison (i.e., which is better?). Secondly, nothing is said about the time and space complexity of the Sketch model used to find solutions, nor about its hardware demands or the amount of wall-clock time it requires. Does it take a cluster running for weeks to solve these problems, or does the whole thing run in seconds? Finally, the authors provide no guarantees, approximation bounds, or evidence that the theories are even decidable. My limited understanding is that in general SMT theories are undecidable, a very serious problem, but the authors may be able to show that they are only considering a decidable subset and perhaps discuss the size of this subset as a function of the properties of the input. Even then, I don't know in what sense the overall approach can be called "exhaustive but efficient" but also a naive approach is said to have a "steep computational cost": that can only be concluded if the theories are all decidable, the time and space complexity of solution is some reasonably slow-growing polynomial function. Do counter-example synthesis and test-driven

synthesis accomplish this? I feel strongly that the paper is incomplete without discussion on these notes. One obvious comparison to be made is against the so-called input strictly local functions studied by Chandee and colleagues, which seem to cover a wide variety of morphophonological functions and which have much stronger learning guarantees, though the representations learned arguably less closely resemble those used by linguists.

Do you think this paper will influence thinking in the field? I would tentatively say yes once again. This is by no means the first work using program synthesis approaches to morphophonology, but this would be the first and probably most high-profile journal article detailing the approach that I'm aware of, unless [19] is published first. And clearly there's more work to be done.

Is the paper reproducible? I would suppose so. I briefly looked at the included code assets, which seem to be reasonably well documented. The authors use the Python 2.7 programming language, which is now end-of-life as of January 2020 and will not be widely available in a few years, so a quick port to Python 3.x would make this more future-proof, but is not something I would require the authors to do.

Some major observations, in the order in which they appear in the paper

I think the authors initially oversell the conclusions. This is described in the abstract as a "framework for algorithmically synthesizing models of human language, focusing on morpho-phonology". While this general framework may in fact be a general purpose for all kinds of subareas of human language, I don't see any reason to suppose it is. This possibility could be discussed in more detail in the concluding remarks but it doesn't seem justified by the current paper.

I understand that in the introduction the authors are trying to work with the child-as-scientist metaphor but I find something strange about notions like "language-specific theories". For me, as a working linguist, my account of a single language is an analysis or grammar, and referring to a morphophonological analysis or grammar as a theory is jarring. To add to the complexity, SMT problems are often called "theories" as well. To resolve this terminological morass, I'd suggest the authors use "analysis" or "grammar" to refer to language specific theories and "problem" to refer to what's submitted to the SMT solver, though the authors are welcome to try something different.

The authors' description of UG in the second paragraph is simply not consistent with my understanding of the term. The framework of representations and processes for morphophonology, assumed to be common to all human languages is certainly part of UG, defining a hypothesis space for possible rules, but it's not the sum total of UG, which also includes the induction algorithm and its inductive biases, for instance. Furthermore, children bring more than just UG (and the primary linguistic data) to language acquisition: they also bring domain-general limitations and skills: constraints on audition, attention, working memory, etc. These are often thought of as constraints but they might also be thought of as resources for learning, resources which happen to be finite. Chomsky (2005), in an influential paper, refers to these collectively as "third factors".

"Second, children easily acquire language from quantities of data that are modest by the standards of modern artificial intelligence." I think this is true, but should be backed up with example or citation. How much data do children take in? How much is non-modest for modern AI research?

The identification of /z/ as the regular past is not universally accepted by linguists (though it is probably the best analysis); the authors may want to mention that. Some linguists also transcribe the epenthesis vowel as \bar{i} . The rules are also not properly stated: the prose statement here suggests that all unvoiced consonants select /s/, but then revises that. I would state the rules as surface-true form: "/s/ is selected by unvoiced consonants except ..."

I see why the authors say that the theories are "human understandable" but I think "theories of the

causal relationship between form and meaning" and "human-like AI" said on page 2 is maybe stronger than many linguists would agree. There is of course a long-standing question the degree to which our theories are also theories of production and perception, and whether production and perception grammars are equivalent. The authors presume a very strong equivalence that few linguists would endorse. This assumption is also repeated on 86 when the goal of acquisition is stated as a grammar which "maps back and forth between form...and meaning".

There is a somewhat standard way to write the form/meaning tuples used to discuss the problem on page 3. Roots are written in small caps, with a surd; features are given using Leipzig-style glosses.

Figure 1D is contentless; the point is better illustrated in prose.

It might be helpful, on page 3, to tell the reader what is lost by restricting to concatenative processes. What are types of non-concatenative processes, where can readers read about them, and how common are there? Are there processes that have concatenative and non-concatenative analyses? Etc.

It seems from equation (1) that this is restricted to exactly one prefix and suffix, is that correct? What do we lose by this assumption? This seems like an enormous restriction on the grammar space. How would the authors work around it? This absolutely needs to be discussed.

The discussion of features is somewhat confusing in that it's not made clear that a specification like +nasal denotes a set of phonemes which have that property. Nor is it made clear that a specification is always interpreted conjunctively rather than disjunctively. Or is it the case the authors allow disjunctive feature specifications in their SMT problems?

From where do we get the feature specifications? I mean this question both epistemologically (where do we suppose children get them? are they part of UG?), and in the literal sense of where do the authors get their feature vectors for segments?

What happens if we just get rid of features and only work with segments? What do we lose? Is it possible to run the experiments again with a "dummy" (one-hot) "features" and measure how performance is impacted?

Rather than saying (line 109) that the rules have the power of FSTs I would say instead that the rules, and the morphology + rules, correspond to 2-way rational functions, which in turn correspond to finite-state transducers.

In standard jargon "morphosyntactic category" means {noun, verb, ...}. I would suggest the authors use some new jargon for the notion here.

What does it mean to treat Sketch as an oracle? From context it seems to me that the authors mean that we don't care about how long it takes or what its properties are, but I think the readers should care about those issues, very much.

What does counter-example guided synthesis and test-driven synthesis mean? Give an intuitive explanation. I wasn't able to reason exactly how they work from figure 3.

What does it mean to "ablate...Kleene star"? I know what Kleene star is (though readers won't necessarily, so it should be defined) but I don't know what it would mean to ablate it.

How do [27-30] work? The competing approaches deserve more than a few lines. They're said to scale poorly but as I have said earlier, we don't really have a characterization of the scaling properties of the proposed method either. Similarly, the use of the word "opaque" is a poor choice of words here since that's a term of art in morphophonology and I don't think the authors mean that narrower

sense.

The paragraph on lines 170-192 feels like it is out of the scope of this work. I also find the conclusion unconvincing. Why does the ability of a model to handle AGL and real data entail something about the cognitive resources tapped into AGL exactly?

Lezgian is argued to in fact voice final obstruents. Yu (2004) provides phonetic evidence and discusses how this rule came about. So this is an unfortunate example on the paragraph beginning 193-204.

I found the physics example incomprehensible. Is this going to be useful to the target reader? It seems that it would be fine to say that this kind of heuristic reasoning is common among scientists and technicians.

There is reason to think that the Polish o-raising rule is not a phonological rule at all; see Buckley 2001 and subsequent work. I just mention this in case the authors want to use a less controversial example in figure 7.

Where does "phonological commonsense (sic)" reside? Is it something working linguists have? Do kids have it too? I think the authors should take a stance on this.

The authors should explain what a Pareto front is. I suspect the vast majority of readers will not be familiar with this concept.

I don't think it's fair to call "distributional semantics" (a misnomer) a model of semantics. It is entirely unclear how one goes from word2vec or BERT to inferential semantics.

There is no discussion about the impact of this work for field linguistics/language documentation.

There is no discussion about the impact of this work for deciding between extensionally-equivalent analyses of the same language. The proper analysis of the Catalan phenomenon mentioned herein has been debated for many years; does the BPL approach, or the use of Pareto fronts, help resolve the question in any way? See Hale & Reiss 2008:271f. for discussion of this particular example.

More specifically, what is this model's take on absolute neutralization? Opacity? Can it tell us whether Russian yers are epenthized or deleted?

There could be more discussion about what else could be done to deploy this class of model for morpho-phonology? The careless reader might take away from this that the problem is solved, but that does not seem to be the case.

What if any hyperparameters are important for the use of Sketch? For instance the authors mention the use of minibatches: how big are they and does it matter?

A few finicky details:

* The linguistic standard is to write (Roman) words used as examples (like "horse" on page 2) in italics, and give the gloss of all non-English words in single quotes after. The examples are instead double-quoted; the Serbo-Croat words are not glossed in the body but they are given beforehand (instead of after) in single (instead of double) quotes.

* The authors should put their "minus" characters (e.g., in feature bundles) in math mode. As is they look like short hyphens, not minus signs.

* The term "ablating", used throughout, should be defined on first use. It has a common sense (to burn away) that doesn't immediately match the one used in machine learning...

* "Bayes's": at the risk of being prescriptive, is this not normally written Bayes'?

- * Footnote 1 is missing sentential punctuation.
- * "distill out": the particle "out" seems redundant to me.
- * line 112: "theorieseven"
- * "context-sensitive rewrite": I would suggest a more standard jargon "context-dependent rewrite", so to avoid the confusion with the family of context-sensitive languages.
- * The Turkish example on lines 147f. is discussed too briefly for the reader to easily grasp it. Perhaps segmentation would help. There is also confusion about the use of square vs. angled brackets here. The parenthetical on line 152 should be foregrounded at the start of the paragraph, it's a very important point.
- * "commonsense": "common sense", I think.
- * Stampe's dissertation is dated 1969; a printed version (under the name "A dissertation in natural phonology", published by Garland) appeared in 1973.
- * The language described as "Yawelmani" is probably more properly called "Yowlumne". (I sometimes cite this as something like "Yowlumne, formerly known as Yawelmani".) This is discussed in Weigel's 2005 Berkeley dissertation.

We are grateful for the thoughtful input of all these reviewers. We have performed major revisions to the manuscript to take the reviewer's feedback into account (with changes highlighted in blue). Below, we show the comments of each reviewer in italics and our responses highlighted in blue.

Reviewer #1:

Overview:

This paper proposes a new system which automatically discovers models of morpho-phonology. With this in mind, the paper uses a modified version of Bayesian Program Induction to learn language-specific lexicons, together with morphological and phonological composition rules. Distinguished from much work in morphological inflection, these learned models are fully interpretable, allowing the researcher to get plausible causal hypotheses for those inflection generation procedures.

Comments:

The paper is very interesting and advances the state of knowledge in the area. The idea of using Bayesian Program Induction is interesting and leads to fully interpretable programs, unlike the black box models used in most of the NLP literature. The strategy seems to work well, leading to high compliance in most languages.

A few points weren't clear to me while reading the paper though. If I understood equation 1 correctly, the rules are deterministic (since you use $1\{f=Phonology(Morphology(m))\}$) and all the uncertainty in the model comes from the prior generating them $p(T, L | UG)$. Nonetheless, in Figure 6, you compare the log probabilities of either consistent data instances or inconsistent ones under the training regime $P(\text{consistent} | \text{train})$ and $P(\text{inconsistent} | \text{train})$. How to get these probabilities was not clear to me from the paper. Is this system comparing the probabilities of the extra lexicon and rules which must be added to make the model compliant with these new instances? Or is the model capable of directly producing probabilities for new instances (f', m') on the fly?

Indeed, our initial manuscript presented a formulation with a 0/1 likelihood, which leaves unclear how the model produces probabilistic judgments for new forms. In fact, our formulation is equivalent to a probabilistic model which places a probability distribution over paradigm rows (i.e., tuples of inflections sharing the same stem). Our new draft makes this connection mathematically explicit:

The input data \mathbf{X} typically come from a ‘paradigm’ matrix, whose rows range over inflections and
 whose columns range over stems. In this setting, the theory acts as a probabilistic generative model which
 places a probability distribution over paradigm rows, $P(\mathbf{X}_\sigma)$, where σ is the meaning of a stem, and \mathbf{X}_σ
 is the corresponding row of the paradigm. The likelihood $P(\mathbf{X}_\sigma)$ depends only on rules and affixes. Fig. 2
 illustrates this framing by partitioning the lexicon into stems and affixes. This framing is mathematically
 connected to our MAP setup as:

$$\begin{aligned}
 & \underbrace{P(\mathbb{T}|\mathcal{UG}) \left(\prod_i P(\mathbb{L}(i, \mathbf{pfx}))P(\mathbb{L}(i, \mathbf{sfx})) \right)}_{\text{prior over rules and affixes}} \underbrace{\prod_\sigma \sum_{z_\sigma} P(z_\sigma) \prod_i \mathbb{1} \left[\mathbf{X}_{\sigma,i} = \text{Phonology}(\mathbb{L}(i, \mathbf{pfx}) \cdot z_\sigma \cdot \mathbb{L}(i, \mathbf{sfx})) \right]}_{\text{probabilistic likelihood model: } P(\mathbf{X}_\sigma)} \\
 & \geq P(\mathbb{T}|\mathcal{UG}) \left(\prod_i P(\mathbb{L}(i, \mathbf{pfx}))P(\mathbb{L}(i, \mathbf{sfx})) \right) \prod_\sigma \max_{z_\sigma} P(z_\sigma) \prod_i \mathbb{1} \left[\mathbf{X}_{\sigma,i} = \text{Phonology}(\mathbb{L}(i, \mathbf{pfx}) \cdot z_\sigma \cdot \mathbb{L}(i, \mathbf{sfx})) \right] \\
 & = \text{the MAP objective for inference in Eq. 1} \tag{2}
 \end{aligned}$$

where z_σ is the underlying form of stem σ , namely $\mathbb{L}(\sigma, \mathbf{stem})$. Thus, the MAP objective (Eq. 1) serves as
 a lower bound on the full Bayesian objective above. This bound will be tightest when each paradigm row
 admits very few possible stems. Typically only one stem per row is consistent with the rules and affixes,
 which justifies our MAP objective.

It was also not clear to me how this model learns a stem's morphological paradigm or if these are simply directly memorized by the model's morphology function. It seems to me that a memorization methodology may be necessary for some languages. For instance, how would you know (without memorizing paradigms) solely from the word form that German word "Hund" gets an "-e" suffix in plural, while "Radio" gets an "-s". On the other hand, memorization may be inefficient for languages such as Spanish, where infinitive verb endings already encode the wanted paradigm ("-ar", "-er", "-ir"). This is also related to my previous question, of if these morphological functions are deterministically memorized or probabilistically applied. Our model indeed does effectively work by directly memorizing the morphology function. For languages like those you mentioned, where there are different inflectional classes (such as German plurals), we followed the standard structure of textbook problems, which exposed the underlying inflectional class in such cases. Although discovering inflectional classes is an important part of language acquisition, our primary goal was to build a testbed for linguistic theory-induction, for which we used standard linguistics problems that make certain simplifying assumptions, this being one of them. In principle, an extension of our model would be able to learn this as well, by introducing an extra latent variable for each stem corresponding to its inflectional class. We leave such extensions to future work, and hope to pursue them. We have updated the manuscript (line 102-104) to be clearer about this: "We handle inflectional classes (e.g. declensions) by exposing this information in the observed meanings, which follows the standard textbook problem structure" with a footnote saying "This simplifies the full problem faced by children learning language. In principle, our framing could be extended to learn these classes by introducing an extra latent variable for each stem corresponding to its inflectional class."

From the paper, it was also not clear what a phonological problem from a linguistics textbook is. Is it a subset of a grammar which represents specific phenomena? Such as a Turkish vowel harmony (sub)grammar? I believe this paper will be of interest to NLP researchers who only have informal linguistics training, so clearing this up might be of interest to reach this broader audience. It would also be interesting to make clearer the complexity of the problems the proposed model can solve. Could I apply these methods to get the "full" morphological inflection grammar of a language using its entire unimorph data (<https://unimorph.github.io/>)? Or should I only use this method (for now at least) to investigate smaller phenomena? To be clear, I don't believe it to be a big problem if the second is true, but I couldn't fully grasp this from the paper in its current form.

Phonological problems typically present clean data that isolates a handful of linguistically interesting phenomena, such as Turkish vowel harmony. Thus we do not focus on attaining a "full" morphological inflection grammar. The new manuscript clarifies this by saying in the introduction (line 77-79): "These data sets come from phonology textbooks: they have high linguistic diversity, but are much simpler than full language learning, with tens to hundreds of words at most, and typically isolate just a handful of grammatical phenomena."

If the models are indeed fully deterministic, some explanation of the potential weaknesses (and strengths) of such a choice could be interesting as well. Can this model produce new inflections on the fly for previously unseen words?

Although the model's learned morphology and phonology is deterministic, it can produce new inflections on-the-fly by inferring new stems. As explained in our first point-by-point response the new manuscript clarifies on lines 124-136 how probabilistic judgments are attained, and we use this capability in the artificial grammar learning simulations.

Strengths:

- * Interesting paper, easy to read. Investigates a relevant and well-defined question.
- * Proposes a new fully interpretable model to solve it.
- * Evaluates the proposed model on a large and typologically diverse set of problems.
- * This model produces plausible causal generational hypothesis for the morpho-phonetic data.
- * Unlike many other models used for this task, this model seems *very* data efficient (it can learn from few examples).

Weaknesses:

- * Some experiments of the paper were not fully clear to me.
- ** If model is deterministic, how do authors get $P(\text{consistent} | \text{train})$ and $P(\text{inconsistent} | \text{train})$ in Figure 6?

In addition to the newly clarified Bayesian formulation, we have added a new section to Methods specifically to describe the calculations needed for Figure 6. This explanation

relies on the new Equation 2 (which gives the more clearly Bayesian formulation; lines 368-383):

Few-shot artificial grammar learning
We present our system with a *training set* of words from a target language, such as the ABA language
(e.g. ‘wofewo’, ‘mikami’, ...). Then we compute the likelihood assigned to held-out words either *consistent*
with the target grammar (e.g. following the ABA pattern) or *inconsistent* with the target grammar (e.g.
following the ABB pattern, such as ‘wofefe’, ‘mikaka’, ...). The probability assigned to a held-out test word,
conditioned on the training set, is approximated by marginalizing over the Pareto-optimal grammars for the
train set, rather than marginalizing over all possible grammars:

$$374 \quad P(\text{test}|\text{train}) = \sum_{\mathbb{T}, \mathbb{L}} P(\mathbb{T}, \mathbb{L}|\text{train})P(\text{test}|\mathbb{T}, \mathbb{L}) \quad (8)$$

$$375 \quad \approx \sum_{(\mathbb{T}, \mathbb{L}) \in \text{ParetoFrontier}(\text{train})} P(\mathbb{T}, \mathbb{L}|\text{train})P(\text{test}|\mathbb{T}, \mathbb{L}) \quad (9)$$

which relies on the fact that Sketch has out-of-the-box support for finding Pareto-optimal solutions to
multiobjective optimization problems [51].
We approximate the likelihood $P(\text{test}|\mathbb{T}, \mathbb{L})$ by finding the latent stem z_σ which maximizes the probability
of the test word. This is equivalent to finding the shortest stem which will generate the test word, given the
affixes in \mathbb{L} and the rules in \mathbb{T} . Formally:

$$381 \quad P(\text{test}|\mathbb{T}, \mathbb{L}) \approx \max_{z_\sigma} P(z_\sigma) \mathbf{1}[\text{test} = \text{Phonology}(\mathbb{L}(\mathbf{pfx}) \cdot z_\sigma \cdot \mathbb{L}(\mathbf{sfx}))] \quad (10)$$

where we have reused the notation of Eq. 2 while dropping the dependence of the prefix/suffix on the
inflection, because there is only one inflection for these data sets.

*** Does the morphological function directly memorize all stem paradigms directly?*

*** What exactly is the size/scope of the phonological problems the paper investigates?*

These were addressed above when this reviewer originally raised these issues (“Our model indeed does effectively work by directly memorizing the morphology function”... and “Phonological problems typically present clean data that isolates a handful of linguistically interesting phenomena”...)

Minor comments:

** An extra plot in Figure 5 with a results summary could help the reader understand it faster. Maybe a boxplot for each model containing the span of results for the multiple analysed problems.*

Our new Figure 5 contains a panel with summary results. These summary results show both the average performance of each model at convergence, as well as the rate of convergence as measured by time spent doing search:

— ours (full)
 — ours (CEGIS)
 — ours (simple features)
 — -representation
 — SyPhon (2019)

I think there may be an issue in Figure 2, on the Tibetan bottom derivations box. The derivation of ten+nine in it, shows gubdu, while the red box says durgu (I think the second should be the correct). Also, the derivation of 50 shows nabdu, but the red box shows gubdu (I believe the first should probably be the correct one here).

Thank you for the catch! We've fixed the figure:

Review #2:

This is just a wonderful paper. It tackles a problem of substantial relevance to multiple disciplines (linguistics, cognitive science, AI/ML), while breaking down the problem in an accessible way, with clearly interpretable results. I appreciate the work that must have gone into creating this manuscript. The interpretation of the findings strikes just the right middle ground between explaining the relevance of the results, without overstating the results (for example, a number of simplifying assumptions made in this study mean that the models learned in this study---while informative about language acquisition---ought not to be interpreted as models of human learning themselves). I also found the division of labor between the main text and SI just about perfect (the latter contains enough information to replicate and build on this work; the data and code are shared as well).

I don't often recommend outright acceptance---this is probably the third time in my career to do so---but I think this paper can be published with only minor changes.

*My one request is that it should be stated more clearly (and earlier in the paper) that the problem sets considered here are a much simplified subset of the true problems faced by learners (and linguists). This is first acknowledged on p12, and reading the abstract/introduction, I first expected the paper to deal with much more complex "datasets" that correspond more closely to actual language learning problems. This is not to downplay the insights obtained by this work, but I suspect that other readers will be similarly misled (not intentionally, of course). In particular, the fact that solutions to these problems have been found by linguists (or that they even are *recognized* as problems---i.e., patterns) suggest that the hardest problems that learners and linguists might face are not represented among the 70 problem sets. This should be acknowledged early and transparently.*

Absolutely it is important to acknowledge this early and transparently, especially for readers who may not be familiar with the scope of phonology textbooks. We have now said the following in the introduction: "These data sets come from phonology textbooks: they have high linguistic diversity, but are much simpler than full language learning, with tens to hundreds of words at most, and typically isolate a handful of grammatical phenomena at most." (lines 76-79)

Minor suggestions:

• L127/Fig 3: The idea to incrementally expand / revise the grammar to accommodate counter-examples is sometimes termed 'active learning', and has been proposed specifically for language learning, too. A reference to would seem appropriate here. Perfors et al 2010 also comes to mind, though that study is more about incremental, rather than active learning. Thank you for pointing out the possible connection to active learning. Within the contexts that we are familiar with, "active learning" usually requires an oracle that can provide labels

as needed, and the “active learner” in this context has the ability to query this oracle. For our setting, this oracle does not exist: We assume that we have all the data we will ever see, labelled all at once. We feel that the learning algorithms that we build most directly upon are counter-example-guided-inductive-synthesis and test-driven-synthesis, which the new draft now clarifies in lines 177-181: “we find a counterexample to our current theory, and then use the solver to exhaustively explore the space of all small modifications to the theory which can accommodate this counterexample. This combines ideas from counter-example guided inductive synthesis (which alternates synthesis with a verifier that feeds new counterexamples to the synthesizer) with test-driven synthesis (which synthesizes new conditional branches for each such counterexample)”. That being said, we are happy to draw connections to analogous works in the language learning literature, but are not aware of any that use the same kind of guided synthesis (although there are many incremental learning algorithms) , and as you mention, Perfors et al 2010 does not closely follow the structure of our incremental algorithm.

• P. 6: I would think that it would at least be recommended to provide a transparent sense of the *scope* of these problems in the main text: e.g., how many words are in the actual problem set? (some of this can be gestimated from the figures but a more explicit statement would be preferable)

Thanks for the suggestion. As explained in the earlier point-by-point response, we’ve added this to the introduction: “These data sets come from phonology textbooks: they have high linguistic diversity, but are much simpler than full language learning, with tens to hundreds of words at most, and typically isolate just a handful of grammatical phenomena.” As an extra guide, our qualitative illustrations contain the data set size (Figure 8):

And what's an estimate of the proportion of words in the language that are affected by the patterns present in the problem set?

Most words in the data sets are affected by the processes illustrated in the problem. This is a consequence of using textbook problems, which are designed to isolate and illustrate specific processes.

• Fig 6: the training-test schemes require a bit more explanation: what does aab, etc. refer to (the syllables, I assume) and how does this related to Fig 6A?

Thank you for pointing this out. We have clarified this notation in the caption to Figure 6, because that figure is the only place where this notation is used. In particular, we now say: "Grammar names ABB/ABA/AAx/AxA refer to syllable structure: A/B are variable syllables, and 'x' is a constant syllable. For example, ABB words have three syllables, with the last two syllables being identical." To avoid further confusion, we tweaked Figure 6 so that panels A/B use the same naming convention (namely, all-caps syllable patterns):

A

grammar	example input to learner	inferred grammar	natural language analogues
ABB Marcus et al. 1999.	wofefe lovivi fimumu	$\emptyset \rightarrow \sigma_i / \sigma_i_ \#$	reduplication, e.g., Tagalog
ABA Marcus et al. 1999.	wofewo lovilo fimufi	$\emptyset \rightarrow \sigma_i / \sigma_i \sigma_ \#$	reduplication
AAx Gerken 2006.	wowoka loloka fifika	stem+ka $\emptyset \rightarrow \sigma_i / \#_ \sigma_i$	reduplication concatenative morphology Mandarin (Fig. 6, lower right)
AxA Gerken 2006.	wokawo lokalo fikafi	ka+stem $\emptyset \rightarrow \sigma_i / \#_ \sigma \sigma_i$	infixing, e.g. Arabic reduplication
Pig Latin	pig → igpe latin → atile æsk → æske	$\emptyset \rightarrow C_i / \# C_i []_0_ \#$ $\emptyset \rightarrow e / _ \#$ $C \rightarrow \emptyset / \#_$	child language games

• Fig 7: perhaps note that the y-axis range changes across the two top plots (or fix it).
We have added the following to the caption: “N.B.: Because the grammars and lexica vary in size across panels, the X & Y axes have different scales in each panel.”

• L217: “Critically this procedure is not given access to any ground-truth solutions to the training subset” Consider stating more explicitly that the learning is unsupervised?
We have changed this sentence to: “Critically this *unsupervised* procedure is not given access to any ground-truth solutions to the training subset” (lines 287-288)

• L227-8: Please explain “channel .. biases”

Due to changes in this paragraph motivated by Reviewer 4, we no longer discuss channel/analytic biases. However, we are happy to clarify them here anyway: *Channel biases* are inductive biases which stem from communication considerations over a noisy channel, and thus, should they be present within human learners, do not require explicit instantiation, at least in principle, because they should emerge naturally from the iterative, intergenerational evolution of language. *Analytic biases*, in contrast, are inductive biases which stem from either domain general cognitive limitations, or idiosyncratic, language-specific biases not explainable in terms of general communication principles. The new version of this paragraph says much more simply, in regards to the role of UG, typological tendencies, and our hierarchical Bayesian fragment grammar, saying (lines 296-300) that “[r]ather than capture a learning process, our meta-theorizing is analogous to a discovery process that distills knowledge of typological tendencies, thereby aiding future model synthesis. **However, we believe that children possess implicit knowledge of these and other tendencies, which contributes to their skill as language learners.** Similarly, we believe the linguist’s skill in analysis draws on an explicit understanding of these and other cross-linguistic trends” (where in bold we show the new text which replaces the previous mention of channel/analytic biases). We believe the previous mention was unnecessary and that this revision makes the paper more accessible.

Typos:

• l112 “theorieseven”

• l120 spell out MCMC on first use?

Fixed!

Review #3:

*As someone interested in unsupervised grammar induction, I found this paper interesting and thought-provoking, but also frustrating. As the authors note, their system is able to solve a much wider range of problems than previous work, which makes it worth understanding, yet after several hours (!) going back and forth between the main paper, supplement, and related work, I still feel there are significant gaps in what I could understand both about *how* the*

*method works and *why* it works. Some of the explanations are incomplete, and in places the terminology is inconsistent, which also makes the explanation more difficult to understand. Several specific issues are listed below.*

I also found the framing of the paper hard to understand -- it's not totally clear what the authors intend as the main contribution, or who the primary audience is, since there are several different contributions mentioned, making the message rather diffuse. Parts of the introduction (and later sections) make it sound as though the authors are claiming that their framework is a potential model for the linguist's theory induction, or the child's acquisition of language, while other parts imply that the goal is simply to engineer an AI system that can produce interpretable theories (in a way that may be loosely inspired by linguists or children). As the paper stands, the latter argument is far more convincing to me than the former.

The original submission gave several interpretations of the work: as a model of linguistic analysis; as a model of language learning; and as a model of theory induction. Like reviewer #3, we felt in the initial draft--and continue to feel--that this last interpretation is the strongest, and so have rephrased key parts of the paper to foreground this angle on the work. In particular, we've made the following changes:

- Changed abstract text from: "These results point to more powerful models of language acquisition and suggest routes to machine-enabled discovery of interpretable models in linguistics and other scientific domains" to "These results suggest routes to more powerful machine-enabled discovery of interpretable models in linguistics and other scientific domains." (lines 7-8)
- Changed abstract text to deemphasize more tentative connections to language acquisition. We previously said that the model "captures language learning dynamics, acquiring new morphophonological rules from just one or a few examples, as infants do", and now say that the model "captures few-shot learning dynamics, acquiring new morphophonological rules from just one or a few examples." (lines 14-15)
- Changed the second introductory paragraph's lead sentence from: "In this paper, we study the problem of theory discovery in the domain of human language" to "In this paper, we study the problem of AI-driven theory discovery, using human language as a testbed." (line 25)
- Changed the introduction's strong claim that we "argue that our accounts can shed light on the child's acquisition of language" to the softer claim that "we primarily focus on the linguist's construction of language-specific theories... and synthesis of abstract cross-language meta-theories, but we also propose connections to child language acquisition." (line 27)
- Changed the discussion to lead with: "Our high-level goal was to engineer methods for synthesizing interpretable theories, using morphophonology as a testbed and linguistic analysis as inspiration" (lines 302-303). Previously, the discussion led with natural language, only in the last half discussing theory synthesis more broadly.

However, regardless of the main claim, I'm concerned that some of the framing may be somewhat misleading (or at least unclear). In particular, the authors seem to argue that applying Bayesian Program Learning (and specifically, the Bayesian aspect of it) are key to the success of their system. To me, this implies a probabilistic system, and likely one that integrates over parameters or hypotheses in some way. It also typically implies a search/optimization algorithm that has some kind of guarantees. But in fact the system is doing approximate MAP inference using a heuristic search that (I think) doesn't guarantee anything (though see remaining confusions under my detailed questions below). And although the system is technically probabilistic, the likelihood is degenerate (0/1) and the prior is an MDL prior, which can have a probabilistic interpretation but is essentially just a scoring function. I'm not saying there's anything wrong with these things in principle, but to a casual reader it would be easy to miss these points. (I suggest emphasizing the 0/1 likelihood by explaining eq 1 in words, not just as an equation. There are several precedents for this type of model with 0/1 likelihood from the word and morpheme segmentation literature, e.g. see work of Brent and/or Goldwater).

We agree with the reviewer that it is helpful to foreground both explicitly Bayesian (probabilistic) elements of the work as well as the connections to Minimum Description Length (MDL) learners. Our original submission initially described the model in terms of maximum a posteriori (MAP) inference. Our MAP objective is also formally equivalent to an MDL objective. Only later in the paper did we describe experiments that drew on the probabilistic Bayesian framing, by marginalizing over the latent grammar (averaging hypotheses, weighted by their probabilities) in order to make new predictions (our artificial grammar learning simulations), or performing hierarchical Bayesian inference while modeling uncertainty over each language-specific grammar (for the section on higher-level knowledge). Although we continue to think that it is helpful to narrate the work by first giving the simpler MAP/MDL-equivalent formulation, we understand that this was confusing. So we now preface the first MAP-style description by sign-posting our later use of more fully Bayesian techniques, saying (lines 98-99) "For now we consider maximum a posteriori (MAP) inference-which estimates a single $\langle T, L \rangle$ -but later consider Bayesian uncertainty estimates over $\langle T, L \rangle$, and hierarchical modeling." Then, after writing down our MAP objective for inference in Equation 1, we clarify that this version is equivalent to an MDL learner (lines 115-116): "In words, Eq. 1 seeks the highest probability theory which exactly reproduces the data, like classic MDL learners [21]." ([21] is Brent 1999)

To clarify the probabilistic aspects of the model important for the artificial grammar learning simulations, we have updated the model section to describe how the Bayesian formulation allows making new predictions about unobserved lexemes (i.e. new stems). This clarification requires the notion of a *paradigm matrix*, which is a 2d array of words where the columns range over inflections and the rows range over stems. Our data typically come as a paradigm matrix. Making probabilistic predictions for new lexemes relies on a

refactoring of our learning objective to make it more unambiguously Bayesian, which is now described in the text:

The input data \mathbf{X} typically come from a ‘paradigm’ matrix, whose rows range over inflections and
 whose columns range over stems. In this setting, the theory acts as a probabilistic generative model which
 places a probability distribution over paradigm rows, $P(\mathbf{X}_\sigma)$, where σ is the meaning of a stem, and \mathbf{X}_σ
 is the corresponding row of the paradigm. The likelihood $P(\mathbf{X}_\sigma)$ depends only on rules and affixes. Fig. 2
 illustrates this framing by partitioning the lexicon into stems and affixes. This framing is mathematically
 connected to our MAP setup as:

$$\begin{aligned}
 \quad & \underbrace{P(\mathbb{T}|\mathcal{UG}) \left(\prod_i P(\mathbb{L}(i, \mathbf{pfx})) P(\mathbb{L}(i, \mathbf{sfx})) \right)}_{\text{prior over rules and affixes}} \underbrace{\prod_\sigma \sum_{z_\sigma} P(z_\sigma) \prod_i \mathbb{1}[\mathbf{X}_{\sigma,i} = \text{Phonology}(\mathbb{L}(i, \mathbf{pfx}) \cdot z_\sigma \cdot \mathbb{L}(i, \mathbf{sfx}))]}_{\text{probabilistic likelihood model: } P(\mathbf{X}_\sigma)} \\
 \quad & \geq P(\mathbb{T}|\mathcal{UG}) \left(\prod_i P(\mathbb{L}(i, \mathbf{pfx})) P(\mathbb{L}(i, \mathbf{sfx})) \right) \prod_\sigma \max_{z_\sigma} P(z_\sigma) \prod_i \mathbb{1}[\mathbf{X}_{\sigma,i} = \text{Phonology}(\mathbb{L}(i, \mathbf{pfx}) \cdot z_\sigma \cdot \mathbb{L}(i, \mathbf{sfx}))] \\
 \quad & = \text{the MAP objective for inference in Eq. 1} \tag{2}
 \end{aligned}$$

where z_σ is the underlying form of stem σ , namely $\mathbb{L}(\sigma, \mathbf{stem})$. Thus, the MAP objective (Eq. 1) serves as
 a lower bound on the full Bayesian objective above. This bound will be tightest when each paradigm row
 admits very few possible stems. Typically only one stem per row is consistent with the rules and affixes,
 which justifies our MAP objective.

The reason why both of these formulations are important is because the MAP formulation treats the lexicon as a random variable that we need to infer (thus aligning with how linguists typically think about these problems), while this more manifestly Bayesian formulation is needed for making probabilistic predictions about unseen forms (for which probabilistic uncertainty over the lexicon is necessary).

*Moreover, since an exhaustive search would be left with no counterexamples (ie all problems solved), the search method must not be exhaustive. Therefore solutions are a product of *both* the model and the search heuristics, but this is rather deemphasized in the paper in favor of the "Bayesian" framing. (See more specific questions about this below.) This is particularly strange given that the idea of applying BPL to this problem is not new --- the authors themselves introduced it in a much simpler version earlier, but also citation [32] has a much more extensive treatment. So the main contributions to method here seem to be in the search and the meta-learning.*

Indeed, it is not merely being “Bayesian” which explains why the model works. It’s also a matter of having the right hypothesis space, which we think of as being part of UG. Additionally, having the right search heuristics matters, as does setting up the problem correctly to allow joint reasoning over rules and lexica. Consequently, we have expanded the discussion of our ablations as follows (lines 221-236):

Our results hinge on several factors. A key ingredient is a correct set of constraints on the space of
hypotheses, i.e. a universal grammar. We can systematically vary this factor: switching from phonological
articulatory features to simpler acoustic features degrades performance ('simple features' in Fig. 5A-B). Our
simpler acoustic features come from the first half of a standard phonology text [29], while the articulatory
features come from the latter half, so this comparison loosely models a contrast between novice and expert
phonology students. We can further remove two essential sources of representational power—Kleene star,
which allows arbitrarily long-range dependencies, and phonological features, which allows analogizing and
generalizing across phonemes. Removing these renders only the simplest problems solvable ('-representation'
in Fig. 5A-B). Basic algorithmic details also matter. Building a large theory at once is harder for human
learners, and also for our model ('CEGIS' in Fig. 5A-B). The recent SyPhon [37] algorithm strikes a different
and important point on the accuracy/coverage tradeoff: it aims to solve problems in seconds or minutes,
so that linguists can interactively use it. In contrast, our system's average solution time is 3.6hr (Fig. 5B).
SyPhon's speed comes from strong independence assumptions between lexica and individual rules, and from
disallowing non-local rules. These assumptions degrade coverage: SyPhon fails to solve 76% of our dataset.
We hope that their work and ours sets the stage for future systems that run interactively while also more
fully modelling the richness and diversity of human language.

Our revised introduction also calls out to the important roles of these pieces. We've strengthened the introduction's mention of the necessity of a good linguistic representation/inductive bias by saying that (line 53-54) "Only with this inductive bias [from linguistic formalism] can a BPL model then learn programs capturing a wide diversity of natural language phenomena". Previously, we'd mentioned the inductive bias in the introduction, but not its necessity. As in the original submission, the intro then proceeds to sign-post our need of new search methods: "BPL comes at a steep computational cost, and so we develop new BPL algorithms which combine techniques from program synthesis with intuitions drawn from how scientists build theories and how children learn languages" (lines 57-59). Collectively, we hope that this new extended discussion of the ablations, together with the tweak to the intro, serve to communicate to the reader that it is not merely being "Bayesian" that makes the system work—although it is an important piece.

This review probably sounds rather negative, so I want to emphasize that there were a lot of interesting ideas and strong results in this paper. E.g. it's interesting to me that such a simple MDL prior seems to work so well. I also really liked the meta-learning idea and the learned grammar looks very impressive, although see my notes below regarding lack of explanation of these parts. So I would definitely like to see this work published eventually, but I feel that in the current form it is trying to over-claim to appeal to a broader audience while skimming over important aspects.

Some more specifics about terminology and explanations:

- Inconsistent terminology: on page 3, "theory", "grammar", and \mathbb{T} are all used to refer to the (phonological and morphological) rules only. However, elsewhere (e.g. Figs 2, 3, ...) "grammar" refers to "theory" \mathbb{T} plus lexicon \mathbb{L} . Still elsewhere, (e.g.,

Figs 4 and similar), the term "lexicon" isn't used at all, instead it's referred to as "stems". Fig 5 refers to a "rule learner", when in fact it learns both lexicon and rules (a particularly odd term considering that the evaluation is actually against lexica, **not** rules). p14 refers to $\text{\textbb{T}}_0$ as "containing no phonological rules" -- without mentioning morphological rules.

Thank you for pointing out these terminological inconsistencies. We now use "grammar" in its usual sense: as an object which includes both morphophonological rules and a lexicon. We now consistently use "theory" (T) to refer exclusively to the rules. However, distinguishing affixes from stems is useful when viewing the system as synthesizing a probabilistic generative model over paradigm rows, a view we choose to illustrate in Figures 2/4 because it clarifies the different roles that stems and affixes play here. Other inconsistencies are addressed as follows:

- We now say (lines 93-94): "Such form-meaning pairs (stems, prefixes, suffixes) live in a part of the grammar called the *lexicon*"
- Figure 5A's caption has been changed from "Program-synthesis-based phonological rule learner..." to "Program-synthesis-based learners..."
- We've more neutrally described the initial state of the search algorithm (T₀) as simply "containing no rules" (Methods, line 363).

Due to these inconsistencies and lack of detail in S3, I'm left with considerable uncertainty as to how the search method actually proceeds. A concrete example going through the first one or two iterations of the procedure would help immensely. For example, suppose the first lexeme has two items: <MEET (INF), [mit]>, <MEET (3SG), [mits]>. The "theory" $\text{\textbb{T}}$ starts out empty, so it isn't even obvious to me why these would serve as counterexamples to that $\text{\textbb{T}}$. Why can't these items simply be added to the lexicon as is? ie memorizing all of the items provides an explanation with no counterexamples.

*Perhaps there is an assumption (not stated anywhere, I think) that we know how many morpheme-meanings are in each item, what they are, and that each must have its own form listed in the lexicon (ie we **must** split up the surface forms somehow). Then I suppose we'd need a lexicon with at least 3 meanings (MEET, INF, 3SG), and to get the observed words we would need need at least two rules (?) to combine them. From just these examples, we might decide that /s/ corresponds to (3SG), but then we would eventually see counterexamples such as <NEED (3SG), [needz]>.*

Initially, T₀ contains no rules -- not even the morphological rule for appending a suffix for 3SG. Additionally, our setup assumes that every surface form gets broken up into a stem and (possibly empty) affixes, which the lines 116-118 now clarify: "This equation forces the model to explain every word in terms of rules operating over concatenations of morphemes, and does not allow wholesale "memorizing" of words in the lexicon." For that reason, the particular paradigm row you mention, and indeed the first paradigm row of any problem, constitutes a counterexample. Our revised appendix takes inspiration from your example

and walks the reader through the first few iterations of the algorithm. To draw the reader to this example, we have packaged it with the formal description of the algorithm and explicitly pointed to it in the revised main text (line 182-183: "S3.3 gives a concrete walk-through of the first few iterations.") The corresponding section of the supplement (S3.3) contains this concrete example:

As a concrete example, consider a data set of English verb inflections in infinitive and third-person plurals.
Suppose the batch size is two. If the first paradigm row is [mit] and [mits] (the pronunciations of the words
"meet" and "meets"), then these are the first two words that the system considers. Initially, \mathbb{T}_0 contains
no rules. So, these words serve as a counterexample, because the third-person-singular morphology, namely
that a suffix must be appended (and that in the lexicon this suffix is recorded as /s/) has not yet been
inferred. Running Sketch on this example would update the grammar to contain the third-person-singular
suffix /s/, and introduce no new phonological rules. Suppose that the next paradigm row is [it] and [its]
(the pronunciations of the words "eat" and "eats"). The system will find that the current grammar, when
supplemented with the stem /it/, explains this example, and so it does not serve as a counterexample, because
the morphophonology inferred from the first batch is consistent with this example. Suppose that the next
row of the paradigm is [nid] and [nidz] (pronunciations of the words "need" and "needs"). There is no stem
which can explain this paradigm row, given the current affixes and rules. Therefore it is a counterexample,
and in the next iteration the system will accommodate this counterexample by introducing a phonological
rule about explains the alternation between /s/ and /z/.

*- It is mentioned that the CEGIS method is exhaustive, and that the current search method is needed in order to be more efficient. The way it is described, it sounds like it should still find *some* solution, even if not the best one (p8 Supp: "we progressively increment the maximum edit distance until Sketch discovers a satisfying solution".) Yet this is clearly not the case, since if it were, the evaluation would be 100% for all problems attempted. I assume the limiting factor is still time, but how is this limit operationalized? (For both your system and for CEGIS, as that seems critical to understanding the comparison in the evaluation.)*

We run all methods on each problem for 24 hours. Our new revision says this in the text as previously, but makes it more explicit through the new Figure 5B, which plots lexicon accuracy as a function of runtime for each model (with the runtime axis stopping at 1 day of compute). Yet, even if in those 24 hours the model discovered a grammar which "covered" the data in the problem in the sense of generating the entire paradigm matrix, it still would not necessarily get 100% because we are comparing against ground-truth lexica. In other words, it's possible for the model to discover rules and a lexicon which fit all the data, but where the underlying forms differ from what a linguist would identify as a correct solution. This evaluation is now clarified in the text by saying: "We first measure the model's ability to discover the correct lexicon. Compared to ground-truth lexica, our model finds grammars correctly matching the entirety of the problem's lexicon for..." (lines 198-200).

If the system fails to find a solution, is it possible that a solution could be found if different mini-batches or data ordering were used, or only by increasing the allowed edit distance from the current solution? I gather that different solutions might be found (?) by changing ordering or initialization, otherwise I'm not sure how the multiple different solutions in the artificial grammar section were found.

Should the system fail to find a solution, it is possible that different minibatching—either changing the batch size or changing the data ordering—would result in successful solving. We conjecture that larger batch sizes are generally better, because the solver will see more data at once, and so be less likely to make myopic search moves. The supplement, in section S3.3, now says the following (lines 114-122):

“We conjecture that larger batch size generally leads to better convergence, because this exposes the SAT solver to more data at once, which on balance should lead to less myopic search moves. Yet larger batch sizes increase compute requirements, both because the size of the SAT problem grows linearly with batch size and because the search radius may need to grow larger with increased batch size. Accordingly, for allophony alternation problems, we batch the entire problem at once, because these problems are much easier. Our selection of a minibatch size of 9 was motivated by informal pilot experiments suggesting that after around 9 new words the solver performance degraded severely; due to the high compute cost of running these simulations, we did not perform a systematic hyperparameter sweep, and the ‘optimal’ batch size may differ from the one used.”
The revised manuscript now calls out to this section on lines 182-183 of the main text.

For the artificial grammar learning simulations, we find multiple solutions by computing the Pareto frontier, which relies on the fact that Sketch supports exact methods for finding pareto-optimal solutions out-of-the-box. The Methods section has been revised to describe this:

**Few-shot artificial grammar learning**
We present our system with a *training set* of words from a target language, such as the ABA language
(e.g. ‘wofewo’, ‘mikami’, ...). Then we compute the likelihood assigned to held-out words either *consistent*
with the target grammar (e.g. following the ABA pattern) or *inconsistent* with the target grammar (e.g.
following the ABB pattern, such as ‘wofefe’, ‘mikaka’, ...). The probability assigned to a held-out test word,
conditioned on the training set, is approximated by marginalizing over the Pareto-optimal grammars for the
train set, rather than marginalizing over all possible grammars:

$$374 \quad P(\text{test}|\text{train}) = \sum_{\mathbb{T}, \mathbb{L}} P(\mathbb{T}, \mathbb{L}|\text{train}) P(\text{test}|\mathbb{T}, \mathbb{L}) \quad (8)$$

$$375 \quad \approx \sum_{\langle \mathbb{T}, \mathbb{L} \rangle \in \text{ParetoFrontier}(\text{train})} P(\mathbb{T}, \mathbb{L}|\text{train}) P(\text{test}|\mathbb{T}, \mathbb{L}) \quad (9)$$

which relies on the fact that Sketch has out-of-the-box support for finding Pareto-optimal solutions to
multiobjective optimization problems [51].
We approximate the likelihood $P(\text{test}|\mathbb{T}, \mathbb{L})$ by finding the latent stem z_σ which maximizes the probability
of the test word. This is equivalent to finding the shortest stem which will generate the test word, given the
affixes in \mathbb{L} and the rules in \mathbb{T} . Formally:

$$381 \quad P(\text{test}|\mathbb{T}, \mathbb{L}) \approx \max_{z_\sigma} P(z_\sigma) \mathbf{1} [\text{test} = \text{Phonology}(\mathbb{L}(\text{pfx}) \cdot z_\sigma \cdot \mathbb{L}(\text{sfx}))] \quad (10)$$

where we have reused the notation of Eq. 2 while dropping the dependence of the prefix/suffix on the
inflection, because there is only one inflection for these data sets.

- The meta-learning model is not fully explained anywhere. In particular, how are $P(M)$ and $P(T_d, L_d | M)$ defined? I think the former comes from previous work on fragment grammars (though it's not safe to assume most readers would know that), and the latter seems to be the critical piece, since it replaces the MDL prior of the UG and therefore seems to be responsible for all of the improvement achieved by using the meta-model. Fragment grammars define a probability distribution over rule fragments --- how do these relate to the purely symbolic set of rules shown in S5? Is that set of rules the ones with non-negligible probability under the meta-grammar? Are all other rules still possible?

Thank you for pointing out these ways in which we can clarify the metalearning component. We've revised the Methods section to address your questions by including the following:

Synthesizing a metatheory
At a high level, inference of the cross-language fragment grammar works by maximizing a variational-
particle [52] lower bound on the joint probability:

$$387 \quad \log P(\mathbb{M}, \{\mathbf{X}_d\}_{d=1}^D) \geq \log P(\mathbb{M}) + \sum_{d=1}^D \log \sum_{\substack{(\mathbb{T}_d, \mathbb{L}_d) \in \\ \text{support}[Q_d(\cdot)]}} P(\mathbf{X}_d | \mathbb{T}_d, \mathbb{L}_d) P(\mathbb{T}_d, \mathbb{L}_d | \mathbb{M}) \quad (11)$$

where this bound is written in terms of a set of variational approximate posteriors, $\{Q_d\}_{d=1}^D$, whose support
we constrain to be small, which ensures that the above objective is tractable. We alternate maximization
with respect to \mathbb{M} (i.e., inferring a fragment grammar from the theories in the supports of $\{Q_d\}_{d=1}^D$), and
maximization with respect to $\{Q_d\}_{d=1}^D$ (i.e., finding a small set of theories for each data set that are likely
under the current \mathbb{M}). Our lower bound most increases when the support of each $\{Q_d\}_{d=1}^D$ coincides with
the top- k most likely theories, so at each round of optimization, we ask the program synthesizer to find the
top k theories maximizing $P(\mathbf{X}_d | \mathbb{T}_d, \mathbb{L}_d) P(\mathbb{T}_d, \mathbb{L}_d | \mathbb{M})$. In practice we find the top $k = 100$ theories for each
data set.
We represent \mathbb{M} by adapting the Fragment Grammars formalism [44]. Concretely, \mathbb{M} is a probabilistic
context free grammar (PCFG) that stochastically generates phonological rules. More precisely, \mathbb{M} generates
the syntax tree of a program which implements a phonological rule. In the Fragment Grammars formalism
one first defines a *base grammar*, which is a context-free grammar. Our base grammar is a context free
grammar over SPE rules (Fig. S6). Learning the fragment grammar consists of adding new productions to
this base grammar (the “fragments”), while also assigning probabilities to each production rule. Formally,
each fragment is a subtree of a derivation of a tree generated from a non-terminal symbol in the base
grammar; informally, each fragment is a template for a piece of a tree, and thus acts as a schema for a piece
of a phonological rule. Learning a fragment grammar never changes the set of trees (i.e., programs and
rules) that can be generated from the grammar. Instead, through a combination of estimating probabilities
and defining new productions, it adjusts the probability of different trees. See Figure S6, which shows the
symbolic structure of the learned fragment grammar.
This fragment grammar gives us a learned prior over single phonological rules. We define $P(\mathbb{T}, \mathbb{L} | \mathbb{M})$ by
assuming that rules are generated independently, and that \mathbb{M} does not affect the prior probability of \mathbb{L} :

$$410 \quad P(\mathbb{T}, \mathbb{L} | \mathbb{M}) = P(\mathbb{L} | \text{UG}) \prod_{r \in \mathbb{T}} P(r | \mathbb{M}) \quad (12)$$

Our prior over fragment grammars, $P(\mathbb{M})$, works by following the original work in this space [44] by assuming
that fragments are generated sequentially, with new fragments generated from the current fragment grammar
by stochastically sampling them from the current fragment grammar. This encourages shorter fragments,
as well as reuse across fragments.
We depart from [44] in our inference algorithm: while [44] uses Markov Chain Monte Carlo methods
to stochastically sample from the posterior over fragment grammars, we instead perform hillclimbing upon
the objective in Eq. 11. Each round of hillclimbing proposes new fragments by antiunifying subtrees of
phonological rules in $\{\mathbb{T}_d\}_{d=1}^D$, and re-estimates the continuous parameters of the resulting PCFG using the
classic Inside-Outside algorithm [53]. When running Inside-Outside we place a symmetric Dirichlet prior
over the continuous parameters of the PCFG with pseudocounts equal to 1.

- How does using the meta-grammar lead to more solutions being found? E.g., is it allowing each iteration of the algorithm to run faster (thus, more iterations can run within the time you have available), or allowing better rules to be picked early on so that the search doesn't go off into a bad part of the space where nearby theories can't solve the problem? (My confusion here is related to the previous point, ie. I don't understand what the limits are that cause certain problems to fail.)

The revision now clarifies how learning the metatheory helps us converge to better solutions, saying (lines 292-294): "a better inductive bias steers the incremental synthesizer toward more promising avenues, which decreases its chances of getting stuck in a neighborhood of the search space where no incremental modification offers improvement." This is the second explanation that you offered.

- More terminology: on p7 (and Fig 7), you say that the solutions found are a trade-off between "parsimony (size of grammar) and fit to data (size of lexicon)". But your lexicon is not data, it is latent. *Both* of these terms are part of the parsimony prior, and as noted elsewhere, all of your solutions have a perfect fit to the data. So describing the trade-off in this way is misleading.

Viewing the model as a probabilistic generative model over paradigm rows (as the new Eq. 2 derives) clarifies how these visualizations show a tension between parsimony (high prior) and fit to data (high likelihood). We now say that we seek: "Pareto-optimal solutions [38] to the trade-off between parsimony and fit to data. Here parsimony means size of rules and affixes (the prior in Eq. 2); fit to data means average stem size (the likelihood in Eq. 2); and a Pareto-optimal solution is one which is not worse than any other along both these competing axes" (lines 249-251). This last clarification serves to briefly explain Pareto optimality for readers unfamiliar with the concept, which addresses Reviewer #3's other comment:

- Similarly, I don't know if it is safe to assume all readers will understand what 'Pareto fronts' are (p7).

- When computing the edit distance between two theories, is it assumed that any rule has a distance of 1 from any other rule? I.e., there is no difference between making a small change to a rule, versus introducing a completely different rule?

You are correct that the model makes no distinction between a small change to a rule versus a large change, or even replacing it with a completely different rule. The supplement now clarifies and justifies this choice by saying (lines 103-108): "We define the edit distance, $d(T1,T2)$, between a pair of theories $T1,T2$ by counting the number of insertions, deletions, substitutions, and swaps that separate the sequences of rules for $T1$ and $T2$. Any modification to a rule counts as a complete substitution, so entire rules are resynthesized wholesale rather than, eg, have individual feature matrices 'edited'. This coarse-grained notion of edit distance has the advantage that it can be easily encoded in a SAT solver, and we hypothesize it may be less prone to getting trapped in local optima because it encourages larger search moves."

- You mention some ablated models in Fig 5, but I don't see an explanation anywhere of what the "acoustic features" are as opposed to the phonological articulatory features; or what "ablating phonetic features and Kleene star" would look like. These should be explained in the Supplement.

Thanks for suggesting this. We chose the articulatory/phonetic feature distinction because Odden's phonology textbook starts by introducing phonetic features (such as +fricative) and only later introduces the more sophisticated feature system commonly used in phonology, which is "articulatory" in the sense of expressing motor features of sound production (such as +continuant). Thus this ablation can be thought of as contrasting a phonology student with "day one features" vs "month two features". We've summarized this in the text (lines 223-228): "...switching from phonological articulatory features to simpler acoustic features degrades performance ('simple features' in Fig. 5A-B). Our simpler acoustic features come from the first half of a standard phonology text [25], while the articulatory features come from the latter half, so this comparison loosely models a contrast between novice and expert phonology students. We can further remove two essential sources of representational power—Kleene star, which allows arbitrarily long-range dependencies, and phonetic features, which allows analogizing and generalizing across phonemes."

As you suggested, we have expanded upon these ablations in the supplement (lines 177-199):

S3.5 Ablation studies
We studied several ablations³ of our system; see Fig. 5. We found that basic representational concerns
matter most: one *needs* the right rule representation, which we think of as being part of universal grammar.
We studied the effect of changing the feature system, as well as the effect of ablating two key computational
mechanisms (having features at all, and having Kleene star).
- • We first changed the feature system from so-called ‘articulatory’ features to ‘phonetic’ features. Typi-
cally introductory phonology courses start by introducing phonetic features (features of sounds). Later
one typically learns that these features can be more concisely and more generically expressed in terms of
features of the motor actions required to produce those sounds (‘articulation’). For example, *fricative* is
an phonetic feature which becomes deprecated by the articulatory feature of *continuant*. Concretely, we
took Odden’s text *Introducing Phonology* [4] and identified the features used in the first five chapters
as ‘phonetic’ features. Features in Chapter 6 onward we identify as ‘articulatory features’.
- • For a more drastic demonstration of the centrality of basic computational mechanisms, we further
ablate *all* phonological features. The system can still express rules in terms of specific phonemes,
but cannot generalize and analogize across phonemes. We also remove Kleene star, which means not
allowing the system to express rules whose triggers abstract over the number of times a phoneme
occurs. Recall that this is notated with a subscript ₀, thus all of our example rules with this subscript
are unexpressible by this ablation. In principle, this ablation can still learn rules whose behavior is
identical to the correct rules, simply by memorizing every phoneme for which a rule applies (due to
the ablation of features), and every sequence length for which a rule applies (due to the ablation of
Kleene star). In practice, the system no longer has the inductive bias to learn such generalizations;
and furthermore, search becomes harder because the programs become much longer due to the need
to memorize many specific cases.

where footnote #3 defines an ablation as: “An ablation is a variant of a system with components removed entirely or changed to be less powerful. Ablations are studied to understand the importance of the ablated components.”

- *The comparison to [32] points out that [32] fails to solve 76% of the problems. But [32] claims that their system is 2 orders of magnitude faster than yours. There is no "correct" position on the speed/accuracy trade-off, but I would like to see a clear statement of how long your experiments take to run, and an acknowledgment that [32] is much faster. You mentioned 40 CPUs, but no length of time.*

Both SyPhon [32] and our system strike interesting and important points on the speed/coverage tradeoff. SyPhon aims for fast solution times—on the order of seconds or minutes—so that their system could serve as a real-time interactive tool for linguists. To attain this speed, SyPhon factors the synthesis problem by making strong independence assumptions, which weakens coverage. We aim for broad coverage, but at the expense of orders of magnitude more compute. It’s an interesting question whether one could get SyPhon-level runtime while maintaining our level of coverage, and we feel the first stage to answering that question is to first engineer methods which can solve many of these problems, and then ask how one might speed up those methods. We hope that our work, combined with theirs, can help lead to such developments.

Motivated by your feedback, we have added the following to the revision (lines 230-236): “The recent SyPhon algorithm strikes a different and important point on the accuracy/coverage tradeoff: it aims to solve problems in seconds or minutes, so that

linguists can interactively use it. In contrast, our system's average solution time is 3.6hr (Fig. 5B). SyPhon's speed comes from strong independence assumptions between lexica and individual rules, and from disallowing non-local rules. These assumptions degrade coverage: SyPhon fails to solve 76% of our dataset. We hope that their work and ours sets the stage for future systems that run interactively while also more fully modelling the richness and diversity of human language.

- Finally, if the goal of the system is to learn interpretable theories, then I don't understand why the quantitative evaluation focuses only on whether the stems match the gold standard. You say that there are "many rules" that could potentially explain the data -- but typically linguists would consider only one or two possibilities to be reasonable. Why don't you evaluate your learned rules against the gold standard rules? (The main examples provided are very useful, but since you've also listed the system output for every problem, why not add the gold standard for each of those, and compute some summary statistics?)

Thank you for suggesting computing summary statistics on the rule outputs. We've now added such evaluations to the manuscript.

First, we believe that evaluating rules deserves extra discussion. A key difference between evaluating rules and evaluating lexica is that rules have both *syntax* and *semantics* (i.e., an intensional definition in terms of focus/change/triggers, and an extensional definition in terms of its behavior on phoneme sequences). By evaluating the accuracy of the lexicon, together with our constraint that the lexicon+rules exactly generates the data, our previous evaluation was effectively assessing whether the composition of all the synthesized rules had the right behavior on the gold-truth lexicon. This observation holds because, whenever the synthesized rules can map the gold-truth stem to its observed inflections, the synthesized stem will (almost always) coincide with the gold-truth stem. Thus our previous evaluation metric was generally a lower bound on rule accuracy, when measured by testing the semantics of the composition of the synthesized rules.

Therefore, the natural rule evaluation strategy is to check whether individual predicted rules have the same semantics as gold-truth rules. For almost all of these problems, we do not have gold-truth rules. So, we worked with Adam Albright (a professional linguist and co-author on this paper) to evaluate the predicted rules. We measured *precision* (what fraction of the predicted rules are semantically equivalent to a correct rule) as well as *recall* (what fraction of the correct rules are semantically equivalent to a predicted rule). Because this required manual solving of the problems and "grading" of the synthesized solutions, we randomly sampled 15 problems to evaluate. Encouragingly, both precision and recall are positively correlated with lexicon accuracy (the new Figure 5C), and, intuitively, negatively correlated with the number of gold-truth rules (problems with more processes would generally be harder for a human linguist to analyze, and are also harder for our system). The revision discusses these results on lines 200-210.

Various other minor issues:

Why is [32] referred to as "PhonoSynth"? Their paper calls the system "SyPhon".

Fixed! (PhonoSynth was the name given to the software tool's Github repo, but we should call it SyPhon as you say)

The first sentence of S1 says "brackets" twice when I believe the second one should say "slashes". That said, the remainder of the section seems rather inconsistent about using brackets vs slashes correctly. (I think there are several cases of slashes that should be brackets, or else need the 't' replaced with a 'd'.)

Thank you for pointing out where we had not used brackets for surface forms in S1. This has now been fixed by using slashes only for underlying forms and brackets only for surface forms in supplement lines 19-66.

I'm not sure 'SPE' is ever spelled out, or 'SPE-style' rules defined (or SPE cited).

We have updated footnote 4 (bottom of page 4) to underline the relevant parts of the abbreviation: "These are sometimes called *SPE-style rules* since they were used extensively in the *Sound Pattern of English* [22]" (where [22] refers to Chomsky & Halle, 1968)

Fig S6: I infer from the italicised note that the subscript 0 (as in *FeatureMatrix_0*) is likely your notation for Kleene star, but that could be stated much more explicitly.

The main text now says, when describing how rules are represented (lines 147-149): "The subscript 0 denotes zero or more repetitions of a feature matrix, called the 'Kleene star' operator (i.e., [+voice]₀ means of zero or more repetitions of [+voice] phonemes)"

Review #4:

This submission details the use of program synthesis to solve textbook morphophonology problems. The authors report several results: extensionally correct grammars (as far as the problems, which are often quite brief, are concerned) can be learned by this method, a learned meta-theory/cross-linguistic inductive bias improves performance, and this is useful for modeling child performance in artificial grammar learning experiments as well.

My comments are largely structured around what, normatively, I think the paper should tell the reader. Some of these things are purely rhetorical, others are comparisons or observations I would recommend to the authors before acceptance, others are formal issues whose solutions are out of scope of this paper but will determine the overall influence of this submission.

Below, I many times suggest that the authors provide more information. Realize that there may be length limitations for this format, I would suggest the authors omit discussion of the artificial language learning or relegate it to an appendix. This is an interesting result but feels somewhat far afield; only a small percent of the overall readers will be concerned with the validity of such approaches.

We are happy to relegate the artificial language learning simulations to an appendix. However, we believe that they have the important function of studying the sample efficiency of the proposed model. In the resubmission we have included these results in the main text but can easily refactor the manuscript to have these in the supplemental materials if needed.

Before I begin in more detail, the review form asks me to address a number of specific questions, which I'll address now.

**Is the work convincing?* I would tentatively say yes, though with some strong caveats. First, though there are not really any meaningful comparisons to other work. Figure 5 includes a partial comparison to something called Phonosynth. This method is not described in any detail, the comparison is partial (only some languages were fed to Phonosynth perhaps the authors have used results from that paper rather than replicated the results themselves?), Phonosynth matches or outperforms (depending on condition and language) the proposed methods, and finally no overall accounting is given for the comparison (i.e., which is better?).*

Thank you for this encouragement to further explain our comparison with Phonosynth (which is better called SyPhon—we use this name in the new draft). SyPhon (aka Phonosynth) is a recently developed approach for solving linguistic problems like those we consider (both synthesizing the lexicon and rules). One main difference between SyPhon and our work is that SyPhon strives to find solutions quickly (on the order of minutes or seconds). It accomplishes this runtime efficiency by making strong independence assumptions, both between the lexicon and the rules and also between individual rules. This factors the problem into independent subproblems, which can be exponentially more efficient to solve. SyPhon further assumes local, non-interacting rules. These strong assumptions limit the scope of problems solvable by SyPhon: We ran SyPhon on all 70 benchmarks, but it does not find a solution for most of the benchmarks, hence the 0% accuracy across many problems. In contrast, our system jointly reasons about rules, their interactions, and the lexicon, and also synthesizes over a richer space of rules with non-local interactions: this allows much broader coverage. However, our system also requires orders of magnitude more compute, as the new Figure 5B illustrates.

The revision clarifies these differences by saying (lines 230-236): “The recent SyPhon algorithm strikes a different and important point on the accuracy/coverage tradeoff: it aims to solve problems in seconds or minutes, so that linguists can interactively use it. In contrast, our system's average solution time is 3.6hr (Fig. 5B). SyPhon's speed comes from strong independence assumptions between lexica and individual rules, and from

disallowing non-local rules. These assumptions degrade coverage: SyPhon fails to solve 76% of our dataset. We hope that their work and ours sets the stage for future systems that run interactively while also more fully modelling the richness and diversity of human language.”

Relatedly, we believe SyPhon is the best system to compare with: it considers solving textbook problems in the exact same format as our system, its development overlapped with our work here, and although it solves fewer problems, its runtime efficiency compliments our own work. We provided the authors of SyPhon with an early version of our data set, and they used a subset of the corpus we provided them for their benchmarking experiments. We were in touch with the authors of SyPhon, who explained to us how to run their code on the full benchmarks.

Secondly, nothing is said about the time and space complexity of the Sketch model used to find solutions, nor about its hardware demands or the amount of wall-clock time it requires. Does it take a cluster running for weeks to solve these problems, or does the whole thing run in seconds?

Thank you for encouraging us to explain and analyze the runtime needed to solve these problems using these systems. In addition to the above coverage/runtime trade of discussion, we’ve analyzed the convergence behavior as a function of run time, showing that the model converges, on average, after 3.6 hours (new Figure 5B). A section of the Supplement (S3) explains how incrementally synthesizing programs exposes new opportunities for parallel computation, and the revision now refers to this content twice in the text by saying: “it [incremental program synthesis] also exposes opportunities for parallelism (S3)” (lines 181-182) and “All models are run with a 24 hour timeout on 40 cores. Only our full model can best tap this parallelism (S3)” (caption of Figure 5B).

Virtually all constraint-based program synthesizers are limited in the ways that they can exploit parallelism, because the underlying constraint solver (SAT/SMT) is a serial algorithm. Sketch exploits parallelism during constraint generation, but runs serially during constraint solving. Incremental synthesis allows parallel exploration of independent edits to the current grammar, so allows more of the constraint solving process to execute in parallel. While our method is not a general solution to parallel constraint solving, it allows relatively more parallel computation than Sketch by itself. As mentioned, the revised Supplement (S3) discusses these issues (lines 137-163), which are called out to in the main text.

Finally, the authors provide no guarantees, approximation bounds, or evidence that the theories are even decidable. My limited understanding is that in general SMT theories are undecidable, a very serious problem, but the authors may be able to show that they are only considering a decidable subset and perhaps discuss the size of this subset as a function of the properties of the input.

All Sketch problems are decidable (as are most uses of SMT). Because of how Sketch uses its backend solver, it is equally appropriate to describe it as a Boolean Satisfiability (SAT) solver. We have changed the draft to say that it uses SAT instead of SMT. This description makes the problem more transparently decidable. (As a technical note: both characterizations are equally accurate, and different papers in the program synthesis literature characterize Sketch as based on SAT or SMT. Sketch emits constraint problems that are essentially SAT, except that they have special mutual-exclusivity clauses. Sketch uses a custom SAT solver which specially handles these mutual exclusivity clauses, so strictly speaking it is SAT modulo mutual-exclusivity. But typical use of SMT involves much more complicated “modulo theories”, such as “modulo linear inequalities”.)

Even then, I don't know in what sense the overall approach can be called "exhaustive but efficient" but also a naive approach is said to have a "steep computational cost": that can only be concluded if the theories are all decidable, the time and space complexity of solution is some reasonably slow-growing polynomial function. Do counter-example synthesis and test-driven synthesis accomplish this? I feel strongly that the paper is incomplete without discussion on these notes.

Naively, exhaustively solving these combinatorial search problems would seem to take exponential time in the worst case. However: SMT solvers use clever heuristics that, in practice, can often discover ways of exploiting structure within the constraint problem to solve it more quickly - although there are no general formal guarantees. Thus, it's fair to say that SMT solvers are “surprisingly efficient” yet still come at a steep computational cost, at least as the constraint problem grows in size. We've rephrased our characterization of SMT solving as a “exhaustive but relatively efficient search” (line 165), because SMT is indeed efficient relative to exhaustive enumeration, but we maintain our characterization of these methods as coming at a “steep cost”.

One obvious comparison to be made is against the so-called input strictly local functions studied by Chandee and colleagues, which seem to cover a wide variety of morphophonological functions and which have much stronger learning guarantees, though the representations learned arguably less closely resemble those used by linguists.

Thank you for pointing us to this interesting related work (ie, “Learning Input Strictly Local Functions From Their Composition” Hua & Jardine 2021). We feel that this work is complimentary: While the authors of this work focus on obtaining strong theoretical guarantees for a more restricted class of transducers (2-input strictly local, so rules are sensitive to input bigrams), we focus on obtaining empirical simulation results with (what we believe to be) a more expressive class of rules. That being said, it is possible that the 2-input strictly local restriction would lend itself to more scalable grammar induction and may interact well with the inference techniques we use in this paper, albeit at the cost of

missing at least some phenomena. We think that this is an interesting possibility and a target for future work.

The new draft now raises this related work and places it in context, saying (lines 218-219) that “[o]ther works attain strong theoretical learning guarantees by restricting the class of rules: e.g., Hua & Jardine 2021 [36] considers 2-input strictly local functions”.

**Do you think this paper will influence thinking in the field?* I would tentatively say yes once again. This is by no means the first work using program synthesis approaches to morphophonology, but this would be the first and probably most high-profile journal article detailing the approach that I'm aware of, unless [19] is published first. And clearly there's more work to be done.*

**Is the paper reproducible?* I would suppose so. I briefly looked at the included code assets, which seem to be reasonably well documented. The authors use the Python 2.7 programming language, which is now end-of-lifed in January 2020 and will not be widely available in a few years, so a quick port to Python 3.x would make this more future-proof, but is not something I would require the authors to do.*

Some major observations, in the order in which they appear in the paper

I think the authors initially oversell the conclusions. This is described in the abstract as a "framework for algorithmically synthesizing models of human language, focusing on morpho-phonology". While this general framework may in fact be a general purpose for all kinds of subareas of human language, I don't see any reason to suppose it is. This possibility could be discussed in more detail in the concluding remarks but it doesn't seem justified by the current paper.

Thank you for the suggestion. The abstract has been changed to instead say: “We present a framework for algorithmically synthesizing models of a basic part of human language: morpho-phonology, the system that builds word forms from sounds.”

I understand that in the introduction the authors are trying to work with the child-as-scientist metaphor but I find something strange about notions like "language-specific theories". For me, as a working linguist, my account of a single language is an analysis or grammar, and referring to a morphophonological analysis or grammar as a theory is jarring. To add to the complexity, SMT problems are often called "theories" as well. To resolve this terminological morass, I'd suggest the authors use "analysis" or "grammar" to refer to language specific theories and "problem" to refer to what's submitted to the SMT solver, though the authors are welcome to try something different.

Thank you for suggesting clearer terminology. To address these concerns, as described in our response to Reviewer #3, we have revised the manuscript to consistently distinguish between “grammar”, “lexicon”, and “theory” (which we take to mean morphophonological

rules). As mentioned earlier, we have replaced the claim that Sketch has an SMT backend with the equally-valid claim that it has a SAT backend, which also avoids confusion caused by this other use of the word "theory".

The authors' description of UG in the second paragraph is simply not consistent with my understanding of the term. The framework of representations and processes for morphophonology, assumed to be common to all human languages is certainly part of UG, defining a hypothesis space for possible rules, but it's not the sum total of UG, which also includes the induction algorithm and its inductive biases, for instance. Furthermore, children bring more than just UG (and the primary linguistic data) to language acquisition: they also bring domain-general limitations and skills: constraints on audition, attention, working memory, etc. These are often thought of as constraints but they might also be thought of as resources for learning, resources which happen to be finite. Chomsky (2005), in an influential paper, refers to these collectively as "third factors".

We do not mean to communicate that UG is *only* concerned with morphophonology, nor that language learning cannot draw on more general cognitive resources. We have revised the introduction to characterize UG as "those linguistic resources that they [children] bring to the table for language acquisition", rather than our previous phrasing of "the sum total of linguistic resources that they bring to the table for language acquisition" (line 31-32). This new phrasing leaves open the possibility of more general cognitive resources coming into play, while also avoiding getting bogged down into a nuanced theoretical discussion of UG, which we feel would be out of scope for this work.

"Second, children easily acquire language from quantities of data that are modest by the standards of modern artificial intelligence." I think this is true, but should be backed up with example or citation. How much data do children take in? How much is non-modest for modern AI research?

Although estimates vary, children hear between on the order of a million and ten million words/year (including non-child directed speech; Dupoux 2018, "Cognitive science in the era of artificial intelligence: A roadmap for reverse-engineering the infant language-learner" and Frank, Tenenbaum, & Gibson 2013). While all these numbers are for full language acquisition, we also know that children can acquire a single pattern similar to those studied in our work from just 4 examples (as studied in Gerkin 2006, which we explore in the artificial grammar learning section of the paper). The recent trend in artificial intelligence has been to push for larger and larger data sets: for instance, GPT3 learns from 300 billion tokens. We have footnoted our claim about the relative quantities of data required by children and recent AI systems with:

"Dupoux [16] estimates 460,000 to 25 million words/year, according to SES (including non child directed speech); Frank et al. [17] gives 3-12 million words/year. Recent neural language models train on 3-4 orders of magnitude more data (GPT3: [18])."

The identification of /z/ as the regular past is not universally accepted by linguists (though it is probably the best analysis); the authors may want to mention that. Some linguists also transcribe the epenthesis vowel as bar-i. The rules are also not properly stated: the prose statement here suggests that all unvoiced consonants select /s/, but then revises that. I would state the rules as surface-true form: "/s/ is selected by unvoiced consonants except ..."

Absolutely, you are correct that other, less standard analyses are possible here. But, we feel that the purpose of this introductory paragraph is to give linguistics-naive readers a “crash course” on basic phonology. From that point of view, raising these debates could cause unnecessary confusion. Thank you very much also for noticing the bug in our description of this English phonology. We have changed our example to avoid this problem as follows:

We focus on theories of natural language *morpho-phonology*—the domain of language governing the
interaction of word formation and sound structure. For example, the English plurals for *dogs*, *horses*, and
*cats* are pronounced /dagz/, /hɔrsəz/, and /kæts/, respectively (plural suffixes underlined).² Making sense
of this data involves realizing that the plural suffix is actually /z/ (part of English *morphology*), but this
suffix transforms depending on the sounds in the stem (English *phonology*). The suffix becomes /əz/ for
*horses* (/hɔrsəz/) and other words ending in stridents such as /s/ or /z/; otherwise the suffix becomes /s/
for *cats* (/kæts/) and other words ending in unvoiced consonants. Full English morphophonology explains

I see why the authors say that the theories are "human understandable" but I think "theories of the causal relationship between form and meaning" and "human-like AI" said on page 2 is maybe stronger than many linguists would agree. There is of course a long-standing question the degree to which our theories are also theories of production and perception, and whether production and perception grammars are equivalent. The authors presume a very strong equivalence that few linguists would endorse. This assumption is also repeated on 86 when the goal of acquisition is stated as a grammar which "maps back and forth between form...and meaning".

Thank you for drawing our attention to this wording. We did not mean to imply that we assumed that production and comprehension were necessarily identical to the employment of grammatical knowledge, nor that grammatical knowledge exhausted all the kinds of information present in any linguistic system; only that the relations between form and meaning often described by grammars are an important part of the learning problem. We have softened the language to reflect this, now saying on lines 87-89:

“One central problem of natural language learning is to acquire a grammar which describes some of the relationships between form (perception, articulation, etc.) and meaning (concepts, intentions, thoughts, etc.; S1)”

There is a somewhat standard way to write the form/meaning tuples used to discuss the problem on page 3. Roots are written in small caps, with a surd; features are given using Leipzig-style glosses.

We choose not to give Leipzig-style glosses because such decompositions are not given to the system, and include information “for free” that our setup aims to infer. For example, in

the English past tense, the system does not get a gloss saying Stem+Past, as this would tell the system that the past tense is marked via a suffix (rather than a prefix).

Figure 1D is contentless; the point is better illustrated in prose.

Figure 1D has been removed.

It might be helpful, on page 3, to tell the reader what is lost by restricting to concatenative processes. What are types of non-concatenative processes, where can readers read about them, and how common are there? Are there processes that have concatenative and non-concatenative analyses? Etc.

Thank you for raising this. We now point the interested reader to resources on these other morphologies, and to mention at least one such morphology which is not covered by this restriction. We now say: "We also restrict ourselves to concatenative morphology, which builds words by concatenating stems, prefixes, and suffixes. Nonconcatenative morphologies [20]-such as Tagalog's reduplication, which copies syllables-are not handled." (lines 104-106. [20] is 'Representing Reduplication', Raimy's 1999 PhD thesis)

It seems from equation (1) that this is restricted to exactly one prefix and suffix, is that correct? What do we lose by this assumption? This seems like an enormous restriction on the grammar space. How would the authors work around it? This absolutely needs to be discussed.

You are correct that our problem setup assumes at most one prefix/suffix per inflection type (i.e., column of a paradigm). Equivalently, we assume every language is *fusional*. If a paradigm column has e.g. multiple prefixes, then the lexicon stores the concatenation of those prefixes. This allows the model to emulate multiple prefixes/suffixes by fusing them together. As a concrete example, imagine we have a paradigm matrix with three columns corresponding to (1) past tense, (2) feminine, and (3) past+feminine; and also imagine that the language marks each of these with a prefix. Then, the lexicon will store a prefix for [tense:Past]; another prefix for [gender:Female]; and a third prefix marking [tense:Past; gender:Female]. If this third prefix is a concatenation of the first two, then we can think of the model as "caching" the computation of the morphology of [tense:Past; gender:Female]. This "caching" process is implicit in our objective function, and serves to emulate having more than one prefix/suffix per paradigm column. We now explain this in the manuscript (lines 116-123):

ing" of words in the lexicon. Eq. 1 assumes *fusional* morphology: every distinct combination of inflections
fuses into a new prefix/suffix. This fusional assumption can emulate arbitrary concatenative morphology:
although each inflection seems to have a single prefix/suffix, the lexicon can implicitly "cache" concatena-
tions of morphemes. For instance, if the morpheme marking tense precedes the morpheme marking gender,
then $\mathbb{L}([\text{tense:PAST}; \text{gender:FEMALE}], \text{pfx})$ could equal $\mathbb{L}([\text{tense:PAST}], \text{pfx}) \cdot \mathbb{L}([\text{gender:FEMALE}], \text{pfx})$.

The discussion of features is somewhat confusing in that it's not made clear that a specification like +nasal denotes a set of phonemes which have that property. Nor is it made

clear that a specification is always interpreted conjunctively rather than disjunctively. Or is it the case the authors allow disjunctive feature specifications in their SMT problems?

Yes, it would be helpful for readers unfamiliar with phonological rules to clarify these issues. We now say: “Triggering environments specify combinations of features (characterizing sets of phonemes sometimes called natural classes)” (lines 142-144)

From where do we get the feature specifications? I mean this question both epistemologically (where do we suppose children get them? are they part of UG?), and in the literal sense of where do the authors get their feature vectors for segments?

We get our feature specifications from Odden’s phonology textbook, and one of our ablations contrasts using the simpler “acoustic” feature system introduced in the first half of the textbook with the modern “articulatory” feature system introduced in the second half of the textbook. The manuscript now describes this on lines 223-226 as: “Our simpler acoustic features come from the first half of a standard phonology text, while the articulatory features come from the latter half, so this comparison loosely models a contrast between novice and expert phonology students.” Epistemologically, we think of the features system as part of UG. Although this commitment is not essential to our work, we suggest this epistemological standpoint in Figure 2:

What happens if we just get rid of features and only work with segments? What do we lose? Is it possible to run the experiments again with a “dummy” (one-hot) “features” and measure how performance is impacted?

This is an interesting test to consider, and we actually described a very similar experiment in the initial submission. Our revision now clarifies this experiment, which is exactly what you describe but with the additional removal of Kleene star: “We can further remove two essential sources of representational power—Kleene star, which allows arbitrarily long-range dependencies, and phonological features, which allows analogizing and generalizing across phonemes. Removing these two renders only the simplest problems solvable” (lines 226-228).

Rather than saying (line 109) that the rules have the power of FSTs I would say instead that the rules, and the morphology + rules, correspond to 2-way rational functions, which in turn correspond to finite-state transducers.

Thank you for the suggestion. We now instead say: “When such rules are restricted to not be able to cyclically apply to their own output, the rules and morphology correspond to 2-way rational functions, which in turn correspond to finite-state transducers” (lines 150-151).

In standard jargon "morphosyntactic category" means {noun, verb, ...}. I would suggest the authors use some new jargon for the notion here.

We now use the new jargon “morphological category” instead of “morphosyntactic category” to avoid this naming collision.

What does it mean to treat Sketch as an oracle? From context it seems to me that the authors mean that we don't care about how long it takes or what its properties are, but I think the readers should care about those issues, very much.

Certainly considerations of runtime are important, especially when analyzing empirical results. For this reason, we have created the new Figure 5B which illustrates the runtime/accuracy trade-off for different systems. We have changed the text in the methods to say “Sketch can solve the following constrained optimization problem...” (line 349) rather than saying that Sketch acts as an oracle.

What does counter-example guided synthesis and test-driven synthesis mean? Give an intuitive explanation. I wasn't able to reason exactly how they work from figure 3.

The manuscript now provides the following intuitive explanations on lines 179-181: “[our synthesis algorithm] combines ideas from *counter-example guided inductive synthesis* (which alternates synthesis with a verifier that feeds new counterexamples to the synthesizer) with *test-driven synthesis* (which synthesizes new conditional branches for each such counterexample).”

What does it mean to "ablate...Kleene star"? I know what Kleene star is (though readers won't necessarily, so it should be defined) but I don't know what it would mean to ablate it.

We have removed all use of the term “ablate” from the main manuscript, because readers from outside the machine learning community may not be familiar with it. To “ablate” a component of an AI system means to remove it, typically for the purpose of studying its relative importance. We now say that we “remove” rather than “ablate” and describe Kleene star both in technical terms (line 147-149: “The subscript $_0$ denotes zero or more repetitions of a feature matrix, called the ‘Kleene star’ operator (i.e., $[+voice]_0$ means of zero or more repetitions of $[+voice]$ phonemes)”) and in intuitive terms with respect to its practical impact on learning (line 226: “Kleene star, which allows arbitrarily long-range dependencies”).

How do [27-30] work? The competing approaches deserve more than a few lines. They're said to scale poorly but as I have said earlier, we don't really have a characterization of the scaling properties of the proposed method either.

Indeed, these prior works deserve more discussion and in fact vary in terms of their techniques, strength, limitations, and problem statements. We now say the following:

a single computational model. Prior approaches either abandon theory induction by learning black-box
probabilistic models [32], or they induce interpretable models but do not scale to a wide range of challenging
and realistic data sets. These interpretable alternatives include unsupervised distributional learners, such as
the MDL genetic algorithm in Rasin et al. [33], which learns from raw word frequencies. Other interpretable
models leverage strong supervision: Albright et al. [34] learns rules from input-outputs, while [35] learns
finite state transducers in the same setting. Other works attain strong theoretical learning guarantees by
restricting the class of rules: e.g., Hua & Jardine 2021 [36] considers 2-input strictly local functions. These
interpretable approaches typically consider 2-3 simple rules at most. In contrast, Goldwater et al. [35] scales
to tens of rules on thousands of words by restricting itself to non-interacting local orthographic rules.

Similarly, the use of the word "opaque" is a poor choice of words here since that's a term of art in morphophonology and I don't think the authors mean that narrower sense.

We have replaced the term "opaque" with the term "black-box" (see first line of above screenshot)

The paragraph on lines 170-192 feels like it is out of the scope of this work. I also find the conclusion unconvincing. Why does the ability of a model to handle AGL and real data entail something about the cognitive resources tapped into AGL exactly?

Our argument is not that our ability to handle AGL entails something about the relationship between actual language acquisition and AGL phenomena. Instead, our argument is flipped: that if there is the relationship between actual language acquisition and AGL, then methods applicable to natural language should demonstrate some transfer to AGL. Here we focused on a certain classic AGL studies, aiming to show that the proposed method captures one of their key findings: namely, the extreme sample-efficiency of human learners in these simple settings. Previously, and in the current resubmission, we introduce this section with: "If our model captures aspects of linguistic analysis from naturalistic data, and assuming linguists and children confront similar problems, then our approach should extend to model at least some aspects of the child's linguistic generalization" (lines 237-239), which we believe clarifies that a relationship between AGL and natural language entails something for our model, not vice versa. We've modified the following text to clarify that our simulations here are motivated by the low-data, few-shot feature of this setup. We now say (line 245-247): "These AGL stimuli contain very little data, and thus these few-shot learning problems admit a broad range of possible generalizations. Children select from this space of possible generalizations to select the linguistically plausible ones", which connects to Figures 6-7 where we studied generalization as a function of number of examples.

Lezgian is argued to in fact voice final obstruents. Yu (2004) provides phonetic evidence and discusses how this rule came about. So this is an unfortunate example on the paragraph beginning 193-204.

Thank you for the catch! We have mentioned that there are some languages which voice final obstruents, and used the example you provided. We feel that this is actually a good example of the broader point we're making: that these are *typological tendencies*, not absolute universals - hence, we are not in any sense "learning UG" (mentioned on lines 295-297), which says "To be clear, our mechanized meta-theorizing is not an attempt to "learn universal grammar" (cf. Perfors et al. 2011 [45]). Rather than capture a learning process, our meta-theorizing is analogous to a discovery process that distills knowledge of typological tendencies").

I found the physics example incomprehensible. Is this going to be useful to the target reader? It seems that it would be fine to say that this kind of heuristic reasoning is common among scientists and technicians.

Previously we had illustrated this point by constructing a deliberately implausible physics function, and remarked that one would never find this in practice (a hypothetical "8-way sinusoidal potential"). The current revision opts to instead use examples of common functions in physics, rather than a contrived implausible function, saying "physicists know which potential energy functions tend to occur in practice (radially symmetric, pairwise, etc.)" (line 271)

There is reason to think that the Polish o-raising rule is not a phonological rule at all; see Buckley 2001 and subsequent work. I just mention this in case the authors want to use a less controversial example in figure 7.

Thank you for pointing this out. While this alternative analysis is possible, our focus in this Pareto frontier is to illustrate another alternative analysis of this data. This alternative analysis is particularly salient when the data is viewed as a textbook problem, for which the Buckley analysis would be out of scope. We now clarify why we chose this example in the caption to figure 7, namely, that our system can be used for 'linguistic debugging' of these datasets: "We show the Polish problem because the textbook author accidentally chose data with an unintended extra pattern: all stems vowels are /o/ or /u/, which the upper left solution encodes via an insertion rule. Although the Polish MAP solution is correct, the Pareto frontier can reveal other possible analyses such as this one, thereby serving as a kind of 'linguistic debugging'."

Where does "phonological commonsense (sic)" reside? Is it something working linguists have? Do kids have it too? I think the authors should take a stance on this.

Excellent point to clarify. The revision now says: "we believe that children possess implicit knowledge of these and other tendencies, which contributes to their skill as language learners. Similarly, we believe the linguist's skill in analysis draws on an explicit

understanding of these and other cross-linguistic trends" (line 297-300, where "these and other tendencies", in the context here, refers to phonological common sense).

The authors should explain what a Pareto front is. I suspect the vast majority of readers will not be familiar with this concept.

We now clarify the setup of Pareto optimality within this context by saying the following: "Here parsimony means size of rules and affixes (the prior in Eq. 2); fit to data means average stem size (the likelihood in Eq. 2); and a Pareto-optimal solution is one which is not worse than any other along both these competing axes." (lines 249-2511; Eq 2 is a new equation which refactors our learning objective to be more explicitly expressed in terms of "parsimony" and "data fit" terms).

I don't think it's fair to call "distributional semantics" (a misnomer) a model of semantics. It is entirely unclear how one goes from word2vec or BERT to inferential semantics.

Thank you for raising this point. It is true that what is meant by "distributional semantics" (as primarily used in NLP) is qualitatively very different from semantics as studied as part of linguistics – in its goals, empirical scope, and formal machinery. It is also true that while BERT and similar models are often used in "semantic" applications, they are also used in many other applications in NLP. We have therefore changed the paper to read "distributional models of language structure" (line 325).

There is no discussion about the impact of this work for field linguistics/language documentation.

There is no discussion about the impact of this work for deciding between extensionally-equivalent analyses of the same language. The proper analysis of the Catalan phenomenon mentioned herein has been debated for many years; does the BPL approach, or the use of Pareto fronts, help resolve the question in any way? See Hale & Reiss 2008:271f. for discussion of this particular example.

More specifically, what is this model's take on absolute neutralization? Opacity? Can it tell us whether Russian yers are epenthesized or deleted?

We are excited about using this work to explore questions such as these. More specifically, our work can be used to map out alternative analyses under different objective functions (such as using Pareto frontiers), or under different choices for various components of UG (such as under different feature systems, which we explore in the paper). Indeed, the specific case of Russian yers is actually one that we had been discussing exploring, and we recently presented a non-archival poster at CogSci on using this work to study and computationally model the learnability of opaque interactions as a function of process type. Publishing this paper and the corresponding software—either in this journal or another—can help spur such follow-up work with collaborators and other colleagues. Motivated by both your comment and our ongoing work, we have added the following to the discussion (lines 310-314):

Within phonology, our work offers a computational tool which can be used to study different grammatical
hypotheses: mapping and scoring analyses under different objective functions, and studying the implications
of different inductive biases and representations across a suite of languages. This toolkit can spur quantitative
studies of classic phonological problems, such as probing extensionally-equivalent analyses (eg distinguishing
deletion from epenthesis).

There could be more discussion about what else could be done to deploy this class of model for morpho-phonology? The careless reader might take away from this that the problem is solved, but that does not seem to be the case.

We have added the following text to the introduction (lines 76-79) to clarify the scope of problems we considered within morphophonology: “These data sets come from phonology textbooks: they have high linguistic diversity, but are much simpler than full language learning, with tens to hundreds of words at most, and typically isolate just a handful of grammatical phenomena.”

To further avoid confusion, the discussion reiterates this point by saying (line 322-323) “although the textbook problems we solve are harder than prior work tackles, full morpho-phonology remains larger and more intricate than the problems considered here.”

What if any hyperparameters are important for the use of Sketch? For instance the authors mention the use of minibatches: how big are they and does it matter?

We automatically set the minibatch size such that up to 9 surface forms enter each batch, subject to the constraint that all inflections of each new lexeme enter the batch. As also explained in our response to Reviewer 3, the supplement now explains this 9-form minibatch setting and its implications: “We conjecture that larger batch size generally leads to better convergence, because this exposes the SAT solver to more data at once, which on balance should lead to less myopic search moves. Yet larger batch sizes increase compute requirements, both because the size of the SAT problem grows linearly with batch size and because the search radius may need to grow larger with increased batch size. Accordingly, for allophony alternation problems, we batch the entire problem at once, because these problems are much easier. Our selection of a minibatch size of 9 was motivated by informal pilot experiments suggesting that after around 9 new words the solver performance degraded severely; due to the high compute cost of running these simulations, we did not perform a systematic hyperparameter sweep, and the ‘optimal’ batch size may differ from the one used.” (Supplement, lines 110-112)

Additionally, Sketch works by “finitizing” the space of program execution traces - for example, the maximum number of loop iterations must be upper bounded, and so there are a small number of “finitizing” hyperparameters. Supplement Footnote #2 now explains this hyperparameter as follows: “Sketch requires further hyperparameters relating to converting an infinite program space to a finite one, such as an upper bound on the number of loop iterations. We set such upper bounds automatically so that the system can handle the longest surface form in the data: the loop iteration bound is the length of the longest

surface form plus two, where +2 comes from a +1 buffer and another +1 due to implementation details.”

A few finicky details:

* The linguistic standard is to write (Roman) words used as examples (like "horse" on page 2) in italics, and give the gloss of all non-English words in single quotes after. The examples are instead double-quoted; the Serbo-Croat words are not glossed in the body but they are given beforehand (instead of after) in single (instead of double) quotes.

Thank you - we've adopted the convention of putting glosses in italics.

* The authors should put their "minus" characters (e.g., in feature bundles) in math mode. As is they look like short hyphens, not minus signs.

Fixed!

* The term "ablating", used throughout, should be defined on first use. It has a common sense (to burn away) that doesn't immediately match the one used in machine learning...

We have fixed this by ablating our use of the word ablation (opting for "remove" instead) within the main text. Lines 177-199 of the Supplement discusses our ablations in more detail, though, and we felt it was more natural to keep the word "ablation" for that supplemental section. Footnote 3 of the supplement defines the term as: "An *ablation* is a variant of a system with components removed entirely or changed to be less powerful. Ablations are studied to understand the importance of the ablated components."

* "Bayes's": at the risk of being prescriptive, is this not normally written Bayes'?

* Footnote 1 is missing sentential punctuation.

* "distill out": the particle "out" seems redundant to me.

* line 112: "theorieseven"

* "context-sensitive rewrite": I would suggest a more standard jargon "context-dependent rewrite", so to avoid the confusion with the family of context-sensitive languages.

* "commonsense": "common sense", I think.

Thank you, these have all been fixed in the revision as you suggested.

* The Turkish example on lines 147f. is discussed too briefly for the reader to easily grasp it. Perhaps segmentation would help. There is also confusion about the use of square vs. angled brackets here.

The revision now explicitly segments that example into stems and suffixes by inserting a hyphen, and explains that this hyphen indicates suffix boundaries.

The parenthetical on line 152 should be foregrounded at the start of the paragraph, it's a very important point.

This parenthetical was explaining that we are not evaluating rules. In the new manuscript, we removed that parenthetical because we now include a rule evaluation.

** Stampe's dissertation is dated 1969; a printed version (under the name "A dissertation in natural phonology", published by Garland) appeared in 1973.*

Thanks for the catch - we ended up removing this citation in the updated manuscript when we simplified our discussion of the role of phonological common sense.

** The language described as "Yawelmani" is probably more properly called "Yowlumne". (I sometimes cite this as something like "Yowlumne, formerly known as Yawelmani".) This is discussed in Weigel's 2005 Berkeley dissertation.*

Thank you for pointing this out. We now use the proper name and in the caption we now say: "'Yowlumne" was formerly known as "Yawelmani" [1], where [1] is Weigel's dissertation.

REVIEWER COMMENTS

Reviewer #2 (Remarks to the Author):

The authors have addressed all my remaining concerns. I also read through the replies to all other reviews, and find the revisions appropriate. I think this work will make a valuable contribution to the literature, and be of broad interest.

Reviewer #3 (Remarks to the Author):

My main concerns about the previous version of the paper were about the framing and the insufficient explanation of many of the technical aspects. I feel that the framing problems have been largely addressed, and there is definitely progress on the technical explanations, but unfortunately there is still a good way to go with those: I am still struggling to understand parts of the mathematical exposition.

Comments about terminology, model formulation and exposition

While there has been some improvement in the consistency of terminology, there are still inconsistencies about whether morphology is part of the Lexicon or part of the Theory/rules. Most of the text on p3-4 and Eqs 1-2 treat the lexicon as containing stems and affixes, and the 'rules' as being phonological (except lines 97-98, where "grammatical rules" implies the Theory could include more rules than just phonological ones). Yet the illustration in Fig 2 strongly implies (though is not explicit) that the lexicon only contains stems, and the affixes are actually applied as morphological rules in part of the Theory. Then Fig 4 (and others in Supp) explicitly say the morphology is part of the Theory. I think what's really going on is that the stems and affixes *are* in the lexicon, and there are concatenation rules in the model (maybe these are part of what you call S in the Supplement?), but unlike the phonological rules these are not learned, they are fixed. Showing Morphology as part of the Theory is very confusing, since it implies these rules would need to be covered by the prior, yet the prior only seems to cover phonology rules and lexical items.

You have stated the concept of a "paradigm matrix" in words, but it would really help to give an actual example, especially for readers who are not already familiar with the task and textbook problems. Can you make the "observed data" in Fig 2 actually look like a partial paradigm matrix? (If you need to get rid of one of the languages to make it fit, that would be ok!) You could also give a complete matrix in S3.1.

The parts added on p4 (Eq2 and surrounding paragraph) and p 15-16 (Method for AGL) help a lot in understanding what you did, and I now understand that in some circumstances there are multiple stems you may be marginalizing over. But the way you've formulated this is still not very clear: it seems to me that the likelihood is $P(X | T, L)$ [you did write this in Eq 11, but not around Eq 2], but in practice you have chosen to leave T and part of L (ie the affixes) fixed (using the MAP solution based on the part of the problem seen so far?) while marginalizing over possible stems. You've also written $P(X)$ as if it doesn't condition on anything, which would imply a fully marginalized likelihood, yet as noted you are not marginalizing out most of T, L. Is there some theoretical reason why you marginalize over stems but not rules or affixes, or is this purely practical? Maybe use the term "marginal likelihood" or "partially marginalized likelihood" and say what you're marginalizing over and why that (and only that)?

I realize these confusions re Eq 2 are quite technical and I wonder if rather than trying to explain this more clearly where it is currently in the paper, it would be better to move Eq 2 and related technical explanation to the AGL section of the Method section where you actually need to use it, leaving only a brief mention that there is an alternative Bayesian interpretation (see Methods) is used to produce

predictions for the AGL experiments and to formulate the meta-theory.

Section S3:

The added example in S3.3 is helpful, but as noted above, I would also like to see a concrete example of an input paradigm matrix in S3.1, and ideally this could be used as a running example. Perhaps even work through the prior probability/cost of some iteration of that example in S3.4, which would finally clarify which bits count as "lexicon" and which are "rules"!

Allophony problems: you don't really explain how these are fit into the framework, since there are no affixes (no paradigm matrix). What leads to there being a counterexample to require a rule to be learned? Put another way: according to the prior, a solution with all URs = SRs is actually **better** than one with a phonological rule, since both lexicons have the same word lengths. So do you constrain the learner to only consider solutions where one or the other of the given allphone pair are in URs? How is this done? Please explain, including a concrete example of the input in S3.1.

In S3.5, please list the phonetic features and articulatory features. Please also refer to this list in the main text (lines 224-225).

Other:

Other points about Figures: After sorting out the issue about whether morphology is Theory or Lexicon, consider saying explicitly in Fig 2 which "box" is Theory and which is Lexicon. I also find the purple and dark aqua in Fig 2 (and other figures) very difficult to distinguish from black.

Lines 124-125: what you describe as the rows and columns are reversed.

Line 148: means of \rightarrow means

It seems to make more sense to have Fig 6 come before Fig 5, since it is more basic and is also referred to first in the main text. I suggest referencing it first in the Supplement too, and reminding readers what it is at that point.

I don't think d or D in Eq 11 are defined anywhere.

It might also be useful to define the $\mathds{1}$ symbol.

Reviewer #4 (Remarks to the Author):

It seems to me the authors have addressed nearly all of the concerns raised in my review. Indeed, I am pleasantly impressed that they took as many of my suggestions as seriously as they did and that the text reflects what I hope are improvements to the manuscript.

We are grateful for the continued input from the reviewers. We have revised the manuscript to take this feedback into account, particularly the input from Reviewer 3. Our revision highlights changes in blue. Below, we show the comments of Reviewer 3 in italics and our responses in blue.

Reviewer 3:

My main concerns about the previous version of the paper were about the framing and the insufficient explanation of many of the technical aspects. I feel that the framing problems have been largely addressed, and there is definitely progress on the technical explanations, but unfortunately there is still a good way to go with those: I am still struggling to understand parts of the mathematical exposition.

Comments about terminology, model formulation and exposition

*While there has been some improvement in the consistency of terminology, there are still inconsistencies about whether morphology is part of the Lexicon or part of the Theory/rules. Most of the text on p3-4 and Eqs 1-2 treat the lexicon as containing stems and affixes, and the 'rules' as being phonological (except lines 97-98, where "grammatical rules" implies the Theory could include more rules than just phonological ones). Yet the illustration in Fig 2 strongly implies (though is not explicit) that the lexicon only contains stems, and the affixes are actually applied as morphological rules in part of the Theory. Then Fig 4 (and others in Supp) explicitly say the morphology is part of the Theory. I think what's really going on is that the stems and affixes *are* in the lexicon, and there are concatenation rules in the model (maybe these are part of what you call S in the Supplement?), but unlike the phonological rules these are not learned, they are fixed. Showing Morphology as part of the Theory is very confusing, since it implies these rules would need to be covered by the prior, yet the prior only seems to cover phonology rules and lexical items.*

You have stated the concept of a "paradigm matrix" in words, but it would really help to give an actual example, especially for readers who are not already familiar with the task and textbook problems. Can you make the "observed data" in Fig 2 actually look like a partial paradigm matrix? (If you need to get rid of one of the languages to make it fit, that would be ok!) You could also give a complete matrix in S3.1.

Thank you for suggesting these clarifications. You are correct that affixes live in the lexicon, and that concatenation rules correspond to S in the supplement. We have revised the supplement (lines 47-48) to say "In our model, S corresponds to concatenation of morphemes." To avoid confusion, we have revised Figure 2 to (1) place the affixes alongside the stems in a box now labeled "lexicon"; (2) illustrate example paradigm matrices; (3) explain in the caption that the lexicon contains both stems and affixes. Figure 4, and the analogous supplementary figures (S1-3), have also been changed to place the affixes in the

lexicon. For example, below we show the new revised Figure 2, as well as how the caption has been changed to clarify these issues:

Figure 2: The generative model underlying our approach. We infer grammars for a range of languages, given only form/meaning pairs (orange) and a space of programs (purple). Form/meaning pairs are typically arranged in a stem x inflection matrix. For example, the lower right matrix entry for Catalan means we observe the form/meaning pair $\langle /grizə/, [\text{stem:GREY}; \text{gender:FEM}] \rangle$. Grammars include phonology, which transforms concatenations of stems and affixes into the observed surface forms using a sequence of ordered rules, labeled r_1, r_2 , etc. The grammar's lexicon contains stems, prefixes, and suffixes, and morphology concatenates different suffixes/prefixes to each stem for each inflection. ϵ refers to the empty string. Each

The parts added on p4 (Eq2 and surrounding paragraph) and p 15-16 (Method for AGL) help a lot in understanding what you did, and I now understand that in some circumstances there are multiple stems you may be marginalizing over. But the way you've formulated this is still not very clear: it seems to me that the likelihood is $P(X | T, L)$ [you did write this in Eq 11, but not around Eq 2], but in practice you have chosen to leave T and part of L (ie the affixes) fixed (using the MAP solution based on the part of the problem seen so far?) while marginalizing over possible stems. You've also written $P(X)$ as if it doesn't condition on

anything, which would imply a fully marginalized likelihood, yet as noted you are not marginalizing out most of T, L . Is there some theoretical reason why you marginalize over stems but not rules or affixes, or is this purely practical? Maybe use the term "marginal likelihood" or "partially marginalized likelihood" and say what you're marginalizing over and why that (and only that)?

I realize these confusions re Eq 2 are quite technical and I wonder if rather than trying to explain this more clearly where it is currently in the paper, it would be better to move Eq 2 and related technical explanation to the AGL section of the Method section where you actually need to use it, leaving only a brief mention that there is an alternative Bayesian interpretation (see Methods) is used to produce predictions for the AGL experiments and to formulate the meta-theory.

Thank you for raising these issues, which are central as to why probabilistic scoring of new forms is made more straightforward when we treat affixes as distinct from stems. Linguistic theory typically lumps affixes and stems together into the same object—the lexicon. However, because the rules are deterministic, it would not be possible to define a (nondegenerate) probabilistic generative model in terms of the rules and lexicon. Instead we are defining a probabilistic generative model over paradigm rows in terms of the affixes and rules. Intuitively, this allows the model to learn new stems on-the-fly in the context of existing grammatical knowledge, because it can place probabilities on unseen stems.

To clarify these issues and streamline the main text, we have moved the probabilistic formulation in Eq 2 and its related technical discussion to Methods (lines 371-386). We have also streamlined and refactored the description of the probabilistic formulation. Lines 126-127 of the main text now call out to this formulation by saying that "an equivalent Bayesian framing (Methods) permits probabilistic scoring of new stems by treating the rules and affixes as a generative model over paradigm rows." We hope that these changes better allow the reader to understand the technical core of our model—both the simpler 0/1 likelihood formulation, and the more manifestly probabilistic formulation which treats affixes and stems differently. We also intend that these changes better motivate why we are considering both formulations: so that our model can do inference over (ie probabilistically score) new stems and their corresponding paradigm rows. This functionality is essential for acquiring new words in the context of existing grammatical knowledge.

Section S3:

The added example in S3.3 is helpful, but as noted above, I would also like to see a concrete example of an input paradigm matrix in S3.1, and ideally this could be used as a running example. Perhaps even work through the prior probability/cost of some iteration of that example in S3.4, which would finally clarify which bits count as "lexicon" and which are "rules"! We revised the manuscript to provide an example paradigm matrix in the supplement (Fig S7A), which is described in S3.1 and referred to in the main text on line 126. In S3.4, which specifies priors over rules and lexica, we have stepped through an example calculation for a

solution to the data in the new Fig S7A (supplemental lines 181-193). Additionally, the new S7B includes an example allophony problem. The new Figure S7 is as follows:

A			B (i)		B (ii)	
	Nominative	Genitive		Surface form	Allophones 1	Allophones 2
wagon	vagon	vagona	pigeon	olide?		
car	avtomobil ^l	avtomobil ^l a	hide it! (sg.)	zahset	b	p
evening	vet ^l er	vet ^l era	stocking	galis	d	t
husband	mu ^l	mu ^l ža	tail	odahsa	g	k
pencil	karandaf ^l	karandafa	five	wisk		
eye	glas	glaza	two	degeni		
threshold	porok	poroga	Abraham	aplam		

Figure S7: **A.** An example paradigm matrix from Russian, which is a common format for the input data **X**. Columns correspond to different inflections (nominative, genitive) while the rows correspond to different stems (*wagon*, *car*, *evening*, ...). For example, [vagona] means *wagon* in the genitive form. **B.** Example allophone problem (a subset of data from Mohawk). A set of surface forms are given **(i)** together with pairs of possible allophones **(ii)**, and the challenge is to find rule(s) which can predict one set of allophones from the other. There are 3 allophone pairs in this example.

*Allophony problems: you don't really explain how these are fit into the framework, since there are no affixes (no paradigm matrix). What leads to there being a counterexample to require a rule to be learned? Put another way: according to the prior, a solution with all URs = SRs is actually *better* than one with a phonological rule, since both lexicons have the same word lengths. So do you constrain the learner to only consider solutions where one or the other of the given allphone pair are in URs? How is this done? Please explain, including a concrete example of the input in S3.1.*

Thank you for inviting these additions. Textbook allophony problems come with an allophone substitution—for example, a problem might say that **g** and **k**, as well as **d** and **t**, are allophones, which we think of as a substitution mapping **g**->**k** and **d**->**t**. We constrain the lexicon to either not use phonemes from the domain of this substitution (eg {**g**, **d**}), or not use phonemes from the range of this substitution (eg {**k**, **t**}). Methods has been revised to explain this (lines 357-369):

Allophony problems
Allophony problems comprise the observed form-meaning set **X**, as well as a *substitution*, which is a partial
 function mapping phonemes to phonemes (see S3.1). This mapping operates over phonemes called ‘allo-
 phones.’ The goal of the model is to recover rule(s) which predicts which element of each allophone pair
 is an underlying form, and which is merely an allophone. The underlying phonemes are allowed in the
 lexicon, while the other allophones are not allowed in the lexicon and surface only due phonological rules.
 For example, an allophony substitution could be $\{b \mapsto p, d \mapsto t, g \mapsto k\}$. We extend such substitutions to
 total functions on phoneme sequences by applying the substitution to phonemes in its domain, and not ap-
 plying it otherwise. We call this total function $s(\cdot)$. For instance, using the previous example substitution,
 $s(abkpg) = apkpk$. Solving an allophone problem means finding rules which either map the domain of $s(\cdot)$
 to its range (\mathbb{T}_1 below), or vice versa (\mathbb{T}_2 below):

$$\begin{aligned}
 \quad & \mathbb{L}_1(m) = s(f) \text{ when } \exists \langle f, m \rangle \in \mathbf{X} & \mathbb{L}_2(m) = s^{-1}(f) \text{ when } \exists \langle f, m \rangle \in \mathbf{X} \\
 \quad & \text{For each } i \in \{1, 2\}: \quad \mathbb{T}_i = \arg \max_{\mathbb{T}} P(\mathbb{T}|\mathcal{UG})P(\mathbf{X}|\mathbb{T}, \mathbb{L}_i) \left(= \arg \max_{\mathbb{T}} P(\mathbf{X}, \mathbb{T}, \mathbb{L}_i|\mathcal{UG}) \right) \quad (7)
 \end{aligned}$$

The supplemental text explaining input data formats has also been revised accordingly (supplement lines 80-90):

**S3 Supplemental Methods**
**S3.1 Input data format**
With the exception of allophony problems, each textbook problem consists of a matrix of surface forms,
where the columns correspond to different inflections, and the rows correspond to different stems. Matrix
entries can be empty (unspecified), either due to missing data, or due to a particular inflection not applying
to a particular stem (for example, there is no past tense form of a stem like “pineapple” because it is a noun
and not a verb). Slightly overloading notation used in the main manuscript, we refer to this matrix as **X**,
and can index the rows of **X** (lexemes) and the columns of **X** (inflections). For example, Figure S7A shows
a paradigm matrix **X** for a basic problem from Russian. There are 2 inflections (nominative and genitive),
corresponding to the columns of the matrix, and 4 stems, corresponding to the rows of the matrix.
Allophony problems consist of a set of surface forms along with a set of pairs of phonemes, known as
‘allophones’, which we treat as a substitution on phonemes. Figure S7B shows an allophony problem from
Mohawk.

(The new Figure S7 is shown in the previous response, which includes the example Mohawk problem.)

In S3.5, please list the phonetic features and articulatory features. Please also refer to this list in the main text (lines 224-225).

We now list the phonetic and articulatory features in the supplement where you suggest (supplement lines 206-213), and call out to it in the main text on line 216.

Other:

Other points about Figures: After sorting out the issue about whether morphology is Theory or Lexicon, consider saying explicitly in Fig 2 which "box" is Theory and which is Lexicon.

Thank you for the suggestion. We have explicitly labeled the boxes “Theory” (containing phonological rules) and “Lexicon” (containing affixes and stems), as shown earlier in this response.

I also find the purple and dark aqua in Fig 2 (and other figures) very difficult to distinguish from black.

We have changed the color scheme, and the colors should now be much easier to distinguish both from each other, and from the color black.

Lines 124-125: what you describe as the rows and columns are reversed.

Line 148: means of -> means

Fixed!

It seems to make more sense to have Fig 6 come before Fig 5, since it is more basic and is also referred to first in the main text. I suggest referencing it first in the Supplement too, and reminding readers what it is at that point.

We have taken your suggestion, thanks! The revision now exchanges the order of those supplemental figures, which puts the non-learned CFG before its learned counterpart. We have also revised their corresponding supplemental text (supplement lines 73-77) to remind the reader of the relationship between these two grammars, and what role each plays.

I don't think d or D in Eq 11 are defined anywhere.

Thank you for the catch—we have fixed Methods to specify these definitions on line 404.

It might also be useful to define the $\mathds{1}$ symbol.

We have now defined indicator functions on line 115-116.

REVIEWERS' COMMENTS

Reviewer #3 (Remarks to the Author):

Thank you for addressing my remaining concerns, the clarified equations and added examples are very useful. I have no further requests, except to point out a couple of minor errors in the example in S3.4 which should be fixed:

Line 185+: the final pfx should be sfx

Lines 188-189: most of these words are actually the genitive forms rather than the stems.

We appreciate that Reviewer #3 has continued to provide input on our manuscript. Below we show the reviewer comments in italics and our response in blue:

Reviewer #3 (Remarks to the Author):

Thank you for addressing my remaining concerns, the clarified equations and added examples are very useful. I have no further requests, except to point out a couple of minor errors in the example in S3.4 which should be fixed:

Line 185+: the final pfx should be sfx

Lines 188-189: most of these words are actually the genitive forms rather than the stems.

Thank you for the catches! We have fixed these in the latest revision. We now correctly label the last affix as being a suffix:

In solving Fig. S7A the system also constructs a lexicon, which participates in the prior probability calculation. This contains affixes for the nominative and genitive:

$$\mathbf{L} \supset \{ \langle \epsilon, \text{pfx}, \text{NOMINATIVE} \rangle, \langle \epsilon, \text{pfx}, \text{GENITIVE} \rangle, \langle \epsilon, \text{sfx}, \text{NOMINATIVE} \rangle, \langle \text{a}, \text{sfx}, \text{GENITIVE} \rangle \}$$

And have substituted stems for the erroneously included genitive forms:

$$\mathbf{L} \supset \{ \langle \text{vagon}, \text{stem}, \text{WAGON} \rangle, \langle \text{avtomobil}^{\text{f}}, \text{stem}, \text{CAR} \rangle, \langle \text{vet}^{\text{f}}\text{er}, \text{stem}, \text{EVENING} \rangle, \\ \langle \text{mu}_3, \text{stem}, \text{HUSBAND} \rangle, \langle \text{karandaf}, \text{stem}, \text{PENCIL} \rangle, \langle \text{glaz}, \text{stem}, \text{EYE} \rangle, \langle \text{porog}, \text{stem}, \text{THRESHOLD} \rangle \}$$